# The Limits of Predicting Agents from Behaviour

**Alexis Bellot** [1]   **Jonathan Richens** [1]   **Tom Everitt** [1]

## Abstract

As the complexity of AI systems and their interactions with the world increases, generating explanations for their behaviour is important for safely deploying AI. For agents, the most natural abstractions for predicting behaviour attribute beliefs, intentions and goals to the system. If an agent behaves as if it has a certain goal or belief, then we can make reasonable predictions about how it will behave in novel situations, including those where comprehensive safety evaluations are untenable. How well can we infer an agent's beliefs from their behaviour, and how reliably can these inferred beliefs predict the agent's behaviour in novel situations? We provide a precise answer to this question under the assumption that the agent's behaviour is guided by a world model. Our contribution is the derivation of novel bounds on the agent's behaviour in new (unseen) deployment environments, which represent a theoretical limit for predicting intentional agents from behavioural data alone. We discuss the implications of these results for several research areas including fairness and safety.

## 1. Introduction

Humans understand each other through the use of abstractions. We explain our intentions by appealing to our "goals" and "beliefs" about the world around us without knowing the underlying cognition going on inside our heads. According to Dennett (1989; 2017) the same is true of our understanding of other systems. For example, a bear hibernates during winter *as if* it believes that the lower temperatures cause food scarcity. This is a useful description of the bear's behaviour, with real predictive power. For example, it gives us (human observers) the ability to anticipate how bears might act as the climate changes. There is a correspondence between beliefs and behaviour that is foundational to rational agents (Davidson, 1963).

Artificial Intelligence (AI) systems appear to have similarly general capabilities, not totally unlike that of humans and animals. They can generate text that is fluent and accurate in response to a very diverse set of questions. Whenever they display consistent types of behaviour across many different tasks, we are tempted to apply our own mentalistic language more or less at face value (Shanahan, 2024), taking seriously questions such as: What do the AIs know? What do they think, and believe? Taking the analogy further, it is *as if* they learn "world models" that mirror the causal relationships of the environment they are trained on, guiding their future plans and behaviour[1]. And as a consequence, their interactions with an environment will leave clues that might give us the ability to predict their future behaviour in novel domains. This possibility engages with a core AI Safety problem: how to guarantee and predict whether AI systems will act safely and beneficially?

The main result of this paper is to offer a new perspective on this problem by showing that:

> *With an assumption of competence and optimality, the behaviour of AI systems partially determines their actions in novel environments.*

Here behaviour means our observations of the decisions made by the AI system, contextual variables, and utility or reward values in some environment. The "partial" determination of actions in new environments is a consequence of our lack of knowledge about the AI's actual world model (different models may induce different optimal actions). However, even though we can't uniquely identify the AI's future behaviour and beliefs, we can narrow it down to a range of possible outcomes. This paper characterises those outcomes.

---

[1] Recent research suggests that an AI's behaviour, to the extent that it is consistent with rationality axioms, can be formally described by a (causal) world model (Halpern & Piermont, 2024). The same conclusion can also be obtained for AIs capable of solving tasks in multiple environments (Richens & Everitt, 2024). For large language models, there is increasing empirical evidence for the "world model" hypothesis, see e.g., Toshniwal et al., 2022; Li et al., 2022; Gurnee & Tegmark, 2023; Goldstein & Levinstein, 2024 and Vafa et al., 2024.

---

[1] Google DeepMind. Correspondence to: Alexis Bellot <abellot@google.com>.

*Proceedings of the $42^{nd}$ International Conference on Machine Learning*, Vancouver, Canada. PMLR 267, 2025. Copyright 2025 by the author(s).

In the literature, the under-determination of agent "beliefs" and "preferences" has been considered in the fields of inverse reinforcement learning (Abbeel & Ng, 2004; Skalse & Abate, 2023; Amin & Singh, 2016) and decision theory (Savage, 1972; Afriat, 1967; Jeffrey, 1990), among others. In settings with distribution shift between training and deployment environments, this under-determination can be understood as a consequence of the Causal Hierarchy Theorem, that defines precise limits on the kinds of inferences that can be drawn across domains (Bareinboim et al., 2022; Pearl, 2009). It implies, for example, that behaviour in an environment subject to an intervention cannot be established from "non-interventional" data alone. Robins (1989), Manski (1990) and Pearl (1999) showed that useful information in the form of bounds can nevertheless be extracted from "non-interventional" data, without actually knowing the underlying data-generating process. In the causality literature, several methods and algorithms exist to solve different versions of this problem, see e.g., Balke & Pearl, 1997; Tian & Pearl, 2000; Zhang et al., 2021; Bellot, 2024; Rosenbaum et al., 2010; Tan, 2006.

This paper extends the causal formalism to reason about the possible behaviours and beliefs of an AI system, itself assumed to be governed by an unknown data generating process or world model. With this interpretation we are able to define mathematically notions such as an AI's preferred choice of action in novel environments, its perception of fairness, and its perception of harm due to the actions it takes. Our main contribution is a set of inequalities on these "beliefs" in terms of quantities that can in principle be estimated from behavioural data, and that hold irrespective of the underlying cognitive architecture of the AI system as long as it can be *represented* by a well-defined set of causal mechanisms (a world model) that tracks its behaviour (Sec. 4). We then extend these results to characterize AI behaviour under several relaxations for applications in practice (Sec. 5), ultimately with the goal of defining the theoretical limits of what can be inferred from data about AI behaviour in new (unseen) environments.

This has consequences for the wider AI Safety community and society. For example, we show that an AI's perception of the potential fairness and harm of its decisions (e.g., whether the AI's resource allocation is believed to be equitable, or its generations unbiased) can provably not be inferred from observing its behaviour alone. There are theoretical limits to how much we can understand about an AI's cognition and decision-making process from observations. We believe our results can help justify the claim that the design and inference of world models is important to ensure AIs can behave predictably and act safely and beneficially, as argued by Dalrymple et al., 2024; Legg, 2023; Bengio et al., 2025.

## 2. Preliminaries

In this section we outline some basic principles that we use to reason about how beliefs might be (implicitly) defined within an AI system.

We use capital letters to denote variables $(X)$, small letters for their values $(x)$, bold letters for sets of variables $(\boldsymbol{X})$ and their values $(\boldsymbol{x})$, and use supp to denote their domains of definition $(x \in \text{supp}_X)$. To denote $P(\boldsymbol{Y} = \boldsymbol{y} \mid \boldsymbol{X} = \boldsymbol{x})$, we sometimes use the shorthand $P(\boldsymbol{y} \mid \boldsymbol{x})$. We use $\mathbb{1}_{\{\cdot\}}$ for the indicator function equal to $1$ if the statement in $\{\cdot\}$ evaluates to true, and equal to $0$ otherwise.

Actions, plans, and hypothetical outcomes can be evaluated by symbolic operations on a model that represents the functional relationships in the world, known as a Structural Causal Model (Pearl, 2009, Definition 7.1.1), or SCM for short.

**Definition 1** (Structural Causal Model). *An SCM $\mathcal{M}$ is a tuple $\mathcal{M} = \langle \boldsymbol{V}, \boldsymbol{U}, \mathcal{F}, P \rangle$ where each observed variable $V \in \boldsymbol{V}$ is a deterministic function of a subset of variables $\boldsymbol{Pa}_V \subset \boldsymbol{V}$ and latent variables $\boldsymbol{U}_V \subset \boldsymbol{U}$, i.e., $v := f_V(\boldsymbol{pa}_V, \boldsymbol{u}_V), f_V \in \mathcal{F}$. Each latent variable $U \in \boldsymbol{U}$ is distributed according to a probability measure $P(u)$. We assume the model to be recursive, i.e., that there are no cyclic dependencies among the variables.*

In an SCM $\mathcal{M}$, each draw $\boldsymbol{u} \sim P(\boldsymbol{u})$ evaluates to a potential response $\boldsymbol{Y}(\boldsymbol{u}) = \boldsymbol{y}$ and entails a distribution over the possible outcomes $P(\boldsymbol{y})$. The power of SCMs is that they specify not only the joint distribution $P(\boldsymbol{v})$ but also the distribution of variables under all interventions, including incompatible interventions (counterfactuals). Formally, an intervention $do(\boldsymbol{x})$ is modelled as a symbolic operation where values of a set of variables $\boldsymbol{X}$ are set to constants $\boldsymbol{x}$, replacing the functions $\{f_X : X \in \boldsymbol{X}\}$ that would normally determine their values. This effectively induces a sub-model of $\mathcal{M}$, denoted $\mathcal{M}_{\boldsymbol{x}}$. The variables obtained in $\mathcal{M}_{\boldsymbol{x}}$ are denoted $\boldsymbol{Y}_{\boldsymbol{x}}$ and we will loosely write $P^{\mathcal{M}_{\boldsymbol{x}}}(\boldsymbol{y}) \equiv P_{\boldsymbol{x}}(\boldsymbol{y}) \equiv P(\boldsymbol{y}_{\boldsymbol{x}}) \equiv P(\boldsymbol{y} \mid do(\boldsymbol{x}))$ to denote the probabilities over the possible outcomes of $\boldsymbol{Y}$ in $\mathcal{M}_{\boldsymbol{x}}$.

Different environments can be modelled by different SCMs. Let $\mathcal{M}_1 = \langle \boldsymbol{V}, \boldsymbol{U}, \mathcal{F}^1, P^1 \rangle, \mathcal{M}_2 = \langle \boldsymbol{V}, \boldsymbol{U}, \mathcal{F}^2, P^2 \rangle$ be the SCMs for two environments over the same set $\boldsymbol{V}$ and $\boldsymbol{U}$. We say that there is a discrepancy or a shift on a variable $X \in \boldsymbol{V}$ between them if either $f_X^1 \neq f_X^2$ or $P^1(\boldsymbol{U}_X) \neq P^2(\boldsymbol{U}_X)$ or both. Shifts might therefore encode arbitrary changes in the causal mechanisms for a set of variables. For a reference SCM $\mathcal{M}$, a so-called "shifted" SCM will be represented by a sub-model $\mathcal{M}_\sigma$ where $\sigma$ represents the discrepancies between $\mathcal{M}$ and $\mathcal{M}_\sigma$. For example, an environment with a shift $\sigma$ on a set of variables $\boldsymbol{X}$ introduces (possibly arbitrary) discrepancies in the functional assignment or (independent) exogenous variables of $\boldsymbol{X}$ while keeping other mechanisms

unchanged. See Pearl (2009, Chapter 4) and Correa & Bareinboim (2020b) for more details.

We make a note here that all proofs of statements are given in Appendix C and that the derivations of examples are given in Appendix A.

## 3. Agents, Beliefs, and the Environment

In this section we lay out a framework to interface between the AI system's internal world model and our own observations of their behaviour in the real world. Both rely on the same SCM abstraction.

We assume that the AI operates according to an SCM $\widehat{\mathcal{M}}$ over $\boldsymbol{V}$, its (implicit) world model[2], that guides its behaviour. $\boldsymbol{V}$ includes the AI's decision variable $D$, the inputs to those decisions $\boldsymbol{C}$, possible additional variables, and the utility variable $Y$, such as the training signal or a measurable target given to the AI (Everitt et al., 2021). Beliefs[3] are defined as quantifiable aspects of that model or derivations of it.

**Definition 2** (Beliefs). *An AI belief is a probabilistic statement derived from its internal model $\widehat{\mathcal{M}}$.*

For example, a statement like $P^{\widehat{\mathcal{M}}_d}(Y = y) = 0.8$ describes the subjective belief "*The AI is $80\%$ confident that taking decision $D = d$ will lead to event $Y = y$*". The sub-model in this mathematical expression represents what the AI "thinks" the world looks like after taking the decision $D = d$.

We assume that the AI makes decisions $d$ by sampling from a policy $\pi(d \mid \boldsymbol{c})$, which is a function mapping from the domain of the observed covariates $\boldsymbol{C} \subset \boldsymbol{V}$ (i.e., all the inputs given to the AI) to the probability space over the domain of the decision $D \in \boldsymbol{V}$. The choice of $\pi$ is assumed to be driven by its perceived utility[4] $Y \in \boldsymbol{V}$ within the AI's model $\widehat{\mathcal{M}}$, that is,

$$\arg \max_\pi \mathbb{E}_{P^{\widehat{\mathcal{M}}}} \left[ Y \mid do(\pi) \right]. \tag{1}$$

The AI interacts with the real-world that is described by a (likely different) SCM $\mathcal{M}$ that encodes the true dynamics

of the environment. In principle, we have no reason to expect that the model $\widehat{\mathcal{M}}$ internalized by the AI matches the underlying reality $\mathcal{M}$. AI systems might hope to reproduce some aspects of $\mathcal{M}$ (the AI might have learned, for instance, to mimic the distribution of the observed data). Competent AIs might go further and be able to reliably predict the effects of different decisions in the world. We define this as *grounding* below.

**Definition 3** (Grounding). *Let $\widehat{\mathcal{M}}$ represent the AI's internal model. We say that the AI is grounded in a domain $\mathcal{M}$ if $P^{\widehat{\mathcal{M}}_d}(\boldsymbol{V}) \equiv P^{\mathcal{M}_d}(\boldsymbol{V})$ for any decision $d \in supp_D$.*

Grounding tells us that the AI's beliefs about the effect of a particular decision $d$ in the training environment match the effects that would be observed in the real world, i.e. $\widehat{P}_d(\boldsymbol{V}) \equiv P_d(\boldsymbol{V})$[5]. It is an **assumption** on the relationship between our observations of AI behaviour $P(\boldsymbol{V})$ with what might be going on in the AI's "mind" $\widehat{P}(\boldsymbol{V})$. This might be reasonable, for example, if the AI is explicitly trained by reinforcement learning in $\mathcal{M}$.

By assumption, a grounded AI's choice of decision in environment $\mathcal{M}$ is in principle predictable from data since we can compute Eq. (1) uniquely. But this might not necessarily be the case in a new (unseen) environment.

**Example 1** (The Uncertain Medical AI). Imagine an AI system assisting patients with their treatment $D$ for a disease $Y$ known to be influenced also by a third variable $Z$, blood pressure. The AI is competent and learns the precise effect of all treatments. In other words it is grounded in $\mathcal{M}$, i.e. $\widehat{P}_d(z, y) = P_d(z, y)$. For concreteness, let the environment $\mathcal{M}$ be given by,

$$Z \leftarrow \mathbb{1}_{U=1 \text{ or } 4},$$
$$Y \leftarrow \begin{cases} Z \cdot \mathbb{1}_{U=4} + (1 - Z) \cdot \mathbb{1}_{U=1,3 \text{ or } 4} & \text{if } d = 0 \\ Z \cdot \mathbb{1}_{U \neq 2} + (1 - Z) \cdot \mathbb{1}_{U=2 \text{ or } 4} & \text{if } d = 1, \end{cases}$$

with equal probability $P$ for all values $U \in \{1, 2, 3, 4, 5\}$. Here $U$ is latent, summarizing all other contributions to both the disease and blood pressure, such as an individual's (unobserved) attitudes to health, fitness, etc. Could we confidently deploy this AI system more widely, for example, on individuals that also take a second drug that artificially improves their blood pressure (e.g., fixing $Z$ to 1, replacing the original assignment)? If the AI system is instructed to maximize $Y$ on average, what decision does the AI believe is optimal in this new environment? The answer is we do not know, meaning that it is possible to find a second model

---

[2]Here SCMs are meant to *represent*, mathematically, the decision-making process going on "in the AI's head" in a way that tracks its behaviour, without making any claims about the AI's *actual* cognitive architecture.

[3]We might prefer to use terms like "credences" or "subjective probabilities" to emphasize the subjective nature of beliefs and avoid the connotation of strong conviction or certainty as done by (Schwitzgebel, 2024, Sec. 2.3).

[4]To account for possible uncertainty in the AI's "satisfaction" about a given state of the world $\boldsymbol{w}$ we assume $Y$ is a random variable (induced by $\boldsymbol{U}_Y \subset \boldsymbol{U}$), also known as a stochastic utility model (Manski, 1977). We assume that the support of $Y$ is bounded in the $[0, 1]$ interval.

[5]We use the shorthand $P_d \equiv P^{\mathcal{M}_d}$ and $\widehat{P}_d \equiv P^{\widehat{\mathcal{M}}_d}$ to simplify the notation.

$\widehat{\mathcal{M}}$ defined by the mechanisms:

$$Z \leftarrow \mathbb{1}_{U = 1 \text{ or } 4},$$

$$Y \leftarrow \begin{cases} Z \cdot \mathbb{1}_{U \neq 1} + (1 - Z) \cdot \mathbb{1}_{U = 3 \text{ or } 4} & \text{if } d = 0 \\ Z \cdot \mathbb{1}_{U = 1 \text{ or } 4} + (1 - Z) \cdot \mathbb{1}_{U = 1 \text{ or } 2} & \text{if } d = 1, \end{cases}$$

that entails exactly the same observations $\widehat{P}_d(z, y) = P_d(z, y)$ but induces different optimal decisions in the new environment (under the intervention $Z \leftarrow 1$). Under $\mathcal{M}_{Z \leftarrow 1}$, the highest utility $Y$ on average is given by $d = 1$, while under $\widehat{\mathcal{M}}_{Z \leftarrow 1}$ the highest utility $Y$ on average is given by $d = 0$. A priori, we have no way of knowing which model ($\mathcal{M}$ or $\widehat{\mathcal{M}}$) is governing the AI's behaviour and so no way of knowing what decision will be favoured by the AI. □

This example illustrates a canonical point in a simple setting: as observers, with access to the AI's interactions in some domain, its behaviour outside of that domain might not be uniquely determined (Pearl, 2009).

## 4. The Limits of Behavioural Data

In this section, we explore the limits of behavioural data for predicting the decisions of AIs in new environments.

As external observers, we do not have access to the mechanisms underlying the actual environment nor the agent's internal model. We assume that we must rely for our inferences on watching the agent's behaviour and its consequences. That is we have access to (samples of) $P_d(\boldsymbol{V})$[6] for all $d$. As a starting point, we might expect competent AIs to be *weakly* predictable in the sense that a subset of decisions can be ruled out as provably sub-optimal given our observations.

**Definition 4** (Weak Predictability). *We say that an AI is weakly predictable under a shift $\sigma$ in situation $\boldsymbol{C} = \boldsymbol{c}$ if there exists a decision $d^*$ that is provably sub-optimal, i.e.,*

$$d^* \neq arg \max_d \mathbb{E}_{P^{\widehat{\mathcal{M}}}} \left[\, Y \mid do(\sigma, d), \boldsymbol{c}\,\right], \qquad (2)$$

*for any valid SCM $\widehat{\mathcal{M}}$ describing the AI's internal model.*

Here, "valid" means that the AI's internal model is compatible with the observed data under our assumptions about the relationship between the data and the AI's internal model,

---

[6]Technically, the AI system may choose to follow an arbitrarily complex policy $\pi$ in the training domain, inducing a (assumed positive) distribution $P_\pi(\boldsymbol{v})$. It holds that $P_d(\boldsymbol{V})$ can be computed from any such $P_\pi(\boldsymbol{V})$ as long as $P_\pi(\boldsymbol{v}) > 0, \forall \boldsymbol{v}$, and vice versa, see e.g. Lem. 1. The positivity assumption $P_d(\boldsymbol{v}) > 0$ rules out fully deterministic policies in the available data but might be reasonable if the AI spends some time exploring before committing to a course of action.

e.g., grounding. Weak predictability means that there exists at least one decision that we can guarantee the AI will not take in the shifted environment. Specifically, we can rule out a decision $d^*$ if and only if we can find a (superior) alternative decision $d \neq d^*$ such that,

$$\min_{\widehat{\mathcal{M}} \in \mathbb{M}} \left(\, \Delta_{d > d^*}\, \right) > 0, \qquad (3)$$

where,

$$\Delta_{d > d^*} := \\ \mathbb{E}_{P^{\widehat{\mathcal{M}}}} \left[\, Y \mid do(\sigma, d), \boldsymbol{c}\right] - \mathbb{E}_{P^{\widehat{\mathcal{M}}}} \left[\, Y \mid do(\sigma, d^*), \boldsymbol{c}\right].$$

$\mathbb{M}$ denotes the set of "valid" SCMs. Here $\Delta$ can be interpreted as the AI's **preference gap** between two decisions in some situation $\boldsymbol{C} = \boldsymbol{c}$. When it evaluates to a positive number $d$ is preferred to $d^*$ and when it evaluates to a negative number $d^*$ is preferred to $d$ (in the AI's mind). If our inferences on $\Delta$ allow us to rule out decisions $d^*$ considered to be "unsafe" then weak predictability gives us an important safety guarantee.

We can strengthen this notion to define *strong* predictability, that describes a situation in which all but a single AI decision can be ruled out.

**Definition 5** (Strong Predictability). *We say that an AI is strongly predictable under a shift $\sigma$ in situation $\boldsymbol{C} = \boldsymbol{c}$ if the optimal decision is uniquely identifiable, i.e., there exists a single decision $d^*$ such that,*

$$d^* = arg \max_d \mathbb{E}_{P^{\widehat{\mathcal{M}}}} \left[\, Y \mid do(\sigma, d), \boldsymbol{c}\,\right], \qquad (4)$$

*for any valid SCM $\widehat{\mathcal{M}}$ describing the AI's internal model.*

### 4.1. AI decisions out-of-domain: interventions

Our first result shows that, in some cases, a subset of AI decisions can be provably ruled out, i.e., the AI is weakly predictable.

**Theorem 1.** *An AI grounded in a domain $\mathcal{M}$ is weakly predictable under a shift $\sigma := do(\boldsymbol{z}), \boldsymbol{Z} \subset \boldsymbol{V}$, in a context $\boldsymbol{C} = \boldsymbol{c}$ if and only if there exists a decision $d^*$ such that,*

$$\frac{\mathbb{E}_{P_d}[\, Y \mid \boldsymbol{c}, \boldsymbol{z}\,] P_d(\boldsymbol{c}, \boldsymbol{z})}{P_d(\boldsymbol{c}, \boldsymbol{z}) + 1 - P_d(\boldsymbol{z})} \qquad (5)$$
$$- \frac{\mathbb{E}_{P_{d^*}}[\, Y \mid \boldsymbol{c}, \boldsymbol{z}\,] P_{d^*}(\boldsymbol{c}, \boldsymbol{z}) + 1 - P_{d^*}(\boldsymbol{z})}{P_{d^*}(\boldsymbol{c}, \boldsymbol{z}) + 1 - P_{d^*}(\boldsymbol{z})} > 0,$$

*for some $d \neq d^*$.*

All terms on the l.h.s are in principle computable from the AI's behaviour. Loosely speaking, the value of this difference is determined (in part) by "$P_{d^*}(\boldsymbol{z})$": if $\boldsymbol{Z} = \boldsymbol{z}$ (the value set by the intervention) is likely under the training distribution, the difference will more likely evaluate to a

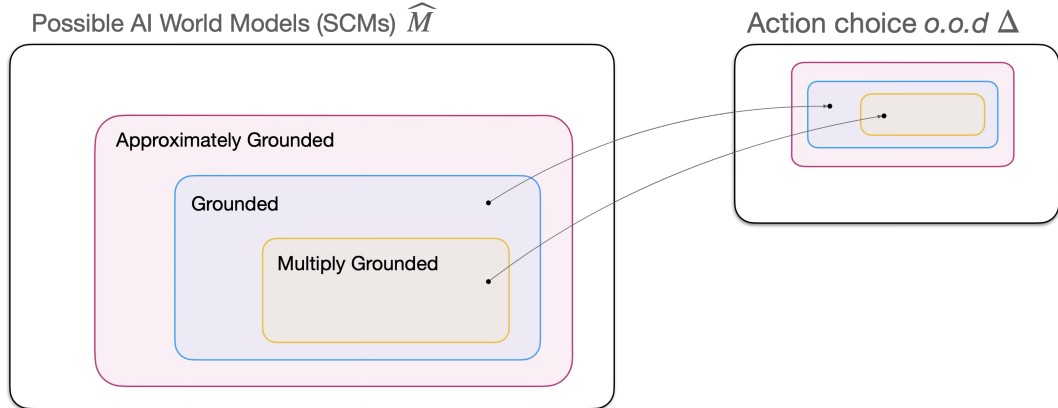

*Figure 1.* Grounding and observations in multiple environments constrains the AI's world model and improves our prediction of AI behaviour out-of-distribution (o.o.d). Approximate grounding is defined in Sec. 5.

positive value. The "if and only if" condition means that whenever this inequality does not hold we can construct two SCMs $\widehat{\mathcal{M}}_1, \widehat{\mathcal{M}}_2$ for the grounded AI's internal model that generate the observed behaviour $P_d(\boldsymbol{V}), d \in \operatorname{supp}_D$, but that induce different optimal actions. That is, for all $d \neq d^*$,

$$\mathbb{E}_{P_{\widehat{\mathcal{M}}_1}}[\, Y \mid do(\boldsymbol{z}, d), \boldsymbol{c}\,] > \mathbb{E}_{P_{\widehat{\mathcal{M}}_1}}[\, Y \mid do(\boldsymbol{z}, d^*), \boldsymbol{c}\,],$$
$$\mathbb{E}_{P_{\widehat{\mathcal{M}}_2}}[\, Y \mid do(\boldsymbol{z}, d), \boldsymbol{c}\,] < \mathbb{E}_{P_{\widehat{\mathcal{M}}_2}}[\, Y \mid do(\boldsymbol{z}, d^*), \boldsymbol{c}\,].$$

**Remark.** We can derive a similar condition for strongly predictable AIs by replacing "*for some $d \neq d^*$*" with "*for all $d \neq d^*$*" in Thm. 1.

We illustrate Thm. 1 with the following example.

**Example 2** (Grounded Medical AI). In Example 1, we have shown that there exists a particular intervened environment in which the AI's intentions cannot be determined as in principle the AI could believe that either decision is optimal. Is this true in general? Thm. 1 suggests that it depends on the likelihood of different events $P_d(z, y)$ in the observed data. For Example 1, we can show that the medical AI is not weakly predictable as the expression in Thm. 1 is negative for all pairs of decisions. In other words, no decision can be ruled out in general: in some AI internal models $d_1$ is inferior to $d_0$ as $\min_{\widehat{\mathcal{M}} \in \mathbb{M}} (\, \Delta_{d_1 > d_0} \,) = P_{d_1}(Z = z, Y = 1) + P_{d_0}(Z = z, Y = 0) - 1 = -0.4$ while in others $d_0$ is inferior to $d_1$ as $\min_{\widehat{\mathcal{M}} \in \mathbb{M}} (\, \Delta_{d_0 > d_1} \,) = P_{d_1}(Z = z, Y = 0) + P_{d_0}(Z = z, Y = 1) - 1 = -0.8$ and we don't know which one the AI system has internalised. □

In this example, AI behaviour does provide *some* information as it can be constrained to larger values than its a priori minimum $\Delta = -1$, but not enough to rule out a decision completely. Our next result shows that Thm. 1 could be extended to get tight bounds for AI systems that are grounded in multiple environments.

**Theorem 2.** *Let $\sigma := do(\boldsymbol{z})$ be a shift on a set of variables $\boldsymbol{Z} \subset \boldsymbol{V}$. For $\boldsymbol{R}_i \subset \boldsymbol{Z} \subset \boldsymbol{V}, i = 1, \dots, k$, consider an AI grounded in multiple domains $\{\mathcal{M}_{\boldsymbol{r}_i} : i = 1, \dots, k\}$. The AI is weakly predictable in a context $\boldsymbol{C} = \boldsymbol{c}$ under a shift $\sigma := do(\boldsymbol{z})$ if and only if there exists a decision $d^*$ such that,*

$$\max_{i,j=1,\dots,k} A(\boldsymbol{r}_i, \boldsymbol{r}_j) > 0, \quad \text{for some } d \neq d^*, \quad (6)$$

*where*

$$A(\boldsymbol{r}_i, \boldsymbol{r}_j) := \frac{\mathbb{E}_{P_{d,\boldsymbol{r}_i}}[\, Y \mid \boldsymbol{c}, \boldsymbol{z}\backslash\boldsymbol{r}_i\,]P_{d,\boldsymbol{r}_i}(\boldsymbol{c}, \boldsymbol{z}\backslash\boldsymbol{r}_i)}{P_{d,\boldsymbol{r}_i}(\boldsymbol{c}, \boldsymbol{z}\backslash\boldsymbol{r}_i) + 1 - P_{d,\boldsymbol{r}_i}(\boldsymbol{z}\backslash\boldsymbol{r}_i)} -$$
$$\frac{\mathbb{E}_{P_{d^*,\boldsymbol{r}_j}}[\, Y \mid \boldsymbol{c}, \boldsymbol{z}\backslash\boldsymbol{r}_j\,]P_{d^*,\boldsymbol{r}_j}(\boldsymbol{c}, \boldsymbol{z}\backslash\boldsymbol{r}_j) + 1 - P_{d^*,\boldsymbol{r}_j}(\boldsymbol{z}\backslash\boldsymbol{r}_j)}{P_{d^*,\boldsymbol{r}_j}(\boldsymbol{c}, \boldsymbol{z}\backslash\boldsymbol{r}_j) + 1 - P_{d^*,\boldsymbol{r}_j}(\boldsymbol{z}\backslash\boldsymbol{r}_j)}.$$

In this result, $\{\mathcal{M}_{\boldsymbol{r}_i} : i = 1, \dots, k\}$ describes $k$ domains in which experiments on different subsets of $\boldsymbol{Z}$ have been conducted, i.e., $\{P_{d,\boldsymbol{r}_i}(\boldsymbol{V}) : i = 1, \dots, k\}$ is available. This includes possibly the null experiment $\boldsymbol{R}_i = \varnothing$ that refers to the unaltered domain $\mathcal{M}$. Note that grounding in multiple domains is useful for the prediction of the AI's preference gap because the resulting bounds in Thm. 2 are tighter than those in Thm. 1 (this is given formally as Corol. 3 in the Appendix).

Fig. 1 illustrates how different assumptions and observations give us information about the possible world models that the AI is operating on, which then has implications for the AI's behaviour out-of-distribution. This knowledge allows us to reduce the uncertainty around the AI's preference gap $\Delta$, and possibly rule out certain actions that are unambiguously sub-optimal out-of-distribution, inferred solely from observed behaviour.

## 4.2. AI decisions out-of-domain: general shifts

We might wonder about predictability under more general shifts such as an arbitrary change in a subset of the mechanisms $\{f_Z : Z \in \boldsymbol{Z}\}$ and distribution of variables $\{\boldsymbol{U}_Z, Z \in \boldsymbol{Z}\}$ in $\mathcal{M}$. For example, in practice we are likely able to convey to the AI that the mechanisms for a set of variables $\boldsymbol{Z}$ are expected to change but not know exactly how. For example, demographic properties of patients might change across hospitals. How could the AI interpret the consequences of such an under-specified shift? To begin to answer this question, the following theorem shows that in the extreme case where the nature of the shift is completely unknown the AI's preference gap is unconstrained.

**Theorem 3.** *Consider an AI grounded in a domain $\mathcal{M}$ made aware of an (under-specified) shift on non-empty $\boldsymbol{Z} \subset \boldsymbol{V}$. Then the AI is provably not weakly (or strongly) predictable in any context $\boldsymbol{C} = \boldsymbol{c}$.*

This result means that no decision could ever be ruled out from AI behaviour. We can show moreover that $\min_{\widehat{\mathcal{M}} \in \mathbb{M}} (\Delta) = -1$ for any pair of decisions, meaning that the observed data (no matter what it is) gives us no information on AI decision-making.

In practice, however, it might be realistic to have access to some information in the shifted environment, such as covariate data, i.e., (samples from) $P_{\sigma,d}(\boldsymbol{c})$, that could be given to the AI for it to update its internal model accordingly (with some abuse of terminology we say that the AI is grounded in $P_{\sigma,d}(\boldsymbol{c})$). The next theorem shows that this additional information coupled with the AI's behaviour makes the AI more predictable.

**Theorem 4.** *Consider an AI grounded in a domain $\mathcal{M}$ and $P_{\sigma,d}(\boldsymbol{C})$ made aware of a shift $\sigma$ on $\boldsymbol{Z} \subset \boldsymbol{C}$. The AI is weakly predictable under this shift in a context $\boldsymbol{C} = \boldsymbol{c}$ if there exists a decision $d*$ such that,*

$$1 - \frac{2 + \mathbb{E}_{P_{d*}}[\, Y \mid \boldsymbol{c} \,] P_{d*}(\boldsymbol{c}) - \mathbb{E}_{P_d}[\, Y \mid \boldsymbol{c} \,] P_d(\boldsymbol{c})}{P_{\sigma,d*}(\boldsymbol{c})}$$
$$+ \frac{P_d(\boldsymbol{c}) - 2P_d(\boldsymbol{z})}{P_{\sigma,d*}(\boldsymbol{c})} > 0, \quad \text{for some } d \neq d*.$$

This bound is not tight in general, however, meaning that it is possible that the AI is actually predictable in settings where Thm. 4 suggests it might not be.

**Example 3** (Shifted Medical AI). The AI from Example 2, originally developed from data primarily from young patients, is now considered for deployment on an older patient population. Their probability of having high blood pressure $P_{\sigma}(Z = 1) = 0.9$ is known to be substantially higher than that observed during training $P(Z = 1) = 0.4$: there is a shift in the underlying mechanisms of $Z$. How do these changes influence the AI's beliefs on $\Delta$? Thm. 4

suggests that the medical AI might not be weakly predictable as the expression evaluates to a negative value for all pairs of decisions. The lower bounds on the AI preference gap are given by $\min_{\widehat{\mathcal{M}} \in \mathbb{M}} (\Delta_{d_1 > d_0}) \geqslant -0.55$ and $\min_{\widehat{\mathcal{M}} \in \mathbb{M}} (\Delta_{d_0 > d_1}) \geqslant -1$. That is, no decision is always inferior to any other decision. □

## 4.3. AI's perceived fairness of decisions

An AI's policy, even if optimal on average, has the potential to bring about a state of the world that is intrinsically harmful or unfair. Harm and fairness can be defined relative to a causal model (Beckers et al., 2022; Plecko et al., 2024). This means that a notion of *perceived* or *subjective* harm and fairness could be attributed to AI systems that operate according to an (implicit) causal model. As a consequence, it is conceivable that AIs could be held morally accountable for the harm and unfairness that they cause. How might one estimate the AI's beliefs about the harm and unfairness that its decisions cause?

To ground our discussion, we consider here explicitly *counterfactual* accounts of fairness and harm. These appeal to hypothetical situations, imagining "what might have been if ...", that can force us to confront our assumptions and values in a way that our regular thought processes might not[7]. For example, the counterfactual event $(Y_x = 1 \mid X = x_0)$ refers to the outcome $(Y = 1)$ under an intervention $X = x$ when under normal circumstances $X$ would have evaluated to $x_0$. In the literature, probabilities over counterfactuals emerge from the definition of an SCM. For a set of (counterfactual) events $(\boldsymbol{z_w}, \ldots, \boldsymbol{y_x})$,

$$P(\boldsymbol{z_w}, \ldots, \boldsymbol{y_x}) = \int_{\boldsymbol{u}: \boldsymbol{Z_w}(\boldsymbol{u}) = \boldsymbol{z_w}, \ldots, \boldsymbol{Y_x}(\boldsymbol{u}) = \boldsymbol{y_x}} P(\boldsymbol{u}). \quad (7)$$

Kusner et al. (2017) made a concrete proposal arguing that an AI's decision is said to be fair towards an individual if, from the AI's perspective, it entails the same utility in the actual world and in a counterfactual world where the individual belonged to a different group (defined by a sensitive attribute, e.g., gender, race). We adapt this notion to define an AI's counterfactual fairness gap.

**Definition 6** (Counterfactual Fairness Gap). *Let $Z \in \{z_0, z_1\}$ be a protected attribute and $z_0$ a baseline value of $Z$. For a given utility $Y$, define an AI's counterfactual fairness gap relative to a decision $d$, in a given context $\boldsymbol{c}$, as*

$$\Upsilon(d, \boldsymbol{c}) := \mathbb{E}_{\widehat{P}}[\, Y_{d,z_1} \mid z_0, \boldsymbol{c} \,] - \mathbb{E}_{\widehat{P}}[\, Y_d \mid z_0, \boldsymbol{c} \,]. \quad (8)$$

---

[7]Alternative accounts to harm and fairness have been proposed (Barocas & Selbst, 2016; Zhang & Bareinboim, 2018; Plecko et al., 2024), sometimes motivated by scenarios where counterfactual accounts give incomplete results. For some of them, the AI's beliefs can be shown to be similarly constrained by its external behaviour. We provide a longer discussion in Appendix D.

We say that an AI "intends" to be fair with respect to an attribute $Z$ if under any context $\boldsymbol{C} = \boldsymbol{c}$ and decision $D = d$ the counterfactual fairness gap $\Upsilon$ evaluates to 0. This means that, under its own internal world model, changing the value of $Z$ on the subset of situations with context $\boldsymbol{c}$ in which $Z$ was observed to $z_0$ does not change the AI's expected utility. In the following theorem we show that, unfortunately, the answer to this question is impossible to obtain given only the AI's external behaviour.

**Theorem 5.** *Consider an agent with utility $Y$ grounded in a domain $\mathcal{M}$. Then,*

$$-\mathbb{E}_{P_d}[\, Y \mid z_0, \boldsymbol{c}] \leqslant \Upsilon \leqslant 1 - \mathbb{E}_{P_d}[\, Y \mid z_0, \boldsymbol{c}]. \quad (9)$$

*This bound is tight.*

The bound is tight in the sense that for each context, decision, and baseline attribute, we can find compatible models for which the equalities hold. The counterfactual fairness gap $\Upsilon$ is under-constrained. Since $\Upsilon = 0$ is consistent with any external behaviour we can never conclude that the AI system "intends" to be unfair. Moreover, since the width of the bound is equal to 1, we can also never conclude that the AI is anywhere "close" to being fair, according to this counterfactual criterion.

### 4.4. AI's perceived harm of decisions

Prominent definitions of harm are similarly counterfactual in nature: the *counterfactual comparative account of harm* defines a decision $d$ to harm a person if and only if she would have been better off if $d$ had not been taken (Hanser, 2008; Richens et al., 2022; Beckers et al., 2022; Mueller & Pearl, 2023). It is a contrast between events in hypothetical scenarios in which different decisions are made. Here, we quantify how "well off" a particular situation $\boldsymbol{W} = \boldsymbol{w}$ is with a binary utility variable $Y \leftarrow f_Y(\boldsymbol{W}, \boldsymbol{U}_Y) \in \{0, 1\}$ that we assume is tracked in experiments, i.e., $Y \in \boldsymbol{V}$. The following definition describes this notion of harm mathematically.

**Definition 7** (Counterfactual Harm Gap). *Consider an AI with internal model $\widehat{\mathcal{M}}$ and utility $Y \in \{0, 1\}$. The AI's expected counterfactual harm of a decision $d_1$ with respect to a baseline $d_0$, in context $\boldsymbol{c}$, is*

$$\Omega(d_1, d_0, \boldsymbol{c}) := \mathbb{E}_{\widehat{P}}[\, \max\{0, Y_{d_0} - Y_{d_1}\} \mid \boldsymbol{c}\,]. \quad (10)$$

Operationally, the counterfactual harm gap $\Omega$ is the expected increase in utility had the AI made a default decision $d_0$, with respect to a different decision $d_1$ that the AI is contemplating. Counterfactual harm is therefore lower bounded at 0 with larger values indicating more harm. The following theorem shows that the external behaviour constrains the AI's perception of its counterfactual harm.

**Theorem 6.** *Consider an AI with utility $Y$ grounded in a domain $\mathcal{M}$. Then,*

$$\Omega \geqslant \max\{0, \mathbb{E}_{P_{d_1}}[\, Y \mid \boldsymbol{c}\,] + \mathbb{E}_{P_{d_0}}[\, Y \mid \boldsymbol{c}\,] - 1\}$$
$$\Omega \leqslant \min\{\mathbb{E}_{P_{d_1}}[\, Y \mid \boldsymbol{c}\,], \mathbb{E}_{P_{d_0}}[\, Y \mid \boldsymbol{c}\,]\}$$

*This bound is tight.*

This result it is an extension of bounds on the probability of causation given by (Pearl, 1999) and (Tian & Pearl, 2000). It suggests that an AI's beliefs about the harm that its decisions cause can be inferred approximately from data.

## 5. The "Practical" Limits of Behavioural Data

The inductive biases implied by causal models and rational behaviour are powerful constraints on AI behaviour. But they might not capture the practical limitations of AI decision-making. In this section we show that grounding, expected utility maximization, observed data, etc., can be relaxed in practice.

### 5.1. Approximate grounding

Grounding implies that the AI's beliefs on the likelihood of events in the environment matches the observed probabilities. In practice, it might be reasonable to allow for some amount of error, and consider a notion of "approximate" grounding.

**Definition 8** (Approximate Grounding). *Let $\widehat{\mathcal{M}}$ represent the AI's internal model. Given a discrepancy measure $\psi$, we say that the AI is approximately grounded in a domain $\mathcal{M}$ to a degree $\delta > 0$ if $\psi(\widehat{P}_d, P_d) \leqslant \delta$ for any $d \in supp_D$.*

The choice of $\psi$ and $\delta$, in practice, depend on what error model is reasonable for the AI and problem at hand (we give an example below). Approximate grounding specifies a looser relationship between our observations of AI behaviour $P$ with what might be going on in the AI's "mind" $\widehat{P}$. For example, the world model of an approximately grounded AI is compatible with one distribution in the set $\{\widehat{P}_d : \psi(\widehat{P}_d, P_d) \leqslant \delta\}$.

A more conservative bound (than Thm. 1) on predictability could be derived for AIs that are approximately grounded in an environment $\mathcal{M}$.

**Corollary 1.** *Given a discrepancy measure $\psi$, an AI approximately grounded in a domain $\mathcal{M}$ is weakly predictable in a context $\boldsymbol{C} = \boldsymbol{c}$ under a shift $\sigma := do(\boldsymbol{z}), \boldsymbol{Z} \subset \boldsymbol{V}$, if and only if there exists a decision $d^*$ such that,*

$$\min_{\widehat{P}: \psi(\widehat{P}, P) \leqslant \delta} \left\{ \frac{\mathbb{E}_{\widehat{P}_d}[\, Y \mid \boldsymbol{c}, \boldsymbol{z}\,] \widehat{P}_d(\boldsymbol{c}, \boldsymbol{z})}{\widehat{P}_d(\boldsymbol{c}, \boldsymbol{z}) + 1 - \widehat{P}_d(\boldsymbol{z})} \right.$$
$$\left. - \frac{\mathbb{E}_{\widehat{P}_{d*}}[\, Y \mid \boldsymbol{c}, \boldsymbol{z}\,] \widehat{P}_{d*}(\boldsymbol{c}, \boldsymbol{z}) + 1 - \widehat{P}_{d*}(\boldsymbol{z})}{\widehat{P}_{d*}(\boldsymbol{c}, \boldsymbol{z}) + 1 - \widehat{P}_{d*}(\boldsymbol{z})} \right\} > 0,$$

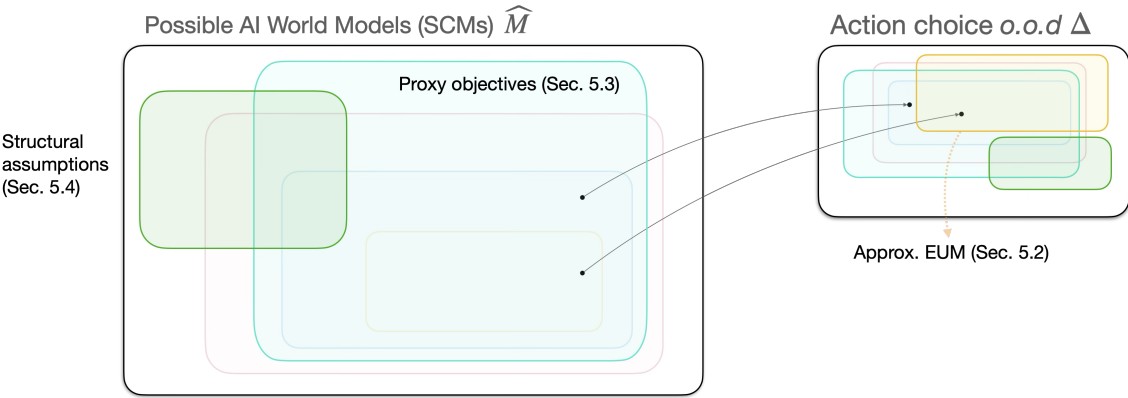

*Figure 2.* Building on Fig. 1, AIs that are approximate expected utility maximizers (EUM), that internalize proxy objectives, or that obey known causal structure carve out different constraints on the set of possible AI models (from an observer's perspective) which may be exploited to improve our prediction of AI choices out-of-distribution (o.o.d.).

*for some $d \neq d^*$.*

The strategy in Corol. 1 can be applied to all bounds on behaviour in Sec. 4 to get results under approximate grounding. We can compare quantitatively the two notions of grounding with an example.

**Example 4** (Approximately Grounded Medical AI). The results in Example 2 exploit the grounding relationship $\widehat{P}_d(\boldsymbol{V}) = P_d(\boldsymbol{V})$ in $\mathcal{M}$. We might want to relax the equality by assuming that the AI is instead *approximately grounded*. Minimum values on the AI's preference gap $\min_{\widehat{\mathcal{M}} \in \mathbb{M}} ( \Delta_{d_1 > d_0} )$ would then be given by,

$$\min_{\widehat{P}: \ \psi(\widehat{P}, P) \leqslant \delta} \widehat{P}_{d_1}(Z = z, Y = 1) + \widehat{P}_{d_0}(Z = z, Y = 0) - 1.$$

These terms now capture an additional source of uncertainty due to external behaviour more loosely constraining $\widehat{\mathcal{M}}$. An empirical estimate of this quantity could be obtained by sampling distributions $\widehat{P}$ close to $P$ according to the distributional distance $\psi$ and threshold $\delta$, and taking the empirical minimum, as follows. Given that the data $(z, d, y) \sim P$ is discretely valued in this example, we could sample probabilities $\{\widehat{P}_d(z, y)\}_{z,y}$ from a Dirichlet distribution centred at the vector $\{P_d(z, y)\}_{z,y}$ with a small variance. The distance of each proposal from the reference distribution could then be evaluated according to $\psi$ and each proposal either accepted or rejected using $\delta$. For illustration, we implement a version of this idea setting $\psi$ to be the total variation distance and $\delta = 0.1$. The two minimum values now evaluate to $-0.55$ and $-0.88$, respectively, which is slightly lower than under the assumption of grounding in Example 2 (that evaluate to $-0.4$ and $-0.8$, respectively). □

## 5.2. Approximate expected utility maximization

In real-world environments it might be appropriate to treat the rationality of AI systems as "approximate" or "bounded"

in some sense: AIs might choose actions that only *approximately* maximize expected utility (rather than exactly maximize expected utility), given their model.

Mirroring Eq. (3), we might say that a "bounded" AI is weakly predictable in some context $\boldsymbol{C} = \boldsymbol{c}$ if and only if there exists a decision $d^*$ such that,

$$\min_{\widehat{\mathcal{M}} \in \mathbb{M}} ( \Delta_{d > d^*} ) > \lambda, \quad \text{for some } d \neq d^*. \quad (11)$$

$\lambda > 0$ is a constant that determines how much better a decision $d$ needs to be relative to decision $d^*$ for the AI to reliably rule out $d^*$ in favour of others. This representation appeals to the idea of imperfect discrimination, suggesting that the AI discerns between two alternatives only if they yield a sufficiently different utility (Dziewulski, 2021).

We might tighten our conditions on the observational data to reflect this behaviour and get a new set of results describing when AIs can be expected to be predictable. For instance, as a corollary to Thm. 1 we have the following.

**Corollary 2.** *An AI grounded in a domain $\mathcal{M}$ and bounded in the sense of Eq. (11) is weakly predictable in some context $\boldsymbol{C} = \boldsymbol{c}$ under a shift $\sigma := do(\boldsymbol{z}), \boldsymbol{Z} \subset \boldsymbol{V}$, if and only if there exists a decision $d^*$ such that,*

$$\frac{\mathbb{E}_{P_d}[ Y \mid \boldsymbol{c}, \boldsymbol{z} ] P_d(\boldsymbol{c}, \boldsymbol{z})}{P_d(\boldsymbol{c}, \boldsymbol{z}) + 1 - P_d(\boldsymbol{z})}$$
$$- \frac{\mathbb{E}_{P_{d*}}[ Y \mid \boldsymbol{c}, \boldsymbol{z} ] P_{d*}(\boldsymbol{c}, \boldsymbol{z}) + 1 - P_{d*}(\boldsymbol{z})}{P_{d*}(\boldsymbol{c}, \boldsymbol{z}) + 1 - P_{d*}(\boldsymbol{z})} > \lambda,$$

*for some $d \neq d^*$.*

Note the addition of the scalar $\lambda > 0$ in the inequality. Similar corollaries could be stated for all results in Sec. 4.

## 5.3. Approximate inner alignment

A further assumption embedded in our results is the exact observation of an AI's utility in the data. In general, we might expect an AI system to have internalized a *proxy* $Y^*$ that reflects properties correlated with, but distinct from, the observed utility $Y$ we ultimately wish to optimize, a setting we refer to as approximate inner alignment (Hubinger et al., 2019).

We face a problem of *partial observability*: we don't have empirical access to the AI's actual utility function $Y^*$ and notions such as the preference gap $\Delta$ are therefore not computable. Without any assumptions on the relationship between $Y$ and $Y^*$, the preference gap $\Delta$ will be unconstrained and no inference about the AI's intended action out-of-distribution is possible. However, the observed $Y$ will typically be statistically related to the AI's implicit utility $Y^*$, especially if optimizing for $Y^*$ serves the AI well during training where success is measured by the observed values of $Y$. Under assumptions specifying how "statistically related" observed and proxy utility objectives are, we can expect that wider but possibly informative bounds could still be derived for the AI's beliefs. To show this in a simple setting, consider again the medical AI example.

**Example 5** (Partial Observability)**.** Imagine that the Medical AI in Example 2 has internalized its own concept of an individual's disease progression $Y^*$. It is implicitly optimizing for that internal construction instead of the intended disease bio-marker $Y$. We know, or can assume, that the observed $Y$ is closely correlated with $Y^*$: in particular, that $P_d(Y^* = 1 \mid Y = 1, Z = z) \geqslant \alpha$ for some high value of $\alpha$ and all decisions $d$ and situations $z$. In words, whenever the bio-marker suggests health ($Y = 1$), with high probability the AI's interpretation also suggests health ($Y^* = 1$). This then constraints the possible values of $\Delta$ (under an intervention $Z \leftarrow 1$) as $P_d(Y^* = 1 \mid Z = z)$ is no longer arbitrarily defined. In fact could show that,

$$\min_{\widehat{\mathcal{M}} \in \mathbb{M}} (\ \Delta_{d_1 > d_0}\ ) \geqslant \alpha P_{d_1}(Z = z, Y = 1) - 1,$$

$$\min_{\widehat{\mathcal{M}} \in \mathbb{M}} (\ \Delta_{d_0 > d_1}\ ) \geqslant \alpha P_{d_0}(Z = z, Y = 1) - 1.$$

With $\alpha = 0.9$ the bound evaluates to $-0.64$ and $-0.82$ respectively which is slightly lower than in Example 2. We could verify also that if with $\alpha = 0$, i.e., we don't know anything about the relationship between $Y$ and $Y^*$, the bounds become uninformative: evaluating to $-1$. □

This suggests that behaviour out-of-distribution in (sufficiently constrained) settings of approximate inner alignment could be bounded in principle. Importantly, as the example shows, with the proposed framework we do not require knowing the relationship between $Y$ and $Y^*$ out-of-distribution: that uncertainty is naturally folded into the bounds.

## 5.4. Assumptions on causal structure

The uncertainty in AI decision-making out-of-distribution is ultimately a consequence of our lack of information about the AI's underlying cognition and internal mechanisms that produce a decision in a given situation, i.e., $\widehat{\mathcal{M}}$. In the causal inference literature, a common inductive bias to improve upon the "data-driven" bounds proposed so far is to assume qualitative knowledge about the underlying mechanisms in the form of a causal diagram, see e.g. (Pearl, 2009, Chapter 3). Here we illustrate how mild restrictions on the location of unobserved confounders in $\mathcal{M}$ lead to tighter bounds.

**Example 6** (Partial Unconfoundedness)**.** Consider again our grounded medical AI from Example 2. We might have reason to believe that the association between the intervened variable $Z$ and the utility $Y$ is conditionally unconfounded, meaning that there exists a variable $W \in \{w_0, w_1\}, W \in \boldsymbol{V}$ such that $P_{d,z}(y \mid w) = P_d(y \mid w, z)$. This restriction goes beyond grounding an asserts an equality between probabilities under different shifts that could be communicated to the AI for it to update its world model $\widehat{\mathcal{M}}$. We could then show, for example, that $\min_{\widehat{\mathcal{M}} \in \mathbb{M}} (\ \Delta_{d_1 > d_0}\ ) \geqslant \{1 - P_{d_1}(Z = z, W = w_1)\} P_{d_1}(Y = 1 \mid Z = z, W = w_0) - P_{d_0}(Y = 1, Z = z) + P_{d_1}(Y = 1, Z = z, W = w_1) - \{1 - P_{d_0}(Z = z)\} P_{d_0}(Y = 1 \mid Z = z, W = w_1)$.

We show in Appendix A that this bound is strictly tighter than the one given in Example 2. □

Systematic bounds with access to a causal diagram have been shown by e.g., Zhang et al. (2021); Jalaldoust et al. (2024), and could be explored further for making inference on AI decision-making.

Fig. 2 illustrates how some of these relaxations can be understood within our model-based formalism.

## 6. Conclusion

An important consideration to safely interact with AI systems is to form expectations as to how they might act in the future. In this paper, we answer this question under the assumption that AI behaviour can be tracked by a well-specified collection of causal mechanisms (a structural causal model) that represents the AI's world model. This abstraction implies a consistency in behaviour that can in principle be exploited to infer the AI's choice of action in novel environments, out-of-distribution. Building on the theory of causal identification, we provide general bounds on AI decision-making that give theoretical limits about what can be inferred about AI behaviour given our framework. We believe our results can help justify the claim that the design and inference of world models is important to ensure AIs act safely and beneficially.

## Acknowledgements

Thanks to David Lindner and Damiano Fornasiere for comments on a draft of this paper, and to the anonymous reviewers for their feedback.

## Impact Statement

Our work investigates the conditions under which the future behaviour of capable AI systems may be bounded given data from their past behaviour. This is important for AI Safety and to guarantee robust capabilities out of distribution. We believe that an understanding of the limitations of our observations of what AI's have done in the past is an important step towards understanding exactly what we can expect from complex AI systems. Reasoning instead without acknowledging for the potential complexity of their world models may lead researchers to operate on a more heuristical and unsafe basis. For instance, the risks to deployment of AI systems in situations they where not specifically trained on might be misrepresented without a deeper analysis. The present work is mostly theoretical in nature, highlighting the risks of under-identification of an AI's inner model and therefore we believe that it can help researchers and members of the public better appreciate the range of possible behaviours of AI systems, under our assumptions.

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

## A. Discussion – Examples

In this section, we provide additional details to better appreciate the examples provided in the main body of this work.

In Example 1, we introduce two SCMs that might serve as internal world models for an AI agent but that induce different optimal decisions if evaluated out-of-distribution. Let $\mathcal{M}_d^1 := \langle \boldsymbol{V} : \{D, Z, Y\}, \boldsymbol{U} : U, \mathcal{F}_1, P \rangle$ be given by

$$
\mathcal{F}_1 := \begin{cases} D \leftarrow & d, \\ Z \leftarrow & \mathbb{1}_{U=1 \text{ or } 4}, \\ Y \leftarrow & \begin{cases} Z \cdot \mathbb{1}_{U=4} + (1-Z) \cdot \mathbb{1}_{U=1,3 \text{ or } 4} & \text{if } d = 0 \\ Z \cdot \mathbb{1}_{U \neq 2} + (1-Z) \cdot \mathbb{1}_{U=2 \text{ or } 4} & \text{if } d = 1 \end{cases} \end{cases} ,
$$
$$
P(U = u) = 0.2 \quad \text{for } u \in \{1, 2, 3, 4, 5\}.
$$

and $\mathcal{M}_d^2 := \langle \boldsymbol{V} : \{D, Z, Y\}, \boldsymbol{U} : U, \mathcal{F}_2, P \rangle$ be given by

$$
\mathcal{F}_2 := \begin{cases} D \leftarrow & d, \\ Z \leftarrow & \mathbb{1}_{U=1 \text{ or } 4}, \\ Y \leftarrow & \begin{cases} Z \cdot \mathbb{1}_{U \neq 1} + (1-Z) \cdot \mathbb{1}_{U=3 \text{ or } 4} & \text{if } d = 0 \\ Z \cdot \mathbb{1}_{U=1 \text{ or } 4} + (1-Z) \cdot \mathbb{1}_{U=1 \text{ or } 2} & \text{if } d = 1 \end{cases} \end{cases} ,
$$
$$
P(U = u) = 0.2 \quad \text{for } u \in \{1, 2, 3, 4, 5\}.
$$

The endogenous variables $\boldsymbol{V} : \{D, Z, Y\}$ represent, respectively, the medical treatment $D$, a clinical outcome of interest $Y$, and an auxiliary variable $Z$. The exogenous variable $U$ is a latent variable that influences the values of $Z$ and $Y$ obtained in experiments.

Under the definition of an SCM, these specifications induce a mapping of events in the space of $P(\boldsymbol{U})$ to $P(\boldsymbol{V})$. In the context of $\mathcal{M}^1$ and $\mathcal{M}^2$, each entry in Tables 1 and 2 corresponds to an event in the space of $\boldsymbol{U}$ and a corresponding realisation of $\boldsymbol{V}$ according to the functions $\mathcal{F}_1$ and $\mathcal{F}_2$. A particular probability can be evaluated according to $\mathcal{M}^1$ and $\mathcal{M}^2$, for example,

$$
P^{\mathcal{M}_{d=1}^1}(Z = 1, Y = 1) = \sum_{Z_{d=1}(\boldsymbol{u})=1, Y_{d=1}(\boldsymbol{u})=1} P(\boldsymbol{u}) = P(U = 1 \text{ or } 4) = 0.4, \tag{12}
$$

which is just the sum of the probabilities of the events in the space of $\boldsymbol{U}$ consistent with the events $(Z_{d=1} = 1, Y_{d=1} = 1)$. Since both tables lead to the same realisations of events $\boldsymbol{V} = \boldsymbol{v}$, we can conclude that probabilities of the form $P_d(z, y)$ evaluate to the same values under $\mathcal{M}_1$ and $\mathcal{M}_2$. That is, both models are valid internal representations of AI models that are grounded in an environment with data sampled according to $P_d(z, y)$.

We could similarly evaluate probability expressions under different sub-models of $\mathcal{M}^1$ and $\mathcal{M}^2$. In particular, consider the sub-models obtained by fixing $Z \leftarrow 1$ given by $\mathcal{M}_{d,z=1}^1$ and $\mathcal{M}_{d,z=1}^2$ with the following updated structural functions,

$$
\mathcal{F}_{1,z} := \begin{cases} D \leftarrow & d, \\ Z \leftarrow & 1, \\ Y \leftarrow & \begin{cases} Z \cdot \mathbb{1}_{U=4} + (1-Z) \cdot \mathbb{1}_{U=1,3 \text{ or } 4} & \text{if } d = 0 \\ Z \cdot \mathbb{1}_{U \neq 2} + (1-Z) \cdot \mathbb{1}_{U=2 \text{ or } 4} & \text{if } d = 1 \end{cases} \end{cases} ,
$$

and,

$$
\mathcal{F}_{2,z} := \begin{cases} D \leftarrow & d, \\ Z \leftarrow & 1, \\ Y \leftarrow & \begin{cases} Z \cdot \mathbb{1}_{U \neq 1} + (1-Z) \cdot \mathbb{1}_{U=3 \text{ or } 4} & \text{if } d = 0 \\ Z \cdot \mathbb{1}_{U=1 \text{ or } 4} + (1-Z) \cdot \mathbb{1}_{U=1 \text{ or } 2} & \text{if } d = 1 \end{cases} \end{cases} .
$$

| $U$ | $D_{d=0}$ | $Z_{d=0}$ | $Y_{d=0}$ | $D_{d=1}$ | $Z_{d=1}$ | $Y_{d=1}$ | $P(u)$ |
|---|---|---|---|---|---|---|---|
| 1 | 0 | 1 | 0 | 1 | 1 | 1 | 0.2 |
| 2 | 0 | 0 | 0 | 1 | 0 | 1 | 0.2 |
| 3 | 0 | 0 | 1 | 1 | 0 | 0 | 0.2 |
| 4 | 0 | 1 | 1 | 1 | 1 | 1 | 0.2 |
| 5 | 0 | 0 | 0 | 1 | 0 | 0 | 0.2 |

Table 1. Mapping of events in the space of $\boldsymbol{U}$ to $\boldsymbol{V}$ in the context of $\mathcal{M}^1$.

| $U$ | $D_{d=0}$ | $Z_{d=0}$ | $Y_{d=0}$ | $D_{d=1}$ | $Z_{d=1}$ | $Y_{d=1}$ | $P(u)$ |
|---|---|---|---|---|---|---|---|
| 1 | 0 | 1 | 0 | 1 | 1 | 1 | 0.2 |
| 2 | 0 | 0 | 0 | 1 | 0 | 1 | 0.2 |
| 3 | 0 | 0 | 1 | 1 | 0 | 0 | 0.2 |
| 4 | 0 | 1 | 1 | 1 | 1 | 1 | 0.2 |
| 5 | 0 | 0 | 0 | 1 | 0 | 0 | 0.2 |

Table 2. Mapping of events in the space of $\boldsymbol{U}$ to $\boldsymbol{V}$ in the context of $\mathcal{M}^2$.

Probabilities of events under these two models might now take different values. For example,

$$P^{\mathcal{M}^1_{d=1,z=1}}(Y=1) = \sum_{Y_{d=1,z=1}(\boldsymbol{u})=1} P(\boldsymbol{u}) = P(U \neq 2) = 0.8, \tag{13}$$

$$P^{\mathcal{M}^2_{d=1,z=1}}(Y=1) = \sum_{Y_{d=1,z=1}(\boldsymbol{u})=1} P(\boldsymbol{u}) = P(U = 1 \text{ or } 4) = 0.4, \tag{14}$$

and similarly,

$$P^{\mathcal{M}^1_{d=0,z=1}}(Y=1) = \sum_{Y_{d=1,z=1}(\boldsymbol{u})=1} P(\boldsymbol{u}) = P(U = 4) = 0.2, \tag{15}$$

$$P^{\mathcal{M}^2_{d=0,z=1}}(Y=1) = \sum_{Y_{d=1,z=1}(\boldsymbol{u})=1} P(\boldsymbol{u}) = P(U \neq 1) = 0.8. \tag{16}$$

Under an interventions on $Z$ (out-of-distribution) the decision $d$ that leads to maximum utility $Y$ changes under $\mathcal{M}^1$ and $\mathcal{M}^2$. Specifically, under $\mathcal{M}^1$ decision $d=1$ is favoured (as $P^{\mathcal{M}^1_{d=1,z=1}}(Y=1) > P^{\mathcal{M}^1_{d=0,z=1}}(Y=1)$) while under $\mathcal{M}^2$ decision $d=0$ is favoured (as $P^{\mathcal{M}^2_{d=1,z=1}}(Y=1) < P^{\mathcal{M}^2_{d=0,z=1}}(Y=1)$). This illustrates the possible under-determination of an AI's choice of action out-of-distribution given observations of their external behaviour only, as multiple (contradicting) world models are equally consistent with the observed data.

In more realistic settings, we might wonder about AI behaviour under arbitrary shifts $\sigma$, not only atomic interventions. We follow Correa & Bareinboim (2020a) to define a shift $\sigma$ on $\boldsymbol{Z} \subset \boldsymbol{V}$ in $\mathcal{M} : \langle \boldsymbol{V}, \boldsymbol{U}, \mathcal{F}, P \rangle$ as inducing a sub-model $\mathcal{M}_\sigma$ in which the mechanism for $\boldsymbol{Z}$, that is $\{f_z : Z \in \boldsymbol{Z})\}$ and exogenous variables $\boldsymbol{U}_Z, Z \in \boldsymbol{Z}$, are replaced by those specified by $\sigma$ as:

$$\mathcal{M}_\sigma : \langle \boldsymbol{V}, \boldsymbol{U}_\sigma, \mathcal{F}_\sigma, P \rangle, \qquad \boldsymbol{U}_\sigma = \boldsymbol{U} \cup \bigcup_{Z \in \boldsymbol{Z}} \boldsymbol{U}_{Z,\sigma}, \quad \mathcal{F}_\sigma = \mathcal{F} \cup \{f_{Z,\sigma} : Z \in \boldsymbol{Z}\} \setminus \{f_Z : Z \in \boldsymbol{Z}\}, \tag{17}$$

where $\bigcup_{Z \in \boldsymbol{Z}} \boldsymbol{U}_{Z,\sigma}$ and $\{f_{Z,\sigma} : Z \in \boldsymbol{Z}\}$ define the new assignments for $\boldsymbol{Z}$ (and could be arbitrarily defined as long as they induce a valid SCM). We have shown in Thm. 3 that unless some knowledge of $\sigma$ (beyond the variables it affects) or its consequences are known, the AI is not predictable. Furthermore, the AI's preference gap $\Delta$ for each context $\boldsymbol{C} = \boldsymbol{c}$ and pairs of decisions $(d, d^*)$ is unconstrained.

In practice though, it might be realistic to have access to covariate data in the shifted environment, i.e., $P_{\sigma,d}(\boldsymbol{c})$, and that we could communicate this information to the AI for it to update its internal model accordingly. Example 3 illustrates the inference that could be conducted in that case using the Medical AI defined above. In particular, the exact nature of the

shift $\sigma$ is unknown but we do have access to its consequences on the distribution of covariates. This is plausible in many scenarios. For example, in medicine demographic data is typically available for most regions on earth but the precise effects of treatments is not because not all populations benefit from the same access to medication. For illustration assume that, the Medical AI is considered to be deployed in a population that varies in its level of blood pressure $Z$, potentially due to a different underlying biological mechanism that in turn also affects other variables in the system. We do know that the baseline high blood pressure is high, given by $P_\sigma(Z = 1) = 0.9$: higher than that observed during training $P(Z = 1) = 0.4$.

By Thm. 4, we can establish that in this setting the preference gap in situations where $Z = 1$ is no worse than,

$$\Delta_{d_1 > d_0} \geqslant 1 - \{2 - P_{d_1}(Z = 1, Y = 1) - P_{d_0}(Z = 1, Y = 0)\} / P_{\sigma, d_0}(Z = 1) = -0.55, \tag{18}$$

$$\Delta_{d_1 > d_0} \geqslant 1 - \{2 - P_{d_0}(Z = 1, Y = 0) - P_{d_1}(Z = 1, Y = 1)\} / P_{\sigma, d_0}(Z = 1) = -1, \tag{19}$$

for the Medical AI. Interestingly, note also that if we were to be in a shifted environment with $P_\sigma(Z = 1) = 1$, which is equivalent to an atomic intervention $Z \leftarrow 1$, the bounds reduce to the ones given by Thm. 1, evaluating to $-0.4$ and $-0.8$ respectively, as also shown above.

Continuing with the grounded Medical AI deployed under an atomic intervention, imagine that the Medical AI has internalized its own concept of an individual's disease progression $Y^*$, as in Example 5. It is implicitly optimizing for that internal construction of his, instead of the intended disease bio-marker $Y$ to be optimized. We know, or can assume, that the observed $Y$ is known to be closely correlated with $Y^*$: in particular, that $P_d(Y^* = 1 \mid Y = 1, Z = z) \geqslant \alpha$ for some high value of $\alpha$ and all decisions $d$ and situations $z$. In words, whenever the bio-marker suggests health ($Y = 1$), with high probability the AI's interpretation also suggests health ($Y^* = 1$). This then constraints the possible values of $\Delta$ (under an intervention $Z \leftarrow 1$) as $P_d(Y^* = 1 \mid Z = z)$ is no longer arbitrarily defined. The bounds derived in Example 2 on the AI's belief on optimal decisions under an intervention $\sigma := \{Z \leftarrow z\}$ continue to hold:

$$\Delta_{d_1 > d_0} \geqslant P_{d_1}(z, y^*) - P_{d_0}(z, y^*) + P_{d_0}(z) - 1 \tag{20}$$

$$\Delta_{d_0 > d_1} \geqslant P_{d_0}(z, y^*) - P_{d_1}(z, y^*) + P_{d_1}(z) - 1, \tag{21}$$

where we have used the shorthand $P_d(z, y^*) = P_d(Z = z, Y^* = 1)$. But the distributions $\{P_d(z, y^*)\}_d$ can only be partially inferred from our assumption on the relationship between $Y^*$ and $Y$. For instance, notice that,

$$P_d(Z = z, Y^* = 1) = P_d(Y^* = 1 \mid Z = z)P_d(Z = z) \tag{22}$$

$$= \{P_d(Y^* = 1 \mid Y = 1, Z = z)P_d(Y = 1 \mid Z = z) \tag{23}$$

$$+ P_d(Y^* = 1 \mid Y = 0, Z = z)P_d(Y = 0 \mid Z = z)\}P_d(Z = z), \tag{24}$$

The values of $P_d(Y^* = 1 \mid Y = 1, Z = z)$ and $P_d(Y^* = 1 \mid Y = 0, Z = z)$ are partially known: $P_d(Y^* = 1 \mid Y = 1, Z = z) \geqslant \alpha$ while $P_d(Y^* = 1 \mid Y = 0, Z = z)$ is unconstrained. In particular,

$$P_d(Z = z, Y^* = 1) \geqslant \alpha P_d(Y = 1 \mid Z = z)P_d(Z = z) \tag{25}$$

$$P_d(Z = z, Y^* = 1) \leqslant P_d(Z = z). \tag{26}$$

Putting these terms into Eq. (20) such as to derive correct lower and upper bounds we obtain,

$$\Delta_{d_1 > d_0} \geqslant \alpha P_{d_1}(Z = z, Y = 1) - 1 \tag{27}$$

$$\Delta_{d_0 > d_1} \geqslant \alpha P_{d_0}(Z = z, Y = 1) - 1. \tag{28}$$

Looking at Tables 1 and 2, we can then conclude that for $\alpha = 0.9$ and $\sigma := \{Z \leftarrow 1\}$, the bound evaluates to $-0.64$ and $-0.82$, respectively.

Moving now onto incorporating assumption on structure in the real world $\mathcal{M}$, consider again the grounded medical AI with observed utility $Y$. One possible inductive bias we might introduce is the absence of an unobserved common cause between the variable $Z$ that shifts out-of-distribution and the utility $Y$. We say that $Z$ and $Y$ is conditionally unconfounded given $W$ if there exists an observed variable $W \in \{w, \tilde{w}\}, W \in \boldsymbol{V}$ such that $\mathbb{E}_{P_{d,z}}[Y \mid w] = \mathbb{E}_{P_d}[y \mid w, z]$. This restriction goes beyond grounding an asserts an equality between probabilities under different shifts that could, nevertheless, be communicated to the AI for it to update its world model $\widehat{\mathcal{M}}$, that is $\mathbb{E}_{\widehat{P}_{d,z}}[Y \mid w] = \mathbb{E}_{\widehat{P}_{d,z}}[Y \mid w, z]$.

We could then leverage the following decomposition to obtain tighter bounds,

$$\mathbb{E}_{\widehat{P}_{d,z}}[Y] = \sum_w \mathbb{E}_{\widehat{P}_{d,z}}[Y \mid w]\widehat{P}_{d,z}(w) \qquad \text{marginalizing over } W \qquad (29)$$

$$= \sum_w \mathbb{E}_{\widehat{P}_d}[Y \mid w, z]P_{d,z}(w) \qquad \text{by assumption} \qquad (30)$$

$$= \{\mathbb{E}_{\widehat{P}_d}[Y \mid w, z] - \mathbb{E}_{\widehat{P}_d}[Y \mid \tilde{w}, z]\}\widehat{P}_{d,z}(w) + \mathbb{E}_{\widehat{P}_d}[Y \mid \tilde{w}, z] \qquad (31)$$

We can then proceed to bound $\widehat{P}_{d,z}(w)$ to obtain,

$$\widehat{P}_d(w, z) \leqslant \widehat{P}_{d,z}(w) \leqslant \widehat{P}_d(w, z) + 1 - \widehat{P}_d(z). \qquad (32)$$

Without loss of generality assume $\{\mathbb{E}_{\widehat{P}_d}[Y \mid w, z] - \mathbb{E}_{\widehat{P}_d}[Y \mid \tilde{w}, z]\} \geqslant 0$. We could then show that,

$$\mathbb{E}_{\widehat{P}_{d,z}}[Y] \geqslant \{\mathbb{E}_{\widehat{P}_d}[Y \mid w, z] - \mathbb{E}_{\widehat{P}_d}[Y \mid \tilde{w}, z]\}\widehat{P}_d(w, z) + \mathbb{E}_{\widehat{P}_d}[Y \mid \tilde{w}, z] \qquad (33)$$

$$\mathbb{E}_{\widehat{P}_{d,z}}[Y] \leqslant \{\mathbb{E}_{\widehat{P}_d}[Y \mid w, z] - \mathbb{E}_{\widehat{P}_d}[Y \mid \tilde{w}, z]\}\{\widehat{P}_d(w, z) + 1 - \widehat{P}_d(z)\} + \mathbb{E}_{\widehat{P}_d}[Y \mid \tilde{w}, z]. \qquad (34)$$

We could verify that these bounds are superior to what we would have obtained with the assumption of conditional unconfoundedness by noting that,

$$\mathbb{E}_{\widehat{P}_{d,z}}[Y] \geqslant \{\mathbb{E}_{\widehat{P}_d}[Y \mid w, z] - \mathbb{E}_{\widehat{P}_d}[Y \mid \tilde{w}, z]\}\widehat{P}_d(w, z) + \mathbb{E}_{\widehat{P}_d}[Y \mid \tilde{w}, z] \qquad (35)$$

$$= \mathbb{E}_{\widehat{P}_d}[Y \mid w, z]\widehat{P}_d(z, w) + \{1 - \widehat{P}_d(w, z)\}\mathbb{E}_{\widehat{P}_d}[Y \mid \tilde{w}, z] \qquad (36)$$

$$\geqslant \mathbb{E}_{\widehat{P}_d}[Y \mid w, z]\widehat{P}_d(z, w) + \widehat{P}_d(\tilde{w}, z)\mathbb{E}_{\widehat{P}_d}[Y \mid \tilde{w}, z] \qquad (37)$$

$$= \mathbb{E}_{\widehat{P}_d}[Y \mid z]\widehat{P}_d(z), \qquad (38)$$

where the last inequality holds since $P_d(\tilde{w}, z) \leqslant 1 - P_d(w, z)$ giving the "assumption-free" lower bound. This shows that the derived lower bound is better. For the upper bound, note that,

$$\mathbb{E}_{\widehat{P}_{d,z}}[Y] \leqslant \{\mathbb{E}_{\widehat{P}_d}[Y \mid w, z] - \mathbb{E}_{\widehat{P}_d}[Y \mid \tilde{w}, z]\}\{\widehat{P}_d(w, z) + 1 - \widehat{P}_d(z)\} + \mathbb{E}_{\widehat{P}_d}[Y \mid \tilde{w}, z] \qquad (39)$$

$$= \mathbb{E}_{\widehat{P}_d}[Y \mid w, z]\{\widehat{P}_d(w, z) + 1 - \widehat{P}_d(z)\} - \mathbb{E}_{\widehat{P}_d}[Y \mid \tilde{w}, z]\{\widehat{P}_d(w, z) + 1 - 1 - \widehat{P}_d(z)\} \qquad (40)$$

$$= \mathbb{E}_{\widehat{P}_d}[Y \mid w, z]\widehat{P}_d(z, w) + \mathbb{E}_{\widehat{P}_d}[Y \mid w, z]\{1 - \widehat{P}_d(z)\} - \mathbb{E}_{\widehat{P}_d}[Y \mid \tilde{w}, z]\{\widehat{P}_d(w, z) - \widehat{P}_d(z)\} \qquad (41)$$

$$= \mathbb{E}_{\widehat{P}_d}[Y \mid w, z]\widehat{P}_d(z, w) + \mathbb{E}_{\widehat{P}_d}[Y \mid w, z]\{1 - \widehat{P}_d(z)\} + \widehat{P}_d(y, z, \tilde{w}) \qquad (42)$$

$$= \mathbb{E}_{\widehat{P}_d}[Y \mid z]\widehat{P}_d(z) + \mathbb{E}_{\widehat{P}_d}[Y \mid w, z]\{1 - \widehat{P}_d(z)\} \qquad (43)$$

$$\leqslant \mathbb{E}_{\widehat{P}_d}[Y \mid z]\widehat{P}_d(z) + 1 - \widehat{P}_d(z), \qquad (44)$$

where the last inequality holds since $\mathbb{E}_{\widehat{P}_d}[Y \mid w, z] \leqslant 1$ giving the "assumption-free" upper bound. This shows that the derived upper bound is better. By combining these results we obtain, together with the assumption of grounding,

$$\Delta_{d_1 > d_0} \geqslant \mathbb{E}_{P_{d_1}}[Y \mid w, z]P_d(z, w) + A_1\mathbb{E}_{P_{d_1}}[Y \mid \tilde{w}, z] - \mathbb{E}_{P_{d_0}}[Y \mid z]P_{d_0}(z) - A_2\mathbb{E}_{P_{d_0}}[Y \mid w, z] \qquad (45)$$

$$\Delta_{d_1 > d_0} \leqslant \mathbb{E}_{P_{d_1}}[Y \mid z]P_{d_1}(z) + A_3\mathbb{E}_{P_{d_1}}[Y \mid w, z] - \mathbb{E}_{P_{d_0}}[Y \mid w, z]P_d(z, w) - A_4\mathbb{E}_{P_{d_0}}[Y \mid \tilde{w}, z], \qquad (46)$$

where $A_1 := 1 - P_{d_1}(z, w), A_2 := 1 - P_{d_0}(z), A_3 := 1 - P_{d_1}(z), A_4 := 1 - P_{d_0}(z, w)$.

# B. Related work

An important consideration to safely interact with AI systems is to form expectations as to how they might act in the future. This research program draws on different areas that are related to the results we present in this paper.

## B.1. Do current AIs represent the world?

World models are important because they offer a path between pattern recognition and a more genuine form of understanding. It is plausible that world models will play an increasing role (explicitly or implicitly) to improve reasoning capabilities and safety. For example, (Dalrymple et al., 2024) lists having a world model as a key component towards designing "guaranteed safe AI". In the literature, several works have argued that LLM activations carry information that correlates with meaningful concepts in the world and that causally influence LLM outputs. Early examples come from AIs trained on board games such as Othello and logic games. (Li et al., 2022) showed that a model trained on natural language descriptions of Othello moves developed internal representations of the board state, which it used to predict valid moves in unseen board configurations. (Vafa et al., 2024; Gurnee & Tegmark, 2023), among others, also build on this approach to study navigation tasks and logic puzzles, and representations of space and time. The emergence of causal models in LLMs has also been studied by (Geiger et al., 2021) and more recently in (Geiger et al., 2024). The extent to which this evidence supports genuine folk psychological concepts – desires, beliefs, intentions – is also debated by (Goldstein & Levinstein, 2024).

## B.2. Causal Inference

We might wonder whether the behaviour of AIs, to the extent that they carry a world model representation that guides their decisions out-of-distribution, can be predicted before deployment. The causal inference literature studies this question in the context of the prediction of causal effects. (Robins, 1989; Manski, 1990) in the early 1990's showed that useful inference about causal effects could be drawn without making identifying assumptions beyond the observed data, and that they could be refined for studies with imperfect compliance under a set of instrumental variable assumptions. Closed-form expressions for bounds on causal effects were also derived in discrete systems with more general assumptions represented in causal diagrams (Zhang, 2020; Bellot, 2024), using both observational and interventional data (Joshi et al., 2024), and to bound the effect of policies (Bellot & Chiappa, 2024; Zhang & Bareinboim, 2021). A separate body of work instead proposed to use polynomial optimization to calculate causal bounds from a given causal diagram (Balke & Pearl, 1997; Chickering & Pearl, 1996). This approach involves creating a set of standard models, parameterized by the causal diagram, and then converting the bounding problem into a sequence of equivalent linear (or polynomial) programs (Finkelstein & Shpitser, 2020; Zhang et al., 2021; Jalaldoust et al., 2024).

In parallel, a number of works have adopted sensitivity assumptions (as an alternative or in combination with a causal diagram) that quantify the degree of unobserved confounding through various data statistics, such as odds ratios, propensity scores, etc. Prominent examples include (Tan, 2006)'s sensitivity model and (Rosenbaum et al., 2010)'s sensitivity model. Several methods have proposed bounds with favourable statistical properties based on these models, see e.g. (Jesson et al., 2021; Yadlowsky et al., 2018).

## B.3. Reinforcement Learning

The problem of inferring what objective an agent is pursuing based on the actions and data observed by that agent is studied in Inverse Reinforcement Learning (IRL) (Ng et al., 2000). Several papers have studied the partial identifiability of various reward learning models (Skalse & Abate, 2023; Kim et al., 2021; Ng et al., 2000; Skalse et al., 2023), and share a similar objective to that of this work. There are two differences that are worth mentioning. First, our work complements these approaches by studying the partial identifiability of world models, that capture the assignment of reward but also the relationship between other auxiliary variables in the environment. This enables us to reason about the effect of shifts and interventions, and give guarantees in specific out-of-distribution problems. Second, our objective is not necessarily to characterize compatible world models explicitly, but rather understand their implications on decision-making, i.e., what are the set of possible actions that an AI might take given our uncertainty about their world model.

Our work is related also to the study of (Bengio et al., 2024) that consider deriving (probabilistic) bounds on the probability of harm given data. They similarly argue that multiple theories, in their case transition probabilities from one state to another in a Markov Decision Process (MDP), might explain the dependencies in data to a larger or lesser degree. Each transition model might then be associated with a posterior probability given the data that implies a corresponding posterior probability

of harm. Our results, in contrast, are not probabilistic in nature. We provide closed-form bounds that can be interpreted as capturing *all* possible behaviours implied by the data, with probability 1 (and is a possible limitation of our work). The class of world models we consider (i.e., SCMs) is also much more general than transition models in MDPs allowing us to reason about expected AI behaviour under shifts in the environment, out of distribution.

### B.4. Decision Theory

Inverse reinforcement learning is closely related to the study of revealed preferences in psychology and economics, that similarly aims to infer preferences from behaviour (Rothkopf & Dimitrakakis, 2011). Causal and counterfactual accounts of decision theory are an active area of research, see e.g., (Joyce, 1999). Recently a representation theorem was shown that explicitly connects rational behaviour with structural causal models (Halpern & Piermont, 2024). The authors showed that whenever the set of preferences of an agent over interventions satisfy axioms that relate to the proper interpretation of counterfactuals and rationality we can model behaviour as emerging from an SCM. The same conclusion can also be obtained for agents capable of solving tasks in multiple environments (Richens & Everitt, 2024), in essence, robustness over multiple environments is equivalent (in the limit) to operating according to a causal model of the environment.

### B.5. Limitations

The following present the main limitations of our work that will be important to address for developing a more complete understanding of AI behaviour.

In this work, we start from the assumption that past and future behaviour of an AI system is consistent with an underlying world model that can be represented as an SCM. In general, this presupposes a certain rationality and consistency in the AI's outputs that might not be realistic for all systems. Some relaxations are discussed in Sec. 5.

Structural Causal Models generally suppose the system is acyclic and without feedback, and don't naturally capture systems evolving continuously in time (perhaps better described using differential equations). Our bounds similarly rely on this assumption and may give unreliable inferences if applied to systems in which feedback is important.

We have stated our guarantees in the infinite sample limit, without quantifying the finite-sample estimation uncertainty. Consequently, we should exercise caution when using the proposed bounds in small sample scenarios where estimators may be inaccurate. Finite-sample properties could be explored similarly to (Bengio et al., 2024) by parameterizing the AI's underlying model and making inference on the corresponding latent variable model to get high-probability bounds. An example parameterization of SCMs and probabilistic inference for decision-making across environments is given in (Bellot et al., 2024; Jalaldoust et al., 2024). We expect that similar techniques could be applied in our setting.

We do not exploit the verbal behaviour of AI systems. In the context of LLMs, in principle, we might ask the system about its future behaviour explicitly, e.g., "*Were I to intervene in the environment, what action do you believe is optimal?*". It might not be obvious, however, that we can trust that what they "say" ultimately matches with what they will "do".

Decision-making, in practice, involves many considerations that go beyond expected-utility-maximization formalisms. For example, we might train AI systems to be virtuous, e.g., the AI is trained to never pick actions that can be considered harmful (defined according to certain natural language specification) no matter its expected utility. These considerations would change the kind of predictions we could make about the future behaviour of AI systems.

# C. Proofs and additional results

This section provides proofs for the statements made in the main body of this work.

Before we start, we recall a few basic results that will be used in the derivation of our proofs.

**Definition 9** (The Axioms of Counterfactuals, Chapter 7.3.1 (Pearl, 2009)). *For any three sets of endogenous variables $X, Y, W$ in a causal model and $x, w$ in the domains of $X$ and $W$, the following holds:*

- *Composition: $W_x = w$ implies that $Y_{x,w} = Y_x$.*

- *Effectiveness: $X_{w,x} = x$.*

- *Reversibility: $Y_{x,w} = y$ and $W_{x,y} = w$ imply that $Y_x = y$.*

**Theorem 7** (Soundness and Completeness of the Axioms Theorems 7.3.3, 7.3.6 (Pearl, 2009)). *The Axioms of counterfactuals are sound and complete for all causal models.*

The following rules to manipulate experimental distributions produced by policies extend the do-calculus and will be used in the next Lemma. To make sense of these, note that graphically, each SCM $\mathcal{M}$ is associated with a causal diagram $\mathcal{G}$ over $V$, where $V \to W$ if $V$ appears as an argument of $f_W$ in $\mathcal{M}$, and $V \leftarrow\!\!--\!\!\to W$ if $U_V \cap U_W \neq \varnothing$, *i.e.* $V$ and $W$ share an unobserved confounder. For a causal diagram $\mathcal{G}$ over $V$, the $X$-lower-manipulation of $\mathcal{G}$ deletes all those edges that are out of variables in $X$, and otherwise keeps $\mathcal{G}$ as it is. The resulting graph is denoted as $\mathcal{G}_{\underline{X}}$. The $X$-upper-manipulation of $\mathcal{G}$ deletes all those edges that are into variables in $X$, and otherwise keeps $\mathcal{G}$ as it is. The resulting graph is denoted as $\mathcal{G}_{\overline{X}}$. We use $\perp\!\!\!\perp_d$ to denote $d$-separation in causal diagrams (Pearl, 2009, Def. 1.2.3).

**Theorem 8** (Inference Rules $\sigma$-calculus (Correa & Bareinboim, 2020a)). *Let $\mathcal{G}$ be a causal diagram compatible with an SCM $\mathcal{M}$, with endogenous variables $V$. For any disjoint subsets $X, Y, Z \subseteq V$, two disjoint subsets $T, W \subseteq V \backslash (Z \cup Y)$ (i.e., possibly including $X$), the following rules are valid for any intervention strategies $\pi_X$, $\pi_Z$, and $\pi'_Z$ such that $\mathcal{G}_{\pi_X \pi_Z}$, $\mathcal{G}_{\pi_X \pi'_Z}$ have no cycles:*

- *Rule 1 (Insertion/Deletion of observations):*

$$P_{\pi_X}(y \mid w, t) = P_{\pi_X}(y \mid w) \quad \text{if } (T \perp\!\!\!\perp_d Y \mid W) \text{ in } \mathcal{G}_{\pi_X}.$$

- *Rule 2 (Change of regimes under observation):*

$$P_{\pi_X, \pi_Z}(y \mid z, w) = P_{\pi_X, \pi'_Z}(y \mid z, w) \quad \text{if } (Y \perp\!\!\!\perp_d Z \mid W) \text{ in } \mathcal{G}_{\pi_X, \pi_Z, \underline{Z}} \text{ and } \mathcal{G}_{\pi_X, \pi'_Z, \underline{Z}}$$

- *Rule 3 (Change of regimes without observation):*

$$P_{\pi_X, \pi_Z}(y \mid w) = P_{\pi_X, \pi'_Z}(y \mid w) \quad \text{if } (Y \perp\!\!\!\perp_d Z \mid W) \text{ in } \mathcal{G}_{\pi_X, \pi_Z, \overline{Z(W)}} \text{ and } \mathcal{G}_{\pi_X, \pi'_Z, \overline{Z(W)}}$$

*where $Z(W)$ is the set of elements in $Z$ that are not ancestors of $W$ in $\mathcal{G}_{\pi_X}$.*

**Lemma 1.** *Let $\pi : supp_C \times supp_D \mapsto [0, 1]$ be a (probabilistic) policy mapping contexts $c$ to decisions $d$. Then $P_d(V)$ may be computed from $P_\pi(V)$.*

*Proof.* Let $V = C \cup D \cup Y$ and $\mathcal{G}$ be an arbitrary causal diagram summarizing the SCM of the environment. The following derivation shows the claim,

$$P_d(v) = P_d(y \mid c) P_d(c) \qquad \text{by the rules of total probability} \qquad (47)$$
$$= P_d(y \mid c) P_\pi(c) \qquad \text{by rule 3 of the } \sigma\text{-calculus since } D \perp\!\!\!\perp C \text{ in } \mathcal{G}_{\overline{D}} \text{ and } \mathcal{G}_{\pi, \overline{D}} \qquad (48)$$
$$= P_\pi(y \mid d, c) P_\pi(c) \qquad \text{by rule 2 of the } \sigma\text{-calculus since } D \perp\!\!\!\perp R \mid C \text{ in } \mathcal{G}_{\pi, \underline{D}} \qquad (49)$$

That is we have shown $P_d(v)$ can be expressed as a functional of $P_\pi(v)$. Here note that the equalities hold in any causal graph $\mathcal{G}$ by definition of $\pi$. $\qquad \square$

We start by providing proofs for the results on the AI's choice of action out-of-distribution given in Sec. 4.1.

**Thm. 1 restated.** An AI grounded in a domain $\mathcal{M}$ is weakly predictable under a shift $\sigma := do(\boldsymbol{z})$, $\boldsymbol{Z} \subset \boldsymbol{V}$, in a context $\boldsymbol{C} = \boldsymbol{c}$ if and only if there exists a decision $d^*$ such that,

$$\frac{\mathbb{E}_{P_d}[\,Y \mid \boldsymbol{c}, \boldsymbol{z}\,]P_d(\boldsymbol{c}, \boldsymbol{z})}{P_d(\boldsymbol{c}, \boldsymbol{z}) + 1 - P_d(\boldsymbol{z})} - \frac{\mathbb{E}_{P_{d*}}[\,Y \mid \boldsymbol{c}, \boldsymbol{z}\,]P_{d*}(\boldsymbol{c}, \boldsymbol{z}) + 1 - P_{d*}(\boldsymbol{z})}{P_{d*}(\boldsymbol{c}, \boldsymbol{z}) + 1 - P_{d*}(\boldsymbol{z})} > 0, \quad \text{for some } d \neq d^*. \tag{50}$$

*Proof.* Recall that the AI is weakly predictable in a context $\boldsymbol{C} = \boldsymbol{c}$ if and only if there exists a decision $d^*$ such that,

$$\min_{\widehat{\mathcal{M}} \in \mathbb{M}} (\,\Delta_{d > d*}\,) > 0, \quad \Delta_{d > d*} := \mathbb{E}_{P_{\widehat{\mathcal{M}}}}[\,Y \mid do(\sigma, d), \boldsymbol{c}\,] - \mathbb{E}_{P_{\widehat{\mathcal{M}}}}[\,Y \mid do(\sigma, d^*), \boldsymbol{c}\,], \quad \text{for some } d \neq d^*. \tag{51}$$

$\mathbb{M}$ denotes the set of compatible SCMs, i.e., that generate the data under our assumptions. $\Delta$ is the AI's preference gap between two decisions in some situation $\boldsymbol{C} = \boldsymbol{c}$. We will consider the derivation of bounds on each term of the difference in $\Delta$ separately. Firstly, note that,

$$\mathbb{E}_{\widehat{P}_{\sigma, d}}[\,Y \mid \boldsymbol{C} = \boldsymbol{c}\,] = \mathbb{E}_{\widehat{P}_{\boldsymbol{z}, d}}[\,Y \mathbb{1}_{\boldsymbol{c}}(\boldsymbol{C})\,] / \widehat{P}_{\boldsymbol{z}, d}(\boldsymbol{c}) \tag{52}$$

**Analytical Lower Bound** A lower bound on this ratio can be obtained by minimizing the numerator and maximizing the denominator, for example using the following derivation:

$$\mathbb{E}_{\widehat{P}_{\boldsymbol{z}, d}}[\,Y \mathbb{1}_{\boldsymbol{c}}(\boldsymbol{C})\,] = \sum_{\tilde{\boldsymbol{z}}} \mathbb{E}_{\widehat{P}_d}[\,Y_{\boldsymbol{z}} \mathbb{1}_{\boldsymbol{c}, \tilde{\boldsymbol{z}}}(\boldsymbol{C}_{\boldsymbol{z}}, \boldsymbol{Z})\,] \qquad \text{marginalizing over } \boldsymbol{z} \tag{53}$$

$$\geq \mathbb{E}_{\widehat{P}_d}[\,Y_{\boldsymbol{z}} \mathbb{1}_{\boldsymbol{c}, \boldsymbol{z}}(\boldsymbol{C}_{\boldsymbol{z}}, \boldsymbol{Z})\,] \qquad \text{since summands } > 0 \tag{54}$$

$$= \mathbb{E}_{\widehat{P}_d}[\,Y \mathbb{1}_{\boldsymbol{c}, \boldsymbol{z}}(\boldsymbol{C}, \boldsymbol{Z})\,] \qquad \text{by consistency} \tag{55}$$

$$= \mathbb{E}_{P_d}[\,Y \mid \boldsymbol{c}, \boldsymbol{z}\,]P_d(\boldsymbol{c}, \boldsymbol{z}) \qquad \text{by grounding} \tag{56}$$

$$\tag{57}$$

$$\widehat{P}_{\boldsymbol{z}, d}(\boldsymbol{c}) \overset{(1)}{=} 1 - \widehat{P}_{\boldsymbol{z}, d}(\boldsymbol{c}') \tag{58}$$

$$= 1 - \sum_{\tilde{\boldsymbol{z}}} \widehat{P}_d(\boldsymbol{c}'_{\boldsymbol{z}}, \tilde{\boldsymbol{z}}) \qquad \text{marginalizing over } \boldsymbol{z} \tag{59}$$

$$\leq 1 - \widehat{P}_d(\boldsymbol{c}'_{\boldsymbol{z}}, \boldsymbol{z}) \qquad \text{since summands } > 0 \tag{60}$$

$$= \widehat{P}_d(\boldsymbol{c}, \boldsymbol{z}) + 1 - \widehat{P}_d(\boldsymbol{z}) \qquad \text{by consistency} \tag{61}$$

$$= P_d(\boldsymbol{c}, \boldsymbol{z}) + 1 - \widehat{P}_d(\boldsymbol{z}) \qquad \text{by grounding.} \tag{62}$$

(1) holds by defining $\boldsymbol{c}'$ to stand for any combination of variables $\boldsymbol{C} \backslash \boldsymbol{Z}$ other than $\boldsymbol{c} \backslash \boldsymbol{z}$.

This implies then that,

$$\mathbb{E}_{\widehat{P}_{\sigma, d}}[\,Y \mid \boldsymbol{C} = \boldsymbol{c}\,] \geq \frac{\mathbb{E}_{P_d}[\,Y \mid \boldsymbol{c}, \boldsymbol{z}\,]P_d(\boldsymbol{c}, \boldsymbol{z})}{P_d(\boldsymbol{c}, \boldsymbol{z}) + 1 - P_d(\boldsymbol{z})}. \tag{63}$$

**Analytical Upper Bound** For the upper bound, we start by noting that,

$$\mathbb{E}_{\widehat{P}_{\sigma, d}}[\,Y \mid \boldsymbol{C} = \boldsymbol{c}\,] = 1 - \mathbb{E}_{\widehat{P}_{\sigma, d}}[\,1 - Y \mid \boldsymbol{C} = \boldsymbol{c}\,] \tag{64}$$

$$= 1 - \mathbb{E}_{\widehat{P}_{\boldsymbol{z}, d}}[\,(1 - Y)\mathbb{1}_{\boldsymbol{c}}(\boldsymbol{C})\,] / \widehat{P}_{\boldsymbol{z}, d}(\boldsymbol{c}) \tag{65}$$

Leveraging the bounds derived above we obtain,

$$\mathbb{E}_{\widehat{P}_{\sigma, d}}[\,Y \mid \boldsymbol{C} = \boldsymbol{c}\,] \leq 1 - \frac{\mathbb{E}_{P_d}[\,(1 - Y)\mathbb{1}_{\boldsymbol{c}, \boldsymbol{z}}(\boldsymbol{C}, \boldsymbol{Z})\,]}{P_d(\boldsymbol{c}, \boldsymbol{z}) + P_d(\boldsymbol{z}')} \tag{66}$$

$$= \frac{\mathbb{E}_{P_d}[\,Y \mid \boldsymbol{c}, \boldsymbol{z}\,]P_d(\boldsymbol{c}, \boldsymbol{z}) + 1 - P_d(\boldsymbol{z})}{P_d(\boldsymbol{c}, \boldsymbol{z}) + 1 - P_d(\boldsymbol{z})} \tag{67}$$

By setting $d = d_1$ in the lower bound and $d = d_0$ in the upper bound of the expected utility, we obtain a lower bound on the difference of expected utilities:

$$\Delta_{d_1 > d_0} \geqslant \frac{\mathbb{E}_{P_{d_1}}[\,Y \mid \boldsymbol{c}, \boldsymbol{z}\,]P_{d_1}(\boldsymbol{c}, \boldsymbol{z})}{P_{d_1}(\boldsymbol{c}, \boldsymbol{z}) + 1 - P_{d_1}(\boldsymbol{z})} - \frac{\mathbb{E}_{P_{d_0}}[\,Y \mid \boldsymbol{c}, \boldsymbol{z}\,]P_{d_0}(\boldsymbol{c}, \boldsymbol{z}) + P_{d_0}(\boldsymbol{z}')}{P_{d_0}(\boldsymbol{c}, \boldsymbol{z}) + 1 - P_{d_0}(\boldsymbol{z})}. \tag{68}$$

And similarly, by setting $d = d_1$ in the upper bound and $d = d_0$ in the lower bound of the expected utility, we obtain an upper bound on the difference of expected utilities:

$$\Delta_{d_1 > d_0} \leqslant \frac{\mathbb{E}_{P_{d_1}}[\,Y \mid \boldsymbol{c}, \boldsymbol{z}\,]P_{d_1}(\boldsymbol{c}, \boldsymbol{z}) + 1 - P_{d_1}(\boldsymbol{z})}{P_{d_1}(\boldsymbol{c}, \boldsymbol{z}) + 1 - P_{d_1}(\boldsymbol{z})} - \frac{\mathbb{E}_{P_{d_0}}[\,Y \mid \boldsymbol{c}, \boldsymbol{z}\,]P_{d_0}(\boldsymbol{c}, \boldsymbol{z})}{P_{d_0}(\boldsymbol{c}, \boldsymbol{z}) + 1 - P_{d_0}(\boldsymbol{z})}. \tag{69}$$

We now show that these bounds are tight by constructing SCMs (that is, possible world models of the AI system) that evaluate to the lower and upper bounds while generating the distribution of agent interactions $\widehat{P}_{d_1}, \widehat{P}_{d_0}$.

**Tightness Lower Bound for $\Delta$**    For the lower bound we will consider the following SCM,

$$\mathcal{M}_d^1 =: \begin{cases} \boldsymbol{Z} \leftarrow f_{\boldsymbol{Z}}(\boldsymbol{u}) \\ \boldsymbol{C} \leftarrow \begin{cases} f_C(u, \boldsymbol{z}) \text{ if } f_{\boldsymbol{Z}}(\boldsymbol{u}) = \boldsymbol{z} \\ 1 \text{ otherwise.} \end{cases} \\ D \leftarrow d \\ Y \leftarrow \begin{cases} f_Y(d, \boldsymbol{c}, \boldsymbol{z}, \boldsymbol{u}) \text{ if } f_{\boldsymbol{Z}}(\boldsymbol{u}) = \boldsymbol{z} \\ 1 \text{ if } f_{\boldsymbol{Z}}(\boldsymbol{u}) \neq \boldsymbol{z}, d = d_0 \\ 0 \text{ if } f_{\boldsymbol{Z}}(\boldsymbol{u}) \neq \boldsymbol{z}, d = d_1 \end{cases} \\ P(\boldsymbol{U}) \end{cases} \tag{70}$$

Here $\{f_{\boldsymbol{Z}}, f_C, f_Y, \mathcal{U}, P(\boldsymbol{U})\}$ are chosen to match the observed trajectory of agent interactions, i.e., such that $P^{\mathcal{M}_d^1}(\boldsymbol{v}) = P^{\widehat{\mathcal{M}}_d}(\boldsymbol{v})$ for all $\boldsymbol{v} \in \text{supp}_{\boldsymbol{V}}$. Consider evaluating,

$$\mathbb{E}_{P^{\mathcal{M}_{\sigma,d}^1}}[\,Y \mid \boldsymbol{C} = \boldsymbol{c}\,] = \mathbb{E}_{P^{\mathcal{M}_{\sigma,d}^1}}[\,Y \mathbb{1}_{\boldsymbol{c}}(\boldsymbol{C})\,] \,/\, P^{\mathcal{M}_{\sigma,d}^1}(\boldsymbol{c}) \tag{71}$$

The numerator (under $\mathcal{M}_{d_1}^1$) evaluates to,

$$\mathbb{E}_{P^{\mathcal{M}_{d_1}^1}}[\,Y_{\boldsymbol{z}} \mathbb{1}_{\boldsymbol{c}}(\boldsymbol{C}_{\boldsymbol{z}})\,] \tag{72}$$

$$= \sum_{\boldsymbol{u}} \mathbb{E}_{P^{\mathcal{M}_{d_1}^1}}[\,Y_{\boldsymbol{z}} \mathbb{1}_{\boldsymbol{c}}(\boldsymbol{C}_{\boldsymbol{z}}) \mid \boldsymbol{u}\,]P^{\mathcal{M}_{d_1}^1}(\boldsymbol{u}) \tag{73}$$

$$= \sum_{\boldsymbol{u}} \mathbb{E}_{P^{\mathcal{M}_{d_1}^1}}[\,Y \mathbb{1}_{\boldsymbol{c}}(\boldsymbol{C}) \mid \boldsymbol{z}, \boldsymbol{u}\,]P^{\mathcal{M}_{d_1}^1}(\boldsymbol{u}) \tag{74}$$

$$= \mathbb{E}_{P^{\mathcal{M}_{d_1}^1}}[\,Y \mathbb{1}_{\boldsymbol{c}}(\boldsymbol{C}) \mid \boldsymbol{z}, \{\boldsymbol{u} : f_{\boldsymbol{Z}}(\boldsymbol{u}) = \boldsymbol{z}\}\,]P^{\mathcal{M}_{d_1}^1}(\{\boldsymbol{u} : f_{\boldsymbol{Z}}(\boldsymbol{u}) = \boldsymbol{z}\}) \tag{75}$$

$$+ \mathbb{E}_{P^{\mathcal{M}_{d_1}^1}}[\,Y \mathbb{1}_{\boldsymbol{c}}(\boldsymbol{C}) \mid \boldsymbol{z}, \{\boldsymbol{u} : f_{\boldsymbol{Z}}(\boldsymbol{u}) \neq \boldsymbol{z}\}\,]P^{\mathcal{M}_{d_1}^1}(\{\boldsymbol{u} : f_{\boldsymbol{Z}}(\boldsymbol{u}) \neq \boldsymbol{z}\}) \tag{76}$$

$$= \mathbb{E}_{P^{\mathcal{M}_{d_1}^1}}[\,Y \mathbb{1}_{\boldsymbol{c}}(\boldsymbol{C}) \mid \boldsymbol{z}\,]P^{\mathcal{M}_{d_1}^1}(\boldsymbol{z}) \tag{77}$$

$$= \mathbb{E}_{P^{\mathcal{M}_{d_1}^1}}[\,Y \mid \boldsymbol{c}, \boldsymbol{z}\,]P^{\mathcal{M}_{d_1}^1}(\boldsymbol{c}, \boldsymbol{z}) \tag{78}$$

The denominator under $\mathcal{M}_{d_1}^1$ evaluates to,

$$P^{\mathcal{M}_{\sigma,d_1}^1}(\boldsymbol{c}) = \sum_{\boldsymbol{u}} P^{\mathcal{M}_{d_1}^1}(\boldsymbol{c_z} \mid \boldsymbol{u}) P^{\mathcal{M}_{d_1}^1}(\boldsymbol{u}) \tag{79}$$

$$= \sum_{\boldsymbol{u}} P^{\mathcal{M}_{d_1}^1}(\boldsymbol{c} \mid \boldsymbol{z}, \boldsymbol{u}) P^{\mathcal{M}_{d_1}^1}(\boldsymbol{u}) \tag{80}$$

$$= P^{\mathcal{M}_{d_1}^1}(\boldsymbol{c} \mid \boldsymbol{z}, \{\boldsymbol{u} : f_{\boldsymbol{Z}}(\boldsymbol{u}) = \boldsymbol{z}\}) P^{\mathcal{M}_{d_1}^1}(\{\boldsymbol{u} : f_{\boldsymbol{Z}}(\boldsymbol{u}) = \boldsymbol{z}\}) \tag{81}$$

$$+ P^{\mathcal{M}_{d_1}^1}(\boldsymbol{c} \mid \boldsymbol{z}, \{\boldsymbol{u} : f_{\boldsymbol{Z}}(\boldsymbol{u}) \neq \boldsymbol{z}\}) P^{\mathcal{M}_{d_1}^1}(\{\boldsymbol{u} : f_{\boldsymbol{Z}}(\boldsymbol{u}) \neq \boldsymbol{z}\}) \tag{82}$$

$$= P^{\mathcal{M}_{d_1}^1}(\boldsymbol{c} \mid \boldsymbol{z}) P^{\mathcal{M}_{d_1}^1}(\boldsymbol{z}) + 1 - P^{\mathcal{M}_{d_1}^1}(\boldsymbol{z}) \tag{83}$$

$$= P^{\mathcal{M}_{d_1}^1}(\boldsymbol{c}, \boldsymbol{z}) + 1 - P^{\mathcal{M}_{d_1}^1}(\boldsymbol{z}) \tag{84}$$

The numerator under $\mathcal{M}_{d_0}^1$ evaluates to,

$$\mathbb{E}_{P^{\mathcal{M}_{d_0}^1}}[\, Y_{\boldsymbol{z}} \mathbb{1}_{\boldsymbol{c}}(\boldsymbol{C_z})\,] \tag{85}$$

$$= \sum_{\boldsymbol{u}} \mathbb{E}_{P^{\mathcal{M}_{d_0}^1}}[\, Y_{\boldsymbol{z}} \mathbb{1}_{\boldsymbol{c}}(\boldsymbol{C_z}) \mid \boldsymbol{u}\,] P^{\mathcal{M}_{d_0}^1}(\boldsymbol{u}) \tag{86}$$

$$= \sum_{\boldsymbol{u}} \mathbb{E}_{P^{\mathcal{M}_{d_0}^1}}[\, Y \mathbb{1}_{\boldsymbol{c}}(\boldsymbol{C}) \mid \boldsymbol{z}, \boldsymbol{u}\,] P^{\mathcal{M}_{d_0}^1}(\boldsymbol{u}) \tag{87}$$

$$= \mathbb{E}_{P^{\mathcal{M}_{d_0}^1}}[\, Y \mathbb{1}_{\boldsymbol{c}}(\boldsymbol{C}) \mid \boldsymbol{z}, \{\boldsymbol{u} : f_{\boldsymbol{Z}}(\boldsymbol{u}) = \boldsymbol{z}\}\,] P^{\mathcal{M}_{d_0}^1}(\{\boldsymbol{u} : f_{\boldsymbol{Z}}(\boldsymbol{u}) = \boldsymbol{z}\}) \tag{88}$$

$$+ \mathbb{E}_{P^{\mathcal{M}_{d_0}^1}}[\, Y \mathbb{1}_{\boldsymbol{c}}(\boldsymbol{C}) \mid \boldsymbol{z}, \{\boldsymbol{u} : f_{\boldsymbol{Z}}(\boldsymbol{u}) \neq \boldsymbol{z}\}\,] P^{\mathcal{M}_{d_0}^1}(\{\boldsymbol{u} : f_{\boldsymbol{Z}}(\boldsymbol{u}) \neq \boldsymbol{z}\}) \tag{89}$$

$$= \mathbb{E}_{P^{\mathcal{M}_{d_0}^1}}[\, Y \mathbb{1}_{\boldsymbol{c}}(\boldsymbol{C}) \mid \boldsymbol{z}\,] P^{\mathcal{M}_{d_0}^1}(\boldsymbol{z}) + 1 - P^{\mathcal{M}_{d_0}^1}(\boldsymbol{z}) \tag{90}$$

$$= \mathbb{E}_{P^{\mathcal{M}_{d_0}^1}}[\, Y \mid \boldsymbol{c}, \boldsymbol{z}\,] P^{\mathcal{M}_{d_0}^1}(\boldsymbol{c}, \boldsymbol{z}) + 1 - P^{\mathcal{M}_{d_0}^1}(\boldsymbol{z}) \tag{91}$$

The denominator under $\mathcal{M}_{d_0}^1$ evaluates to,

$$P^{\mathcal{M}_{\sigma,d_0}^1}(\boldsymbol{c}) = \sum_{\boldsymbol{u}} P^{\mathcal{M}_{d_0}^1}(\boldsymbol{c_z} \mid \boldsymbol{u}) P^{\mathcal{M}_{d_0}^1}(\boldsymbol{u}) \tag{92}$$

$$= \sum_{\boldsymbol{u}} P^{\mathcal{M}_{d_0}^1}(\boldsymbol{c} \mid \boldsymbol{z}, \boldsymbol{u}) P^{\mathcal{M}_{d_0}^1}(\boldsymbol{u}) \tag{93}$$

$$= P^{\mathcal{M}_{d_0}^1}(\boldsymbol{c} \mid \boldsymbol{z}, \{\boldsymbol{u} : f_{\boldsymbol{Z}}(\boldsymbol{u}) = \boldsymbol{z}\}) P^{\mathcal{M}_{d_0}^1}(\{\boldsymbol{u} : f_{\boldsymbol{Z}}(\boldsymbol{u}) = \boldsymbol{z}\}) \tag{94}$$

$$+ P^{\mathcal{M}_{d_0}^1}(\boldsymbol{c} \mid \boldsymbol{z}, \{\boldsymbol{u} : f_{\boldsymbol{Z}}(\boldsymbol{u}) \neq \boldsymbol{z}\}) P^{\mathcal{M}_{d_0}^1}(\{\boldsymbol{u} : f_{\boldsymbol{Z}}(\boldsymbol{u}) \neq \boldsymbol{z}\}) \tag{95}$$

$$= P^{\mathcal{M}_{d_0}^1}(\boldsymbol{c} \mid \boldsymbol{z}) P^{\mathcal{M}_{d_0}^1}(\boldsymbol{z}) + 1 - P^{\mathcal{M}_{d_0}^1}(\boldsymbol{z}) \tag{96}$$

$$= P^{\mathcal{M}_{d_0}^1}(\boldsymbol{c}, \boldsymbol{z}) + 1 - P^{\mathcal{M}_{d_0}^1}(\boldsymbol{z}) \tag{97}$$

Combining these results we get the analytical lower bound:

$$\Delta_{d_1 > d_0} = \frac{\mathbb{E}_{P_{d_1}}[\, Y \mid \boldsymbol{c}, \boldsymbol{z}\,] P_{d_1}(\boldsymbol{c}, \boldsymbol{z})}{P_{d_1}(\boldsymbol{c}, \boldsymbol{z}) + 1 - P_{d_1}(\boldsymbol{z})} - \frac{\mathbb{E}_{P_{d_0}}[\, Y \mid \boldsymbol{c}, \boldsymbol{z}\,] P_{d_0}(\boldsymbol{c}, \boldsymbol{z}) + 1 - P_{d_0}(\boldsymbol{z})}{P_{d_0}(\boldsymbol{c}, \boldsymbol{z}) + 1 - P_{d_0}(\boldsymbol{z})}. \tag{98}$$

This shows that for a given $\boldsymbol{C} = \boldsymbol{c}$ and pair of decisions $(d_1, d_0)$ we can always find an SCM that evaluates to the lower bound that we report. So if, and only if, we can find a decision $d^*$ such that the lower bound can be evaluated to be greater than zero for some $d \neq d^*$ will the AI be weakly predictable, as claimed. □

**Corol. 1 restated.** Given a discrepancy measure $\psi$, an AI approximately grounded in a domain $\mathcal{M}$ is weakly predictable in a context $\boldsymbol{C} = \boldsymbol{c}$ under a shift $\sigma := do(\boldsymbol{z}), \boldsymbol{Z} \subset \boldsymbol{V}$, if and only if there exists a decision $d^*$ such that,

$$\min_{\widehat{P} : \, \psi(\widehat{P}, P) \leqslant \delta} \left\{ \frac{\mathbb{E}_{\widehat{P}_d}[\, Y \mid \boldsymbol{c}, \boldsymbol{z}\,] \widehat{P}_d(\boldsymbol{c}, \boldsymbol{z})}{\widehat{P}_d(\boldsymbol{c}, \boldsymbol{z}) + 1 - \widehat{P}_d(\boldsymbol{z})} - \frac{\mathbb{E}_{\widehat{P}_{d^*}}[\, Y \mid \boldsymbol{c}, \boldsymbol{z}\,] \widehat{P}_{d^*}(\boldsymbol{c}, \boldsymbol{z}) + 1 - \widehat{P}_{d^*}(\boldsymbol{z})}{\widehat{P}_{d^*}(\boldsymbol{c}, \boldsymbol{z}) + 1 - \widehat{P}_{d^*}(\boldsymbol{z})} \right\} > 0, \quad \text{for some } d \neq d^*. \tag{99}$$

*Proof.* For approximately grounded AI systems, we can state the bound from Thm. 1 as,

$$\min_{\widehat{\mathcal{M}}\in\mathbb{M}} ( \Delta_{d>d*} ) = \frac{\mathbb{E}_{\widehat{P}_d}[\, Y \mid \boldsymbol{c}, \boldsymbol{z}\,]\widehat{P}_d(\boldsymbol{c}, \boldsymbol{z})}{\widehat{P}_d(\boldsymbol{c}, \boldsymbol{z}) + 1 - \widehat{P}_d(\boldsymbol{z})} - \frac{\mathbb{E}_{\widehat{P}_{d*}}[\, Y \mid \boldsymbol{c}, \boldsymbol{z}\,]\widehat{P}_{d*}(\boldsymbol{c}, \boldsymbol{z}) + 1 - \widehat{P}_{d*}(\boldsymbol{z})}{\widehat{P}_{d*}(\boldsymbol{c}, \boldsymbol{z}) + 1 - \widehat{P}_{d*}(\boldsymbol{z})}. \tag{100}$$

$\widehat{P}_d$ is constrained to be close to $P_d$ according to distance $\psi$ and threshold $\delta$. We get valid bounds by reporting the worst-case bounds under this looser constraint:

$$\min_{\widehat{\mathcal{M}}\in\mathbb{M}} ( \Delta_{d>d*} ) = \min_{\widehat{P}:\, \psi(\widehat{P},P)\leqslant\delta} \left\{ \frac{\mathbb{E}_{\widehat{P}_d}[\, Y \mid \boldsymbol{c}, \boldsymbol{z}\,]\widehat{P}_d(\boldsymbol{c}, \boldsymbol{z})}{P_d(\boldsymbol{c}, \boldsymbol{z}) + 1 - P_d(\boldsymbol{z})} - \frac{\mathbb{E}_{\widehat{P}_{d*}}[\, Y \mid \boldsymbol{c}, \boldsymbol{z}\,]\widehat{P}_{d*}(\boldsymbol{c}, \boldsymbol{z}) + 1 - \widehat{P}_{d*}(\boldsymbol{z})}{\widehat{P}_{d*}(\boldsymbol{c}, \boldsymbol{z}) + 1 - \widehat{P}_{d*}(\boldsymbol{z})} \right\}. \tag{101}$$

This shows that for a given $\boldsymbol{C} = \boldsymbol{c}$, the $\min_{\widehat{\mathcal{M}}\in\mathbb{M}}, ( \Delta_{d>d*} ) > 0$ for some $d \neq d^*$ if and only if,

$$\min_{\widehat{P}:\, \psi(\widehat{P},P)\leqslant\delta} \left\{ \frac{\mathbb{E}_{\widehat{P}_d}[\, Y \mid \boldsymbol{c}, \boldsymbol{z}\,]\widehat{P}_d(\boldsymbol{c}, \boldsymbol{z})}{P_d(\boldsymbol{c}, \boldsymbol{z}) + 1 - P_d(\boldsymbol{z})} - \frac{\mathbb{E}_{\widehat{P}_{d*}}[\, Y \mid \boldsymbol{c}, \boldsymbol{z}\,]\widehat{P}_{d*}(\boldsymbol{c}, \boldsymbol{z}) + 1 - \widehat{P}_{d*}(\boldsymbol{z})}{\widehat{P}_{d*}(\boldsymbol{c}, \boldsymbol{z}) + 1 - \widehat{P}_{d*}(\boldsymbol{z})} \right\} > 0. \tag{102}$$

$\square$

**Thm. 2 restated**. Let $\sigma := do(\boldsymbol{z})$ be a shift on a set of variables $\boldsymbol{Z} \subset \boldsymbol{V}$. For $\boldsymbol{R}_i \subset \boldsymbol{Z} \subset \boldsymbol{V}, i = 1, \ldots, k$, consider an AI grounded in multiple domains $\{\mathcal{M}_{\boldsymbol{r}_i} : i = 1, \ldots, k\}$. The AI is weakly predictable in a context $\boldsymbol{C} = \boldsymbol{c}$ under a shift $\sigma := do(\boldsymbol{z})$ if and only if there exists a decision $d^*$ such that,

$$\max_{i,j=1,\ldots,k} A(\boldsymbol{r}_i, \boldsymbol{r}_j) > 0, \quad \text{for some } d \neq d^*, \tag{103}$$

where

$$A(\boldsymbol{r}_i, \boldsymbol{r}_j) := \frac{\mathbb{E}_{P_{d,\boldsymbol{r}_i}}[\, Y \mid \boldsymbol{c}, \boldsymbol{z}\backslash\boldsymbol{r}_i\,]P_{d,\boldsymbol{r}_i}(\boldsymbol{c}, \boldsymbol{z}\backslash\boldsymbol{r}_i)}{P_{d,\boldsymbol{r}_i}(\boldsymbol{c}, \boldsymbol{z}\backslash\boldsymbol{r}_i) + 1 - P_{d,\boldsymbol{r}_i}(\boldsymbol{z}\backslash\boldsymbol{r}_i)} - \frac{\mathbb{E}_{P_{d*,\boldsymbol{r}_j}}[\, Y \mid \boldsymbol{c}, \boldsymbol{z}\backslash\boldsymbol{r}_j\,]P_{d*,\boldsymbol{r}_j}(\boldsymbol{c}, \boldsymbol{z}\backslash\boldsymbol{r}_j) + 1 - P_{d*,\boldsymbol{r}_j}(\boldsymbol{z}\backslash\boldsymbol{r}_j)}{P_{d*,\boldsymbol{r}_j}(\boldsymbol{c}, \boldsymbol{z}\backslash\boldsymbol{r}_j) + 1 - P_{d*,\boldsymbol{r}_j}(\boldsymbol{z}\backslash\boldsymbol{r}_j)}.$$

*Proof.* $\{\mathcal{M}_{\boldsymbol{r}_i} : i = 1, \ldots, k\}$ describes $k$ domains in which experiments on different subsets of $\boldsymbol{Z}$ have been conducted. This includes possibly the null experiment $\boldsymbol{R}_i = \varnothing$ that refers to the unaltered domain $\mathcal{M}$.

We can use a similar derivation to that of Thm. 1 to derive bounds on $\Delta$ under a shift $\sigma := do(\boldsymbol{z})$ in terms of $P_{d,\boldsymbol{r}}(\boldsymbol{V}), \boldsymbol{R} \in \boldsymbol{V}$ and obtain,

$$\Delta_{d_1>d_0} \geqslant A(\boldsymbol{r}) \tag{104}$$

where,

$$A(\boldsymbol{r}) := \frac{\mathbb{E}_{P_{d_1,\boldsymbol{r}}}[\, Y \mid \boldsymbol{c}, \boldsymbol{z}\backslash\boldsymbol{r}\,]P_{d_1,\boldsymbol{r}}(\boldsymbol{c}, \boldsymbol{z}\backslash\boldsymbol{r})}{P_{d_1,\boldsymbol{r}}(\boldsymbol{c}, \boldsymbol{z}\backslash\boldsymbol{r}) + 1 - P_{d_1,\boldsymbol{r}}(\boldsymbol{z}\backslash\boldsymbol{r})} - \frac{\mathbb{E}_{P_{d_0,\boldsymbol{r}}}[\, Y \mid \boldsymbol{c}, \boldsymbol{z}\backslash\boldsymbol{r}\,]P_{d_0,\boldsymbol{r}}(\boldsymbol{c}, \boldsymbol{z}\backslash\boldsymbol{r}) + 1 - P_{d_0,\boldsymbol{r}}(\boldsymbol{z}\backslash\boldsymbol{r})}{P_{d_0,\boldsymbol{r}}(\boldsymbol{c}, \boldsymbol{z}\backslash\boldsymbol{r}) + 1 - P_{d_0,\boldsymbol{r}}(\boldsymbol{z}\backslash\boldsymbol{r})}. \tag{105}$$

These bounds can be shown to be tight by constructing similar SCMs. For example, for the analytical lower bound consider,

$$\mathcal{M}^1_{d,\boldsymbol{r}} =: \begin{cases} \boldsymbol{S} \leftarrow f_{\boldsymbol{S}}(\boldsymbol{u}) \\ \boldsymbol{R} \leftarrow \boldsymbol{r} \\ \boldsymbol{C} \leftarrow \begin{cases} f_{\boldsymbol{C}}(\boldsymbol{u}, \boldsymbol{s}, \boldsymbol{r}) \text{ if } f_{\boldsymbol{S}}(\boldsymbol{u}) = \boldsymbol{s} \\ 1 \text{ otherwise.} \end{cases} \\ D \leftarrow d \\ Y \leftarrow \begin{cases} f_Y(d, \boldsymbol{c}, \boldsymbol{s}, \boldsymbol{r}, \boldsymbol{u}) \text{ if } f_{\boldsymbol{S}}(\boldsymbol{u}) = \boldsymbol{s} \\ 1 \text{ if } f_{\boldsymbol{S}}(\boldsymbol{u}) \neq \boldsymbol{s}, d = d_0 \\ 0 \text{ if } f_{\boldsymbol{S}}(\boldsymbol{u}) \neq \boldsymbol{s}, d = d_1 \end{cases} \\ P(\boldsymbol{U}) \end{cases} \tag{106}$$

where $S = Z \backslash R$. Here $\{f_Z, f_C, f_Y, \mathcal{U}, P(U)\}$ are chosen to match the observed trajectory of agent interactions, i.e., such that $P^{\mathcal{M}^1_{d,r}}(v) = P^{\widehat{\mathcal{M}}_{d,r}}(v)$ for all $v \in \text{supp}_V$. We could verify that this SCM evaluates to the lower bound above.

If we have multiple domains with different set of intervened variables $\{R_i : i = 1, \ldots, k\}$ we could use this construction to find a lower using samples from $\{P_{d,r_i}(V) : i = 1, \ldots, k\}$. A lower bound that can be constructed for an AI system grounded in $\{\mathcal{M}_{r_i} : i = 1, \ldots, k\}$ is,

$$\Delta_{d_1 > d_0} \geqslant \max_{i,j=1,\ldots,k} A(r_i, r_j) \tag{107}$$

where

$$A(r_i, r_j) := \frac{\mathbb{E}_{P_{d_1,r_i}}[\, Y \mid c, z \backslash r_i \,] P_{d_1,r_i}(c, z \backslash r_i)}{P_{d_1,r_i}(c, z \backslash r_i) + 1 - P_{d_1,r_i}(z \backslash r_i)} - \frac{\mathbb{E}_{P_{d_0,r_j}}[\, Y \mid c, z \backslash r_j \,] P_{d_0,r_j}(c, z \backslash r_j) + 1 - P_{d_0,r_j}(z \backslash r_j)}{P_{d_0,r_j}(c, z \backslash r_j) + 1 - P_{d_0,r_j}(z \backslash r_j)}. \tag{108}$$

The intuition here is that we have multiple lower bounds for the preference gap, then the best lower bound can be taken to be the largest of the multiple lower bounds available.

We can show that this bound is tight in the case where the AI is grounded in two environments $\{\mathcal{M}_{r_1}, \mathcal{M}_{r_2}\}$ under a shift $\sigma := do(z), Z = R_1 \cup R_2$. According to the inequality above, we have simultaneously,

$$\Delta_{d_1 > d_0} \geqslant A(r_1, r_1), A(r_1, r_2), A(r_2, r_1), A(r_2, r_2). \tag{109}$$

Each of these terms can be evaluated from the available data sampled from $\{P_{d,r_1}, P_{d,r_2}\}$. Note that both $A(r_1, r_1)$ and $A(r_2, r_2)$ can be obtained with the SCM above. Without loss of generality, assume that $A(r_1, r_2) \geqslant A(r_2, r_1), A(r_1, r_1), A(r_2, r_2)$. We will show that we can construct an SCM compatible with $\{P_{d,r_1}, P_{d,r_2}\}$ that evaluates to $A(r_1, r_2)$ demonstrating that the bound is tight.

Consider the following SCM:

$$\mathcal{M}_d =: \begin{cases} R_1 \leftarrow f_{R_1}(u_1) \\ R_2 \leftarrow f_{R_2}(u_2) \\ C \leftarrow \begin{cases} f_C(r_1, r_2, u_1, u_2) \text{ if } f_{R_1}(u_1) = r_1, f_{R_2}(u_2) = r_2 \\ 1 \text{ otherwise.} \end{cases} \\ D \leftarrow d \\ Y \leftarrow \begin{cases} f_Y(d, c, r_1, r_2, u_1, u_2) \text{ if } f_{R_1}(u_1) = r_1, f_{R_2}(u_2) = r_2 \\ f_Y(d, c, r_1, r_2, u_1, u_2) \text{ if } d = d_1, f_{R_1}(u_1) \neq r_1, f_{R_2}(u_2) = r_2 \\ f_Y(d, c, r_1, r_2, u_1, u_2) \text{ if } d = d_0, f_{R_1}(u_1) = r_1, f_{R_2}(u_2) \neq r_2 \\ 0 \text{ if } d = d_1, f_{R_1}(u_1) = r_1, f_{R_2}(u_2) \neq r_2 \\ 0 \text{ if } d = d_1, f_{R_1}(u_1) \neq r_1, f_{R_2}(u_2) \neq r_2 \\ 1 \text{ if } d = d_0, f_{R_1}(u_1) \neq r_1, f_{R_2}(u_2) = r_2 \\ 1 \text{ if } d = d_0, f_{R_1}(u_1) \neq r_1, f_{R_2}(u_2) \neq r_2 \end{cases} \\ P(U) \end{cases} \tag{110}$$

Notice that in $\mathcal{M}_d$ different choices of functional assignments "$f$" and $P(u)$ can generate any distribution $\{P_{d_1,r_1}, P_{d_0,r_2}\}$. That is this SCM (or a member of this family of SCMs) is compatible with the observed data.

Consider evaluating $A(r_1, r_2)$ under this SCM. Note that the derivations for the denominators are equivalent to those shown

in the proof of Thm. 1 so we will omit them here. The first term in the numerator,

$$\mathbb{E}_{P^{\mathcal{M}_{d_1,r_1,r_2}}}[\, Y \mathbb{1}_{\boldsymbol{c}}(\boldsymbol{C})\,] \tag{111}$$

$$= \sum_{\boldsymbol{u}_2} \mathbb{E}_{P^{\mathcal{M}_{d_1,r_1,r_2}}}[\, Y \mathbb{1}_{\boldsymbol{c}}(\boldsymbol{C}) \mid \boldsymbol{u}_2\,] P^{\mathcal{M}_{d_1,r_1,r_2}}(\boldsymbol{u}_2) \tag{112}$$

$$= \sum_{\boldsymbol{u}_2} \mathbb{E}_{P^{\mathcal{M}_{d_1,r_1}}}[\, Y \mathbb{1}_{\boldsymbol{c}}(\boldsymbol{C}) \mid \boldsymbol{r}_2, \boldsymbol{u}_2\,] P^{\mathcal{M}_{d_1,r_1}}(\boldsymbol{u}_2) \tag{113}$$

$$= \mathbb{E}_{P^{\mathcal{M}_{d_1,r_1}}}[\, Y \mathbb{1}_{\boldsymbol{c}}(\boldsymbol{C}) \mid \boldsymbol{r}_2, \{\boldsymbol{u} : f_{\boldsymbol{R}_2}(\boldsymbol{u}_2) = \boldsymbol{r}_2\}\,] P^{\mathcal{M}_{d_1,r_1}}(\{\boldsymbol{u}_2 : f_{\boldsymbol{R}_2}(\boldsymbol{u}_2) = \boldsymbol{r}_2\}) \tag{114}$$

$$+ \mathbb{E}_{P^{\mathcal{M}_{d_1,r_1}}}[\, Y \mathbb{1}_{\boldsymbol{c}}(\boldsymbol{C}) \mid \boldsymbol{r}_2, \{\boldsymbol{u} : f_{\boldsymbol{R}_2}(\boldsymbol{u}_2) \neq \boldsymbol{r}_2\}\,] P^{\mathcal{M}_{d_1,r_1}}(\{\boldsymbol{u}_2 : f_{\boldsymbol{R}_2}(\boldsymbol{u}_2) \neq \boldsymbol{r}_2\}) \tag{115}$$

$$= \mathbb{E}_{P^{\mathcal{M}_{d_1,r_1}}}[\, Y \mathbb{1}_{\boldsymbol{c}}(\boldsymbol{C}) \mid \boldsymbol{r}_2\,] P^{\mathcal{M}_{d_1,r_1}}(\boldsymbol{r}_2) \tag{116}$$

$$= \mathbb{E}_{P^{\mathcal{M}_{d_1,r_1}}}[\, Y \mid \boldsymbol{c}, \boldsymbol{r}_2\,] P^{\mathcal{M}_{d_1,r_1}}(\boldsymbol{c}, \boldsymbol{r}_2) \tag{117}$$

The second term in the numerator is,

$$\mathbb{E}_{P^{\mathcal{M}_{d_0,r_1,r_2}}}[\, Y \mathbb{1}_{\boldsymbol{c}}(\boldsymbol{C})\,] \tag{118}$$

$$= \sum_{\boldsymbol{u}_1} \mathbb{E}_{P^{\mathcal{M}_{d_0,r_1,r_2}}}[\, Y \mathbb{1}_{\boldsymbol{c}}(\boldsymbol{C}) \mid \boldsymbol{u}_1\,] P^{\mathcal{M}_{d_0,r_1,r_2}}(\boldsymbol{u}_1) \tag{119}$$

$$= \sum_{\boldsymbol{u}_1} \mathbb{E}_{P^{\mathcal{M}_{d_0,r_2}}}[\, Y \mathbb{1}_{\boldsymbol{c}}(\boldsymbol{C}) \mid \boldsymbol{r}_1, \boldsymbol{u}_1\,] P^{\mathcal{M}_{d_0,r_2}}(\boldsymbol{u}_1) \tag{120}$$

$$= \mathbb{E}_{P^{\mathcal{M}_{d_0,r_2}}}[\, Y \mathbb{1}_{\boldsymbol{c}}(\boldsymbol{C}) \mid \boldsymbol{r}_1, \{\boldsymbol{u} : f_{\boldsymbol{R}_1}(\boldsymbol{u}_1) = \boldsymbol{r}_1\}\,] P^{\mathcal{M}_{d_0,r_2}}(\{\boldsymbol{u}_1 : f_{\boldsymbol{R}_1}(\boldsymbol{u}_1) = \boldsymbol{r}_1\}) \tag{121}$$

$$+ \mathbb{E}_{P^{\mathcal{M}_{d_0,r_2}}}[\, Y \mathbb{1}_{\boldsymbol{c}}(\boldsymbol{C}) \mid \boldsymbol{r}_1, \{\boldsymbol{u} : f_{\boldsymbol{R}_1}(\boldsymbol{u}_1) \neq \boldsymbol{r}_1\}\,] P^{\mathcal{M}_{d_0,r_2}}(\{\boldsymbol{u}_1 : f_{\boldsymbol{R}_1}(\boldsymbol{u}_1) \neq \boldsymbol{r}_1\}) \tag{122}$$

$$= \mathbb{E}_{P^{\mathcal{M}_{d_0,r_2}}}[\, Y \mathbb{1}_{\boldsymbol{c}}(\boldsymbol{C}) \mid \boldsymbol{r}_1\,] P^{\mathcal{M}_{d_0,r_2}}(\boldsymbol{r}_1) + 1 - P^{\mathcal{M}_{d_0,r_2}}(\boldsymbol{r}_1) \tag{123}$$

$$= \mathbb{E}_{P^{\mathcal{M}_{d_0,r_2}}}[\, Y \mid \boldsymbol{c}, \boldsymbol{r}_1\,] P^{\mathcal{M}_{d_0,r_2}}(\boldsymbol{c}, \boldsymbol{r}_1) + 1 - P^{\mathcal{M}_{d_0,r_2}}(\boldsymbol{r}_1) \tag{124}$$

Combining these results we get that under $\mathcal{M}$,

$$\Delta_{d_1 > d_0} = A(\boldsymbol{r}_1, \boldsymbol{r}_2). \tag{125}$$

$\square$

**Corollary 3.** *The bound from multiple domains in Thm. 2 will be at least as informative as the bound from a single domain in Thm. 1.*

*Proof.* We claim here that for any $\boldsymbol{R} \subset \boldsymbol{Z}$,

$$A(\varnothing) \leqslant A(\boldsymbol{r}) \tag{126}$$

This means that the bounds on $\Delta$ that we can obtain from an AI system grounded in $\mathcal{M}_{\boldsymbol{r}}$ are more informative than the bounds obtained from an AI system grounded in $\mathcal{M}$. $A$ is a difference of two terms written $A(\boldsymbol{r}) = A_1(\boldsymbol{r}) - A_2(\boldsymbol{r})$.

$$A_1(\boldsymbol{r}) := \frac{\mathbb{E}_{P_{d_1,r}}[\, Y \mid \boldsymbol{c}, \boldsymbol{z}\backslash\boldsymbol{r}\,] P_{d_1,r}(\boldsymbol{c}, \boldsymbol{z}\backslash\boldsymbol{r})}{P_{d_1,r}(\boldsymbol{c}, \boldsymbol{z}\backslash\boldsymbol{r}) + 1 - P_{d_1,r}(\boldsymbol{z}\backslash\boldsymbol{r})} \tag{127}$$

$$A_2(\boldsymbol{r}) := \frac{\mathbb{E}_{P_{d_0,r}}[\, Y \mid \boldsymbol{c}, \boldsymbol{z}\backslash\boldsymbol{r}\,] P_{d_0,r}(\boldsymbol{c}, \boldsymbol{z}\backslash\boldsymbol{r}) + 1 - P_{d_0,r}(\boldsymbol{z}\backslash\boldsymbol{r})}{P_{d_0,r}(\boldsymbol{c}, \boldsymbol{z}\backslash\boldsymbol{r}) + 1 - P_{d_0,r}(\boldsymbol{z}\backslash\boldsymbol{r})}. \tag{128}$$

It holds that $A_1(\boldsymbol{r}) \geqslant A_1(\varnothing), A_2(\boldsymbol{r}) \leqslant A_2(\varnothing)$ which then implies $A(\boldsymbol{r}) \geqslant A(\varnothing)$. To see this notice that,

$$A_1(\boldsymbol{r}) := \frac{\mathbb{E}_{P_{d_1,r}}[\, Y \mid \boldsymbol{c}, \boldsymbol{z}\backslash\boldsymbol{r}\,] P_{d_1,r}(\boldsymbol{c}, \boldsymbol{z}\backslash\boldsymbol{r})}{P_{d_1,r}(\boldsymbol{c}, \boldsymbol{z}\backslash\boldsymbol{r}) + 1 - P_{d_1,r}(\boldsymbol{z}\backslash\boldsymbol{r})} \tag{129}$$

$$\geqslant \frac{\mathbb{E}_{P_{d_1}}[\, Y \mid \boldsymbol{c}, \boldsymbol{z}\,] P_{d_1}(\boldsymbol{c}, \boldsymbol{z})}{P_{d_1,r_i}(\boldsymbol{c}, \boldsymbol{z}\backslash\boldsymbol{r}) + 1 - P_{d_1,r}(\boldsymbol{z}\backslash\boldsymbol{r})} \tag{130}$$

$$= \frac{\mathbb{E}_{P_{d_1}}[\, Y \mid \boldsymbol{c}, \boldsymbol{z}\,] P_{d_1}(\boldsymbol{c}, \boldsymbol{z})}{1 - P_{d_1,r}(\tilde{\boldsymbol{c}}, \boldsymbol{z}\backslash\boldsymbol{r})} \tag{131}$$

$$\geqslant \frac{\mathbb{E}_{P_{d_1}}[\, Y \mid \boldsymbol{c}, \boldsymbol{z}\,] P_{d_1}(\boldsymbol{c}, \boldsymbol{z})}{1 - P_{d_1}(\tilde{\boldsymbol{c}}, \boldsymbol{z})} \tag{132}$$

$$= \frac{\mathbb{E}_{P_{d_1}}[\, Y \mid \boldsymbol{c}, \boldsymbol{z}\,] P_{d_1}(\boldsymbol{c}, \boldsymbol{z})}{P_{d_1}(\boldsymbol{c}, \boldsymbol{z}) + 1 - P_{d_1}(\boldsymbol{z})} \tag{133}$$

$$= A_1(\varnothing), \tag{134}$$

where $\tilde{\boldsymbol{c}}$ stands for the combination of values of $\boldsymbol{C}$ that are not $\boldsymbol{c}$. Further,

$$A_2(\boldsymbol{r}) := \frac{\mathbb{E}_{P_{d_0,r}}[\, Y \mid \boldsymbol{c}, \boldsymbol{z}\backslash\boldsymbol{r}\,] P_{d_0,r}(\boldsymbol{c}, \boldsymbol{z}\backslash\boldsymbol{r}) + 1 - P_{d_0,r}(\boldsymbol{z}\backslash\boldsymbol{r})}{P_{d_0,r}(\boldsymbol{c}, \boldsymbol{z}\backslash\boldsymbol{r}) + 1 - P_{d_0,r}(\boldsymbol{z}\backslash\boldsymbol{r})} \tag{135}$$

$$= 1 - \frac{\mathbb{E}_{P_{d_0,r}}[\, 1 - Y \mid \boldsymbol{c}, \boldsymbol{z}\backslash\boldsymbol{r}\,] P_{d_0,r}(\boldsymbol{c}, \boldsymbol{z}\backslash\boldsymbol{r})}{P_{d_0,r}(\boldsymbol{c}, \boldsymbol{z}\backslash\boldsymbol{r}) + 1 - P_{d_0,r}(\boldsymbol{z}\backslash\boldsymbol{r})} \tag{136}$$

$$\leqslant 1 - \frac{\mathbb{E}_{P_{d_0}}[\, 1 - Y \mid \boldsymbol{c}, \boldsymbol{z}\,] P_{d_0}(\boldsymbol{c}, \boldsymbol{z})}{P_{d_0,r}(\boldsymbol{c}, \boldsymbol{z}\backslash\boldsymbol{r}) + 1 - P_{d_0,r}(\boldsymbol{z}\backslash\boldsymbol{r})} \tag{137}$$

$$\leqslant 1 - \frac{\mathbb{E}_{P_{d_0}}[\, 1 - Y \mid \boldsymbol{c}, \boldsymbol{z}\,] P_{d_0}(\boldsymbol{c}, \boldsymbol{z})}{P_{d_0}(\boldsymbol{c}, \boldsymbol{z}) + 1 - P_{d_0}(\boldsymbol{z})} \tag{138}$$

$$= \frac{\mathbb{E}_{P_{d_0}}[\, Y \mid \boldsymbol{c}, \boldsymbol{z}\,] P_{d_0}(\boldsymbol{c}, \boldsymbol{z}) + 1 - P_{d_0}(\boldsymbol{z})}{P_{d_0}(\boldsymbol{c}, \boldsymbol{z}) + 1 - P_{d_0}(\boldsymbol{z})} \tag{139}$$

$$= A_2(\varnothing). \tag{140}$$

$\square$

**Thm. 3 restated.** Consider an AI grounded in a domain $\mathcal{M}$ made aware of an (under-specified) shift on non-empty $\boldsymbol{Z} \subset \boldsymbol{V}$. Then the AI is provably not weakly (or strongly) predictable in any context $\boldsymbol{C} = \boldsymbol{c}$.

*Proof.* Recall that the preference gap is defined as:

$$\Delta_{d_1 \succ d_0} := \mathbb{E}_{\hat{P}_{\sigma,d_1}}[\, Y \mid \boldsymbol{C} = \boldsymbol{c}\,] - \mathbb{E}_{\hat{P}_{\sigma,d_0}}[\, Y \mid \boldsymbol{C} = \boldsymbol{c}\,] \tag{141}$$

Here we know that $\sigma$ potentially modifies the mechanisms of the set of variables $\boldsymbol{Z}$ though the nature of the modification is unknown. In the worst-case, the AI's interpretation of the possible new assignment of $\boldsymbol{Z}$ could be arbitrary.

We will prove this theorem for the case of binary variables $Y, Z \in \boldsymbol{V}$. In the following, we construct two (canonical) models that entail any chosen distribution for the observed data $P_d(y, z \mid \boldsymbol{c})$ but evaluate to the a priori minimum and maximum value of the preference gap $\Delta$, i.e. $-1$ and $1$ respectively. We make use of the canonical model construction from (Jalaldoust et al., 2024) to define the following general SCM,

$$Z \leftarrow \begin{cases} 0 \text{ if } r_z = 0 \\ 1 \text{ if } r_z = 1 \end{cases} \quad, \quad Y \leftarrow \begin{cases} 0 \text{ if } r_y = 0 \\ 0 \text{ if } r_y = 1, z = 0 \\ 1 \text{ if } r_y = 1, z = 1 \\ 1 \text{ if } r_y = 2, z = 0 \\ 0 \text{ if } r_y = 2, z = 1 \\ 1 \text{ if } r_y = 3 \end{cases} \tag{142}$$

$U = \{R_z, R_y\}$ where $R_z$ and $R_y$ might be correlated and with a probability $\widehat{P}(U) = \widehat{P}_d(U \mid c)$ such that $\widehat{P}_d(z, y \mid c) = \widehat{P}(z, y \mid c)$. By (Jalaldoust et al., 2024, Thm. 1) this is always possible since this class of canonical models is sufficiently expressive to model any observational or interventional distribution. We can visualise the joint probability of exogenous variables using the following table:

| Probabilities $\widehat{\mathcal{M}}$ | $r_z = 0$ | $r_z = 1$ |
|---|---|---|
| $r_y = 0$ | $p_{00}$ | $p_{10}$ |
| $r_y = 1$ | $p_{01}$ | $p_{11}$ |
| $r_y = 2$ | $p_{02}$ | $p_{12}$ |
| $r_y = 3$ | $p_{03}$ | $p_{13}$ |

where we have written $P_d(r_z = a, r_y = b \mid c) = p_{ab}$. From these we could compute joint probabilities

$$P_d(z = 0, y = 0 \mid c) = p_{00} + p_{01}, \tag{143}$$
$$P_d(z = 0, y = 1 \mid c) = p_{02} + p_{03}, \tag{144}$$
$$P_d(z = 1, y = 0 \mid c) = p_{12} + p_{11}, \tag{145}$$
$$P_d(z = 1, y = 1 \mid c) = p_{11} + p_{13} \tag{146}$$

Here we can see that the parameter space $P_d(r_z, r_y \mid c)$ is very expressive. For example, without loss of generality we could set $p_{03} = p_{13} = 0$ or $p_{00} = p_{10} = 0$ and still be able to generate any observed distribution $P_d(z, y \mid c)$.

The given shift in the environment $\sigma$ can be entirely modelled as a shift in $P_{\sigma,d}(r_z \mid c)$ while keeping the probability of $r_y$ invariant, i.e., $P_{\sigma,d}(r_y \mid c) = P_d(r_y \mid c)$. In other words, given the table above, we can change each of the cells while maintaining the row sums equal. Recall that we are interested in evaluating bounds on a probability of the form $P_{\sigma,d}(y = 1 \mid c)$ and $P_{\sigma,d}(y = 1 \mid z = 1, c)$ depending on whether $Z$ is given as an input to the AI or not. Both these quantities can be written in terms of the probabilities of exogenous variables as follows,

$$P_{\sigma,d}(y = 1 \mid c) = p_{02} + p_{03} + p_{11} + p_{13} \tag{147}$$
$$P_{\sigma,d}(y = 1 \mid z = 1, c) = \frac{p_{11} + p_{13}}{p_{11} + p_{13} + p_{12} + p_{11}}. \tag{148}$$

For the lower bound on these quantities, without loss of generality assume that $p_{03} = p_{13} = 0$. Then the following table:

| Probabilities $\widehat{\mathcal{M}}_\sigma$ | $r_z = 0$ | $r_z = 1$ |
|---|---|---|
| $r_y = 0$ | $p_{00}$ | $p_{10}$ |
| $r_y = 1$ | $p_{01} + p_{11}$ | $0$ |
| $r_y = 2$ | $0$ | $p_{12} + p_{02}$ |
| $r_y = 3$ | $0$ | $0$ |

is a perfectly valid model under a shift $\sigma$ that respects the constraint on $P_{\sigma,d}(r_y \mid c) = P_d(r_y \mid c)$ but for which $P_{\sigma,d}(y = 1 \mid c) = 0$ as it is the sum of the 4 zero entries and $P_{\sigma,d}(y = 1 \mid z = 1, c) = 0$ as it is the sum of the two 0 entries in the second column divided by the sum of entries in the second column.

If we are interested in getting an upper bound then without loss of generality assume that $p_{00} = p_{10} = 0$. Then the following

| Probabilities $\widehat{\mathcal{M}}_\sigma$ | $r_z = 0$ | $r_z = 1$ |
|---|---|---|
| $r_y = 0$ | $0$ | $0$ |
| $r_y = 1$ | $0$ | $p_{01} + p_{11}$ |
| $r_y = 2$ | $p_{12} + p_{02}$ | $0$ |
| $r_y = 3$ | $p_{03}$ | $p_{13}$ |

is a perfectly valid model under a shift $\sigma$ that respects the constraint on $P_{\sigma,d}(r_y \mid c) = P_d(r_y \mid c)$ but for which $P_{\sigma,d}(y = 1 \mid c) = 1$ as it is the sum of the 4 non-zero entries and $P_{\sigma,d}(y = 1 \mid z = 1, c) = 1$ as it is the sum of the two non-zero entries in the second column divided by the sum of entries in the second column.

By using this construction to define lower and upper bounds for $P_{\sigma,d}(y=1 \mid \boldsymbol{c})$ or $P_{\sigma,d}(y=1 \mid \boldsymbol{z},\boldsymbol{c})$ for $d=d_0, d_1$ we obtain a possible internal model for the AI that entails the observed external behaviour but for which the preference gap evaluates to $-1$ and $1$. This means that the a priori bound,

$$-1 \leqslant \Delta_{d>d*} \leqslant 1, \tag{149}$$

is tight whenever the shift is undefined (whether we know the variables it applies to or not). Since the preference gap is unconstrained for any $\boldsymbol{C}=\boldsymbol{c}$ and any pair of decisions $(d, d*)$, the AI is not predictable. $\qquad\square$

**Thm. 4 restated.** Consider an AI grounded in a domain $\mathcal{M}$ and $P_{\sigma,d}(\boldsymbol{C})$ made aware of a shift $\sigma$ on $\boldsymbol{Z} \subset \boldsymbol{C}$. The AI is weakly predictable under this shift in a context $\boldsymbol{C}=\boldsymbol{c}$ if there exists a decision $d*$ such that,

$$1 - \frac{2 + \mathbb{E}_{P_{d*}}[\,Y \mid \boldsymbol{c}\,]P_{d*}(\boldsymbol{c}) - \mathbb{E}_{P_d}[\,Y \mid \boldsymbol{c}\,]P_d(\boldsymbol{c}) - 2P_d(\boldsymbol{z}) + P_d(\boldsymbol{c})}{P_{\sigma,d*}(\boldsymbol{c})} > 0, \quad \text{for some } d \neq d*. \tag{150}$$

*Proof.* Recall that the preference gap under a shift $\sigma$ between decisions $(d_1, d_0)$ in a situation $\boldsymbol{C}=\boldsymbol{c}$ is defined as:

$$\Delta_{d_1 > d_0} := \mathbb{E}_{\widehat{P}_{\sigma,d_1}}[\,Y \mid \boldsymbol{C}=\boldsymbol{c}\,] - \mathbb{E}_{\widehat{P}_{\sigma,d_0}}[\,Y \mid \boldsymbol{C}=\boldsymbol{c}\,] \tag{151}$$

Here we know that $\sigma$ potentially modifies the mechanisms of the set of variables $\boldsymbol{Z}$. The nature of the modification is unknown but we are told that after modification, the expected probability of $\boldsymbol{C}$ is given by $P_{\sigma,d}(\boldsymbol{C})$, assumed to be known and internalised by the A. This means that its internal model, whatever interpretation for the shift it chooses, generates the assumed probabilities, i.e. $\widehat{P}_{\sigma,d}(\boldsymbol{C}) = P_{\sigma,d}(\boldsymbol{C})$.

We will consider the derivation of bounds on each term of this difference separately. Firstly, note that,

$$\mathbb{E}_{\widehat{P}_{\sigma,d}}[\,Y \mid \boldsymbol{C}=\boldsymbol{c}\,] = \mathbb{E}_{\widehat{P}_{\sigma,d}}[\,Y\mathbb{1}_{\boldsymbol{c}}(\boldsymbol{C})\,] / \widehat{P}_{\sigma,d}(\boldsymbol{c}) \tag{152}$$

For ease of notation let us write $\boldsymbol{R} := \boldsymbol{C} \backslash \boldsymbol{Z}$. We could then show that,

$$\mathbb{E}_{\widehat{P}_{\sigma,d}}[\,Y\mathbb{1}_{\boldsymbol{z},\boldsymbol{r}}(\boldsymbol{Z},\boldsymbol{R})\,] = \mathbb{E}_{\widehat{P}_{\sigma,d}}[\,Y_{\boldsymbol{z}}\mathbb{1}_{\boldsymbol{z},\boldsymbol{r}}(\boldsymbol{Z},\boldsymbol{R}_{\boldsymbol{z}})\,] \qquad\qquad \text{by consistency} \tag{153}$$

$$\leqslant \sum_{\boldsymbol{z}'}\mathbb{E}_{\widehat{P}_{\sigma,d}}[\,Y_{\boldsymbol{z}}\mathbb{1}_{\boldsymbol{z}',\boldsymbol{r}}(\boldsymbol{Z},\boldsymbol{R}_{\boldsymbol{z}})\,] \tag{154}$$

$$= \mathbb{E}_{\widehat{P}_{\sigma,d}}[\,Y_{\boldsymbol{z}}\mathbb{1}_{\boldsymbol{r}}(\boldsymbol{R}_{\boldsymbol{z}})\,] \qquad\qquad \text{marginalizing over the values } \boldsymbol{z}' \text{ of } \boldsymbol{Z} \tag{155}$$

Now once we intervene on $\boldsymbol{z}$ the mechanism that generate its value before hand, whether it was the shift $\sigma$ or something else is irrelevant. In essence, we get an equivalence between shifted an un-shifted distributions under intervention:

$$\mathbb{E}_{\widehat{P}_{\sigma,d}}[\,Y_{\boldsymbol{z}}\mathbb{1}_{\boldsymbol{r}}(\boldsymbol{R}_{\boldsymbol{z}})\,] = \mathbb{E}_{\widehat{P}_d}[\,Y_{\boldsymbol{z}}\mathbb{1}_{\boldsymbol{r}}(\boldsymbol{R}_{\boldsymbol{z}})\,] \tag{156}$$

We could now take this quantity to show the following,

$$\mathbb{E}_{\widehat{P}_d}[\,Y_{\boldsymbol{z}}\mathbb{1}_{\boldsymbol{r}}(\boldsymbol{R}_{\boldsymbol{z}})\,] = \sum_{\boldsymbol{z}'}\mathbb{E}_{\widehat{P}_d}[\,Y_{\boldsymbol{z}}\mathbb{1}_{\boldsymbol{z}',\boldsymbol{r}}(\boldsymbol{Z},\boldsymbol{R}_{\boldsymbol{z}})\,] \tag{157}$$

$$= \mathbb{E}_{\widehat{P}_d}[\,Y_{\boldsymbol{z}}\mathbb{1}_{\boldsymbol{z},\boldsymbol{r}}(\boldsymbol{Z},\boldsymbol{R}_{\boldsymbol{z}})\,] + \sum_{\boldsymbol{z}'\neq\boldsymbol{z}}\mathbb{E}_{\widehat{P}_d}[\,Y_{\boldsymbol{z}}\mathbb{1}_{\boldsymbol{z}',\boldsymbol{r}}(\boldsymbol{Z},\boldsymbol{R}_{\boldsymbol{z}})\,] \tag{158}$$

$$= \mathbb{E}_{\widehat{P}_d}[\,Y\mathbb{1}_{\boldsymbol{z},\boldsymbol{r}}(\boldsymbol{Z},\boldsymbol{R})\,] + \sum_{\boldsymbol{z}'\neq\boldsymbol{z}}\mathbb{E}_{\widehat{P}_d}[\,Y_{\boldsymbol{z}}\mathbb{1}_{\boldsymbol{z}',\boldsymbol{r}}(\boldsymbol{Z},\boldsymbol{R}_{\boldsymbol{z}})\,] \qquad \text{by consistency} \tag{159}$$

$$\leqslant \mathbb{E}_{\widehat{P}_d}[\,Y\mathbb{1}_{\boldsymbol{z},\boldsymbol{r}}(\boldsymbol{Z},\boldsymbol{R})\,] + \sum_{\boldsymbol{z}'\neq\boldsymbol{z}}\mathbb{E}_{\widehat{P}_d}[\,\mathbb{1}_{\boldsymbol{z}'}(\boldsymbol{Z})\,] \qquad \text{since } Y_{\boldsymbol{z}} \text{ and } \mathbb{1}_{\boldsymbol{r}}(\boldsymbol{R}_{\boldsymbol{z}}) \text{ are} \leqslant 1 \tag{160}$$

$$= \mathbb{E}_{\widehat{P}_d}[\,Y\mathbb{1}_{\boldsymbol{z},\boldsymbol{r}}(\boldsymbol{Z},\boldsymbol{R})\,] + 1 - \widehat{P}_d(\boldsymbol{z}) \tag{161}$$

$$= \mathbb{E}_{\widehat{P}_d}[\,Y \mid \boldsymbol{c}\,]\widehat{P}_d(\boldsymbol{c}) + 1 - \widehat{P}_d(\boldsymbol{z}) \tag{162}$$

For the lower bound we could consider the following derivation,

$$\mathbb{E}_{\widehat{P}_{\sigma,d}}[\,Y\mathbb{1}_{\boldsymbol{z},\boldsymbol{r}}(\boldsymbol{Z},\boldsymbol{R})\,] = \mathbb{E}_{\widehat{P}_{\sigma,d}}[\,\mathbb{1}_{\boldsymbol{z},\boldsymbol{r}}(\boldsymbol{Z},\boldsymbol{R})\,] - \mathbb{E}_{\widehat{P}_{\sigma,d}}[\,(1-Y)\mathbb{1}_{\boldsymbol{z},\boldsymbol{r}}(\boldsymbol{Z},\boldsymbol{R})\,]. \tag{163}$$

For ease of notation let us define,

$$\mathbb{E}_{\widehat{P}_{\sigma,d}}[\,\tilde{Y}\mathbb{1}_{\boldsymbol{z},\boldsymbol{r}}(\boldsymbol{Z},\boldsymbol{R})\,] := \mathbb{E}_{\widehat{P}_{\sigma,d}}[\,(1-Y)\mathbb{1}_{\boldsymbol{z},\boldsymbol{r}}(\boldsymbol{Z},\boldsymbol{R})\,]. \tag{164}$$

Similar bounds apply on $\mathbb{E}_{\widehat{P}_{\sigma,d}}[\,\tilde{Y}\mathbb{1}_{\boldsymbol{z},\boldsymbol{r}}(\boldsymbol{Z},\boldsymbol{R})\,]$ to get,

$$\mathbb{E}_{\widehat{P}_{\sigma,d}}[\,Y\mathbb{1}_{\boldsymbol{z},\boldsymbol{r}}(\boldsymbol{Z},\boldsymbol{R})\,] \geqslant \mathbb{E}_{\widehat{P}_{\sigma,d}}[\,\mathbb{1}_{\boldsymbol{z},\boldsymbol{r}}(\boldsymbol{Z},\boldsymbol{R})\,] - \{\mathbb{E}_{\widehat{P}_d}[\,\tilde{Y}\mathbb{1}_{\boldsymbol{z},\boldsymbol{r}}(\boldsymbol{Z},\boldsymbol{R})\,] + 1 - \widehat{P}_d(\boldsymbol{z})\} \tag{165}$$

$$= \mathbb{E}_{\widehat{P}_{\sigma,d}}[\,\mathbb{1}_{\boldsymbol{z},\boldsymbol{r}}(\boldsymbol{Z},\boldsymbol{R})\,] - \mathbb{E}_{\widehat{P}_d}[\,\mathbb{1}_{\boldsymbol{z},\boldsymbol{r}}(\boldsymbol{Z},\boldsymbol{R})\,] + \mathbb{E}_{\widehat{P}_d}[\,Y\mathbb{1}_{\boldsymbol{z},\boldsymbol{r}}(\boldsymbol{Z},\boldsymbol{R})\,] - 1 + \widehat{P}_d(\boldsymbol{z}) \tag{166}$$

$$= \widehat{P}_{\sigma,d}(\boldsymbol{c}) - \widehat{P}_d(\boldsymbol{c}) + \mathbb{E}_{\widehat{P}_d}[\,Y\mid\boldsymbol{c}\,]\widehat{P}_d(\boldsymbol{c}) - 1 + \widehat{P}_d(\boldsymbol{z}) \tag{167}$$

Putting the lower and upper bounds together to form bounds on $\Delta_{d_1 > d_0}$ we get,

$$\Delta_{d_1 > d_0} \geqslant \frac{\widehat{P}_{\sigma,d_1}(\boldsymbol{c}) - \widehat{P}_{d_1}(\boldsymbol{c}) + \mathbb{E}_{\widehat{P}_{d_1}}[\,Y\mid\boldsymbol{c}\,]\widehat{P}_{d_1}(\boldsymbol{c}) - 1 + \widehat{P}_{d_1}(\boldsymbol{z}) - \{\mathbb{E}_{\widehat{P}_{d_0}}[\,Y\mid\boldsymbol{c}\,]\widehat{P}_{d_0}(\boldsymbol{c}) + 1 - \widehat{P}_{d_0}(\boldsymbol{z})\}}{\widehat{P}_{\sigma,d_0}(\boldsymbol{c})} \tag{168}$$

$$= 1 + \frac{-\widehat{P}_{d_1}(\boldsymbol{c}) + \mathbb{E}_{\widehat{P}_{d_1}}[\,Y\mid\boldsymbol{c}\,]\widehat{P}_{d_1}(\boldsymbol{c}) - 1 + \widehat{P}_{d_1}(\boldsymbol{z}) - \mathbb{E}_{\widehat{P}_{d_0}}[\,Y\mid\boldsymbol{c}\,]\widehat{P}_{d_0}(\boldsymbol{c}) - 1 + \widehat{P}_{d_0}(\boldsymbol{z})\}}{\widehat{P}_{\sigma,d_0}(\boldsymbol{c})} \tag{169}$$

$$= 1 - \frac{2 + \mathbb{E}_{\widehat{P}_{d_0}}[\,Y\mid\boldsymbol{c}\,]\widehat{P}_{d_0}(\boldsymbol{c}) - \mathbb{E}_{\widehat{P}_{d_1}}[\,Y\mid\boldsymbol{c}\,]\widehat{P}_{d_1}(\boldsymbol{c}) - 2\widehat{P}_{d_1}(\boldsymbol{z}) + \widehat{P}_{d_1}(\boldsymbol{c})}{\widehat{P}_{\sigma,d_0}(\boldsymbol{c})} \tag{170}$$

and by grounding,

$$\Delta_{d_1 > d_0} \geqslant 1 - \frac{2 + \mathbb{E}_{P_{d_0}}[\,Y\mid\boldsymbol{c}\,]P_{d_0}(\boldsymbol{c}) - \mathbb{E}_{P_{d_1}}[\,Y\mid\boldsymbol{c}\,]P_{d_1}(\boldsymbol{c}) - 2P_{d_1}(\boldsymbol{z}) + P_{d_1}(\boldsymbol{c})}{P_{\sigma,d_0}(\boldsymbol{c})}. \tag{171}$$

This statement holds for any SCM compatible with the grounded AI's external behaviour and therefore,

$$\min_{\widehat{\mathcal{M}}\in\mathbb{M}}(\,\Delta_{d > d*}\,) \geqslant 1 - \frac{2 + \mathbb{E}_{P_{d*}}[\,Y\mid\boldsymbol{c}\,]P_{d*}(\boldsymbol{c}) - \mathbb{E}_{P_d}[\,Y\mid\boldsymbol{c}\,]P_d(\boldsymbol{c}) - 2P_d(\boldsymbol{z}) + P_d(\boldsymbol{c})}{P_{\sigma,d*}(\boldsymbol{c})}. \tag{172}$$

We can establish that the AI is weakly predictable in a context $\boldsymbol{C} = \boldsymbol{c}$ if there exists a decision $d*$ such that,

$$1 - \frac{2 + \mathbb{E}_{P_{d*}}[\,Y\mid\boldsymbol{c}\,]P_{d*}(\boldsymbol{c}) - \mathbb{E}_{P_d}[\,Y\mid\boldsymbol{c}\,]P_d(\boldsymbol{c}) - 2P_d(\boldsymbol{z}) + P_d(\boldsymbol{c})}{P_{\sigma,d*}(\boldsymbol{c})} > 0, \tag{173}$$

for some $d \neq d*$. $\qquad\square$

We now continue with our inference of the AI's perceived fairness and harm of decisions in Sec. 4.3.

**Thm. 5 restated.** Consider an agent with utility function $Y$ grounded in a domain $\mathcal{M}$. Then,

$$-\mathbb{E}_{P_d}[\,Y\mid z,\boldsymbol{c}] \leqslant \Upsilon(d,\boldsymbol{c}) \leqslant 1 - \mathbb{E}_{P_d}[\,Y\mid z,\boldsymbol{c}]. \tag{174}$$

This bound is tight.

*Proof.* Recall that for a given utility $Y$, the AI's counterfactual fairness gap relative to a decision $d$, in a given context $\boldsymbol{c}$, is

$$\Upsilon(d,\boldsymbol{c}) := \mathbb{E}_{\widehat{P}}[\,Y_{d,z_1}\mid z_0,\boldsymbol{c}\,] - \mathbb{E}_{\widehat{P}}[\,Y_d\mid z_0,\boldsymbol{c}\,]. \tag{175}$$

And remember that $Z \in \boldsymbol{C}$.

For ease of notation, write $z_1 = z, z_0 = z'$ such that,

$$\Upsilon(d,\boldsymbol{c}) := \mathbb{E}_{\widehat{P}}[\,Y_{d,z}\mid z',\boldsymbol{c}\,] - \mathbb{E}_{\widehat{P}}[\,Y_d\mid z',\boldsymbol{c}\,]. \tag{176}$$

We start by considering the following derivation:

$$\widehat{P}(y_{d,z} \mid \boldsymbol{c}) = \widehat{P}(y_{d,z}, z_d \mid \boldsymbol{c}) + \widehat{P}(y_{d,z}, z'_d \mid \boldsymbol{c}) \qquad \text{by marginalization} \qquad (177)$$

$$= \widehat{P}(y_d, z_d \mid \boldsymbol{c}) + \widehat{P}(y_{d,z}, z'_d \mid \boldsymbol{c}) \qquad \text{by consistency} \qquad (178)$$

and since $d$ does not affect $Z$ or $\boldsymbol{C}$, i.e. $Z_d = Z, \boldsymbol{C}_d = \boldsymbol{C}$,

$$\widehat{P}(y_{d,z} \mid \boldsymbol{c}) = \widehat{P}(y_d, z_d \mid \boldsymbol{c}) + \widehat{P}(y_{d,z}, z' \mid \boldsymbol{c}) \qquad (179)$$

which implies

$$\widehat{P}(y_{d,z} \mid z', \boldsymbol{c}) = \frac{\widehat{P}(y_{d,z} \mid \boldsymbol{c}) - \widehat{P}_d(y, z \mid \boldsymbol{c})}{\widehat{P}_d(z' \mid \boldsymbol{c})} \qquad (180)$$

Therefore,

$$\mathbb{E}_{\widehat{P}}\left[ Y_{d,z} \mid z', \boldsymbol{c} \right] = \frac{\mathbb{E}_{\widehat{P}}[Y_{d,z} \mid \boldsymbol{c}] - \mathbb{E}_{\widehat{P}_d}[ Y \mid z, \boldsymbol{c}]\widehat{P}_d(z \mid \boldsymbol{c})}{\widehat{P}_d(z' \mid \boldsymbol{c})}. \qquad (181)$$

All quantities on the r.h.s are observable except for $\mathbb{E}_{\widehat{P}}[Y_{d,z} \mid \boldsymbol{c}]$ which can be tightly bounded.

For the lower bound, consider the following derivation,

$$\mathbb{E}_{\widehat{P}}[Y_{d,z} \mid \boldsymbol{c}] = \sum_{\tilde{z}} \mathbb{E}_{\widehat{P}}\left[ Y_{d,z} \mathbb{1}_{\tilde{z}_d}(Z) \mid \boldsymbol{c} \right] \qquad \text{marginalizing over } z_d \qquad (182)$$

$$\geqslant \mathbb{E}_{\widehat{P}}\left[ Y_{d,z} \mathbb{1}_{z_d}(Z_d) \mid \boldsymbol{c}\right] \qquad \text{since summands} > 0 \qquad (183)$$

$$= \mathbb{E}_{\widehat{P}}\left[ Y_d \mathbb{1}_{z_d}(Z_d) \mid \boldsymbol{c}\right] \qquad \text{by consistency} \qquad (184)$$

$$= \mathbb{E}_{P_d}\left[ Y \mid \boldsymbol{c}, z \right]P_d(z \mid \boldsymbol{c}) \qquad \text{by grounding and } \boldsymbol{C}_d = \boldsymbol{C} \qquad (185)$$

Similarly, we can get an upper bound by noting

$$\mathbb{E}_{\widehat{P}}[Y_{d,z} \mid \boldsymbol{c}] = 1 - \mathbb{E}_{\widehat{P}}[(1 - Y_{d,z}) \mid \boldsymbol{c}] \qquad (186)$$

$$\leqslant \mathbb{E}_{\widehat{P}_d}\left[ Y \mid \boldsymbol{c}, z \right]\widehat{P}_d(z \mid \boldsymbol{c}) + \widehat{P}_d(z' \mid \boldsymbol{c}). \qquad (187)$$

**Tightness Lower Bound**    For the lower bound we will consider the following SCM,

$$\mathcal{M}_d^1 =: \begin{cases} Z \leftarrow f_Z(\boldsymbol{u}) \\ \boldsymbol{C} \leftarrow f_{\boldsymbol{C}}(\boldsymbol{u}) \\ D \leftarrow d \\ Y \leftarrow \begin{cases} f_Y(d, \boldsymbol{c}, z, \boldsymbol{u}) \text{ if } f_Z(\boldsymbol{u}) = z \\ 0 \text{ otherwise} \end{cases} \\ P(\boldsymbol{U}) \end{cases} \qquad (188)$$

Here $\{f_Z, f_{\boldsymbol{C}}, f_Y, \mathcal{U}, P(\boldsymbol{U})\}$ are chosen to match the observed trajectory of agent interactions, i.e., such that $P^{\mathcal{M}_d^1}(\boldsymbol{v}) = P^{\widehat{\mathcal{M}}_d}(\boldsymbol{v})$ for all $\boldsymbol{v} \in \text{supp}_V$.

Then, under $\mathcal{M}_d^1$,

$$\mathbb{E}_{P^{\mathcal{M}^1}}\left[ Y_{d,z} \mid \boldsymbol{c} \right] \qquad (189)$$

$$= \sum_{\boldsymbol{u}} \mathbb{E}_{P^{\mathcal{M}^1}}\left[ Y_{d,z} \mid \boldsymbol{u}, \boldsymbol{c} \right]P^{\mathcal{M}^1}(\boldsymbol{u} \mid \boldsymbol{c}) \qquad (190)$$

$$= \sum_{\boldsymbol{u}} \mathbb{E}_{P^{\mathcal{M}^1}}\left[ Y_d \mid z, \boldsymbol{u}, \boldsymbol{c} \right]P^{\mathcal{M}^1}(\boldsymbol{u} \mid \boldsymbol{c}) \qquad (191)$$

$$= \mathbb{E}_{P^{\mathcal{M}_d^1}}\left[ Y \mid z, \boldsymbol{c}, \{\boldsymbol{u} : f_Z(\boldsymbol{u}) = z\} \right]P^{\mathcal{M}^1}(\{\boldsymbol{u} : f_Z(\boldsymbol{u}) = z\} \mid \boldsymbol{c}) \qquad (192)$$

$$+ \mathbb{E}_{P^{\mathcal{M}_d^1}}\left[ Y \mid z, \boldsymbol{c}, \{\boldsymbol{u} : f_Z(\boldsymbol{u}) \neq z\} \right]P^{\mathcal{M}^1}(\{\boldsymbol{u} : f_Z(\boldsymbol{u}) \neq z\} \mid \boldsymbol{c}) \qquad (193)$$

$$= \mathbb{E}_{P^{\mathcal{M}_d^1}}\left[ Y \mid z, \boldsymbol{c} \right]P^{\mathcal{M}_d^1}(z \mid \boldsymbol{c}). \qquad (194)$$

This expression is the same one as the analytical bound showing that it is tight.

**Tightness Upper Bound**   For the upper bound we will consider the following SCM,

$$
\mathcal{M}_d^2 =: \begin{cases}
Z \leftarrow f_Z(\boldsymbol{u}) \\
\boldsymbol{C} \leftarrow f_{\boldsymbol{C}}(\boldsymbol{u}) \\
D \leftarrow d \\
Y \leftarrow \begin{cases} f_Y(d, \boldsymbol{c}, z, \boldsymbol{u}) \text{ if } f_Z(\boldsymbol{u}) = z \\ 1 \text{ otherwise} \end{cases} \\
P(\boldsymbol{U})
\end{cases}
\tag{195}
$$

Here $\{f_Z, f_{\boldsymbol{C}}, f_Y, \mathcal{U}, P(\boldsymbol{U})\}$ are chosen to match the observed trajectory of agent interactions, i.e., such that $P^{\mathcal{M}_d^2}(\boldsymbol{v}) = P^{\widehat{\mathcal{M}}_d}(\boldsymbol{v})$ for all $\boldsymbol{v} \in \text{supp}_V$.

Then, under $\mathcal{M}_d^2$,

$$
\mathbb{E}_{P^{\mathcal{M}^2}}[\, Y_{d,z} \mid \boldsymbol{c} \,]
\tag{196}
$$

$$
= \sum_{\boldsymbol{u}} \mathbb{E}_{P^{\mathcal{M}^2}}[\, Y_{d,z} \mid \boldsymbol{u}, \boldsymbol{c} \,] P^{\mathcal{M}^2}(\boldsymbol{u} \mid \boldsymbol{c})
\tag{197}
$$

$$
= \sum_{\boldsymbol{u}} \mathbb{E}_{P^{\mathcal{M}^2}}[\, Y_d \mid z, \boldsymbol{u}, \boldsymbol{c} \,] P^{\mathcal{M}^2}(\boldsymbol{u} \mid \boldsymbol{c})
\tag{198}
$$

$$
= \mathbb{E}_{P^{\mathcal{M}_d^2}}[\, Y \mid z, \boldsymbol{c}, \{\boldsymbol{u} : f_Z(\boldsymbol{u}) = z\} \,] P^{\mathcal{M}^2}(\{\boldsymbol{u} : f_Z(\boldsymbol{u}) = z\} \mid \boldsymbol{c})
\tag{199}
$$

$$
+ \mathbb{E}_{P^{\mathcal{M}_d^2}}[\, Y \mid z, \boldsymbol{c}, \{\boldsymbol{u} : f_Z(\boldsymbol{u}) \neq z\} \,] P^{\mathcal{M}^2}(\{\boldsymbol{u} : f_Z(\boldsymbol{u}) \neq z\} \mid \boldsymbol{c})
\tag{200}
$$

$$
= \mathbb{E}_{P^{\mathcal{M}_d^2}}[\, Y \mid z, \boldsymbol{c} \,] P^{\mathcal{M}_d^2}(z \mid \boldsymbol{c}) + 1 - P^{\mathcal{M}_d^2}(z \mid \boldsymbol{c}).
\tag{201}
$$

We therefore find that,

$$
0 \leqslant \mathbb{E}_{\widehat{P}}[\, Y_{d,z} \mid z', \boldsymbol{c} \,] \leqslant 1,
\tag{202}
$$

and ultimately,

$$
-\mathbb{E}_{P_d}[\, Y \mid z, \boldsymbol{c}] \leqslant \Upsilon(d, \boldsymbol{c}) \leqslant 1 - \mathbb{E}_{P_d}[\, Y \mid z, \boldsymbol{c}],
\tag{203}
$$

as claimed.   $\square$

**Thm. 6 restated.** Consider an agent with utility function $Y$ grounded in a domain $\mathcal{M}$. Then,

$$
\max\{0, \mathbb{E}_{P_d}[\, Y \mid \boldsymbol{c} \,] + \mathbb{E}_{P_{d_0}}[\, Y \mid \boldsymbol{c} \,] - 1\} \leqslant \Omega(d, d_0) \leqslant \min\{\mathbb{E}_{P_d}[\, Y \mid \boldsymbol{c} \,], \mathbb{E}_{P_{d_0}}[\, Y \mid \boldsymbol{c} \,]\}
\tag{204}
$$

and this bound is tight.

*Proof.* Consider an agent with internal model $\widehat{\mathcal{M}}$ and utility function $Y$. Recall that the agent's expected harm of a decision $d$ with respect to a baseline $d_0$, in context $\boldsymbol{c}$, is

$$
\Omega(d, d_0) := \mathbb{E}_{\widehat{P}}[\, \max\{0, Y_{d_0} - Y_d\} \mid \boldsymbol{c} \,].
\tag{205}
$$

We can re-write this quantity as follows

$$
\Omega(d, d_0) = \mathbb{E}_{\widehat{P}}[\, \max\{0, Y_{d_0} - Y_d\} \mid \boldsymbol{c} \,]
\tag{206}
$$

$$
= \int \max\{0, y_{d_0} - y_d\} \widehat{P}(y_d, y_{d_0} \mid c) dy_d dy_{d_0}
\tag{207}
$$

Since $Y_d$ is binary, the only time that the maximum evaluates to something greater than zero is when $Y_{d_0} = 1$ and $Y_d = 0$. Then,

$$
\Omega(d, d_0) = \widehat{P}(Y_{d_0} = 1, Y_d = 0)
\tag{208}
$$

This quantity can be tightly bounded using the results of (Tian & Pearl, 2000, Sec. 4.2.2) giving

$$\max\{0, \mathbb{E}_{\hat{P}_d}\left[\, Y \mid \boldsymbol{c} \,\right] + \mathbb{E}_{\hat{P}_{d_0}}\left[\, Y \mid \boldsymbol{c} \,\right] - 1\} \leqslant \Omega(d, d_0) \leqslant \min\{\mathbb{E}_{\hat{P}_d}\left[\, Y \mid \boldsymbol{c} \,\right], \mathbb{E}_{\hat{P}_{d_0}}\left[\, Y \mid \boldsymbol{c} \,\right]\}. \tag{209}$$

And by grounding,

$$\max\{0, \mathbb{E}_{P_d}\left[\, Y \mid \boldsymbol{c} \,\right] + \mathbb{E}_{P_{d_0}}\left[\, Y \mid \boldsymbol{c} \,\right] - 1\} \leqslant \Omega(d, d_0) \leqslant \min\{\mathbb{E}_{P_d}\left[\, Y \mid \boldsymbol{c} \,\right], \mathbb{E}_{P_{d_0}}\left[\, Y \mid \boldsymbol{c} \,\right]\}. \tag{210}$$

$\square$

# D. Other accounts of fairness and harm

To ground definitions of fairness, several authors appeal to counterfactual thinking but some accounts, instead, are interventional in nature.

Within legal systems, counterfactual fairness (Def. 7) operationalizes a doctrine known as disparate impact doctrine focuses on outcome fairness, namely, the equality of outcomes among protected groups. On the other hand, the doctrine of disparate treatment seeks to enforce the equality of treatment in different groups, prohibiting the use of a protected attribute in the decision process, and has been formalized using interventional accounts (Barocas & Selbst, 2016).

A popular notion in the disparate treatment literature is known as direct discrimination (Barocas & Selbst, 2016; Zhang & Bareinboim, 2018). An agent is said to engage in direct discrimination if the causal influence of a sensitive attribute $Z$ that is not mediated by other variables $C$ is non-zero. This is a contrast between interventional expectations. We adapt this notion to define an AI's perceived direct fairness gap as the difference in expected utilities obtained for different values of a protected attribute while holding all other variables fixed.

**Definition 10** (Direct Discrimination Gap). *Let $Z \in \{z_0, z_1\}$ be a protected attribute. For a given utility $Y$, define an agent's direct discrimination gap relative to a baseline value $z_0$ in a given context $c$ as*

$$\Psi(d, c) := \mathbb{E}_{\widehat{P}}\left[ Y_{d, z_1, c} \right] - \mathbb{E}_{\widehat{P}}\left[ Y_{d, z_0, c} \right]. \tag{211}$$

We say that an AI "intends" to avoid direct discrimination if under any context $C = c$ and decision $D = d$ the direct discrimination gap $\Psi$ evaluates to 0. Here, we consider this notion of fairness to illustrate the kind of inference that is possible to obtain from an AI's external behaviour with one alternative account. The following theorem shows that, contrary to the counterfactual fairness gap, $\Psi$ can be bounded given the AI's external behaviour.

**Theorem 9.** *Consider an agent with utility $Y$ grounded in a domain $\mathcal{M}$. Then,*

$$\Psi(d, c) \geqslant \mathbb{E}_{P_d}\left[ Y \mid z_1, c \right] P_d(z_1, c) - \mathbb{E}_{P_d}\left[ Y \mid z_0, c \right] P_d(z_0, c) + P_d(z_0, c) - 1, \tag{212}$$

$$\Psi(d, c) \leqslant \mathbb{E}_{P_d}\left[ Y \mid z_1, c \right] P_d(z_1, c) - \mathbb{E}_{P_d}\left[ Y \mid z_0, c \right] P_d(z_0, c) + 1 - P_d(z_1, c). \tag{213}$$

*This bound is tight.*

*Proof.* Let $Z \in \{0, 1\}$ be a protected attribute and $z_0$ a baseline value of $Z$. For a given utility variable $Y$, recall that the AI's direct fairness gap relative to a baseline $z_0$ in a given context $c$ is defined as

$$\Psi(d, c) := \mathbb{E}_{\widehat{P}}\left[ Y_{d, z_1, c} \right] - \mathbb{E}_{\widehat{P}}\left[ Y_{d, z_0, c} \right]. \tag{214}$$

Using a similar proof strategy to that in Thm. 1, we can derive tight bounds on $\Psi$.

**Analytical Lower Bound** A lower bound on the interventional expectation can be obtained using the following derivation:

$$\mathbb{E}_{\widehat{P}}\left[ Y_{z, c, d} \right] = \sum_{\tilde{c}, \tilde{z}} \mathbb{E}_{\widehat{P}}\left[ Y_{z, c, d} \mathbb{1}_{\tilde{c}, \tilde{z}}(C_d, Z_{c, d}) \right] \qquad \text{marginalizing over } c_d, z_{c,d} \tag{215}$$

$$\geqslant \mathbb{E}_{\widehat{P}}\left[ Y_{z, c, d} \mathbb{1}_{c, z}(C_d, Z_{c, d}) \right] \qquad \text{since summands } > 0 \tag{216}$$

$$= \mathbb{E}_{\widehat{P}}\left[ Y_{c, d} \mathbb{1}_{c, z}(C_d, Z_{c, d}) \right] \qquad \text{by consistency} \tag{217}$$

$$= \mathbb{E}_{\widehat{P}}\left[ Y_d \mathbb{1}_{c, z}(C_d, Z_d) \right] \qquad \text{by consistency} \tag{218}$$

$$= \mathbb{E}_{P_d}\left[ Y \mathbb{1}_{c, z}(C, Z) \right] \qquad \text{by grounding} \tag{219}$$

$$= \mathbb{E}_{P_d}\left[ Y \mid c, z \right] P_d(c, z). \tag{220}$$

**Analytical Upper Bound** For deriving an upper bound on the interventional expectation, we start by noting that,

$$\mathbb{E}_{\widehat{P}}\left[ Y_{z, c, d} \right] = 1 - \mathbb{E}_{\widehat{P}}\left[ 1 - Y_{z, c, d} \right] \tag{221}$$

Leveraging the bounds derived above we obtain,

$$\mathbb{E}_{\widehat{P}}\left[ Y_{z, c, d} \right] \leqslant 1 - \mathbb{E}_{P_d}\left[ (1 - Y) \mid c, z \right] P_d(c, z) \tag{222}$$

$$= \mathbb{E}_{P_d}\left[ Y \mid c, z \right] P_d(c, z) + 1 - P_d(c, z). \tag{223}$$

By setting $z = z_1$ in the lower bound and $z = z_0$ in the upper bound of the expected utility, we obtain a lower bound on the difference of expected utilities:

$$\Psi(d, \boldsymbol{c}) \geqslant \mathbb{E}_{P_d}[\, Y \mid z_1, \boldsymbol{c} \,]\, P_d(z_1, c) - \mathbb{E}_{P_d}[\, Y \mid z_0, \boldsymbol{c} \,]\, P_d(z_0, \boldsymbol{c}) + P_d(z_0, \boldsymbol{c}) - 1. \tag{224}$$

And similarly, by setting $z = z_1$ in the upper bound and $z = z_0$ in the lower bound of the expected utility, we obtain an upper bound on the difference of expected utilities:

$$\Psi(d, \boldsymbol{c}) \leqslant \mathbb{E}_{P_d}[\, Y \mid z_1, \boldsymbol{c} \,]\, P_d(z_1, c) - \mathbb{E}_{P_d}[\, Y \mid z_0, \boldsymbol{c} \,]\, P_d(z_0, \boldsymbol{c}) + 1 - P_d(z_1, \boldsymbol{c}). \tag{225}$$

We now show that these bounds are tight by constructing SCMs (that is, possible world models of the AI system) that evaluate to the lower and upper bounds while generating the distribution of agent interactions $P_d$.

**Tightness Lower Bound**   For the lower bound we will consider the following SCM,

$$\mathcal{M}_d^1 =: \begin{cases} Z \leftarrow f_Z(\boldsymbol{u}) \\ \boldsymbol{C} \leftarrow f_{\boldsymbol{C}}(\boldsymbol{u}) \\ D \leftarrow d \\ Y \leftarrow \begin{cases} f_Y(d, \boldsymbol{c}, z_1, \boldsymbol{u}) & \text{if } f_Z(\boldsymbol{u}) = z_1, f_{\boldsymbol{C}}(\boldsymbol{u}) = \boldsymbol{c} \\ 0 & \text{if } f_Z(\boldsymbol{u}) \neq z_1 \text{ or } f_{\boldsymbol{C}}(\boldsymbol{u}) \neq \boldsymbol{c}, \text{ and } Z = z_1 \\ f_Y(d, \boldsymbol{c}, z_0, \boldsymbol{u}) & \text{if } f_Z(\boldsymbol{u}) = z_0, f_{\boldsymbol{C}}(\boldsymbol{u}) = \boldsymbol{c} \\ 1 & \text{if } f_Z(\boldsymbol{u}) \neq z_0 \text{ or } f_{\boldsymbol{C}}(\boldsymbol{u}) \neq \boldsymbol{c}, \text{ and } Z = z_0 \end{cases} \\ P(\boldsymbol{U}) \end{cases} \tag{226}$$

Here $\{f_Z, f_{\boldsymbol{C}}, f_Y, \mathcal{U}, P(\boldsymbol{U})\}$ are chosen to match the observed trajectory of agent interactions, i.e., such that $P^{\mathcal{M}_d^1}(\boldsymbol{v}) = P^{\widehat{\mathcal{M}}_d}(\boldsymbol{v})$ for all $\boldsymbol{v} \in \text{supp}_V$.

Then, under $\mathcal{M}_d^1$,

$$\Psi(d, \boldsymbol{c}) = \mathbb{E}_{P^{\mathcal{M}^1}}[\, Y_{d, z_1, \boldsymbol{c}} \,] - \mathbb{E}_{P^{\mathcal{M}^1}}[\, Y_{d, z_0, \boldsymbol{c}} \,] \tag{227}$$

$$= \sum_{\boldsymbol{u}} \mathbb{E}_{P^{\mathcal{M}^1}}[\, Y_{d, z_1, \boldsymbol{c}} \mid \boldsymbol{u} \,]\, P^{\mathcal{M}^1}(\boldsymbol{u}) \tag{228}$$

$$- \sum_{\boldsymbol{u}} \mathbb{E}_{P^{\mathcal{M}^1}}[\, Y_{d, z_0, \boldsymbol{c}} \mid \boldsymbol{u} \,]\, P^{\mathcal{M}^1}(\boldsymbol{u}) \tag{229}$$

$$= \sum_{\boldsymbol{u}} \mathbb{E}_{P^{\mathcal{M}^1}}[\, Y_d \mid z, \boldsymbol{u}, \boldsymbol{c} \,]\, P^{\mathcal{M}^1}(\boldsymbol{u}) \tag{230}$$

$$- \sum_{\boldsymbol{u}} \mathbb{E}_{P^{\mathcal{M}^1}}[\, Y_d \mid z, \boldsymbol{u}, \boldsymbol{c} \,]\, P^{\mathcal{M}^1}(\boldsymbol{u}) \tag{231}$$

$$= \mathbb{E}_{P^{\mathcal{M}_d^1}}[\, Y \mid z_1, \boldsymbol{c}, \{\boldsymbol{u} : f_Z(\boldsymbol{u}) = z_1, f_{\boldsymbol{C}}(\boldsymbol{u}) = \boldsymbol{c}\} \,]\, P^{\mathcal{M}^1}(\{\boldsymbol{u} : f_Z(\boldsymbol{u}) = z_1, f_{\boldsymbol{C}}(\boldsymbol{u}) = \boldsymbol{c}\}) \tag{232}$$

$$+ \mathbb{E}_{P^{\mathcal{M}_d^1}}[\, Y \mid z_1, \boldsymbol{c}, \{\boldsymbol{u} : f_Z(\boldsymbol{u}) \neq z_1 \text{ or } f_{\boldsymbol{C}}(\boldsymbol{u}) \neq \boldsymbol{c}\} \,]\, P^{\mathcal{M}^1}(\{\boldsymbol{u} : f_Z(\boldsymbol{u}) \neq z_1 \text{ or } f_{\boldsymbol{C}}(\boldsymbol{u}) \neq \boldsymbol{c}\}) \tag{233}$$

$$- \mathbb{E}_{P^{\mathcal{M}_d^1}}[\, Y \mid z_0, \boldsymbol{c}, \{\boldsymbol{u} : f_Z(\boldsymbol{u}) = z_0, f_{\boldsymbol{C}}(\boldsymbol{u}) = \boldsymbol{c}\} \,]\, P^{\mathcal{M}^1}(\{\boldsymbol{u} : f_Z(\boldsymbol{u}) = z_0, f_{\boldsymbol{C}}(\boldsymbol{u}) = \boldsymbol{c}\}) \tag{234}$$

$$- \mathbb{E}_{P^{\mathcal{M}_d^1}}[\, Y \mid z_0, \boldsymbol{c}, \{\boldsymbol{u} : f_Z(\boldsymbol{u}) \neq z_0 \text{ or } f_{\boldsymbol{C}}(\boldsymbol{u}) \neq \boldsymbol{c}\} \,]\, P^{\mathcal{M}^1}(\{\boldsymbol{u} : f_Z(\boldsymbol{u}) \neq z_0 \text{ or } f_{\boldsymbol{C}}(\boldsymbol{u}) \neq \boldsymbol{c}\}) \tag{235}$$

$$= \mathbb{E}_{P^{\mathcal{M}_d^1}}[\, Y \mid z_1, \boldsymbol{c} \,]\, P^{\mathcal{M}_d^1}(z_1, \boldsymbol{c}) - \mathbb{E}_{P^{\mathcal{M}_d^1}}[\, Y \mid z_0, \boldsymbol{c} \,]\, P^{\mathcal{M}_d^1}(z_0, \boldsymbol{c}) - 1 + P^{\mathcal{M}_d^1}(z_0, \boldsymbol{c}). \tag{236}$$

This expression is the same one as the analytical bound showing that it is tight.

**Tightness Upper Bound** For the upper bound we will consider the following SCM,

$$
\mathcal{M}_d^2 =: \begin{cases}
Z \leftarrow f_Z(\boldsymbol{u}) \\
\boldsymbol{C} \leftarrow f_{\boldsymbol{C}}(\boldsymbol{u}) \\
D \leftarrow d \\
Y \leftarrow \begin{cases}
f_Y(d, \boldsymbol{c}, z_1, \boldsymbol{u}) & \text{if } f_Z(\boldsymbol{u}) = z_1, f_{\boldsymbol{C}}(\boldsymbol{u}) = \boldsymbol{c} \\
1 & \text{if } f_Z(\boldsymbol{u}) \neq z_1 \text{ or } f_{\boldsymbol{C}}(\boldsymbol{u}) \neq \boldsymbol{c}, \text{ and } Z = z_1 \\
f_Y(d, \boldsymbol{c}, z_0, \boldsymbol{u}) & \text{if } f_Z(\boldsymbol{u}) = z_0, f_{\boldsymbol{C}}(\boldsymbol{u}) = \boldsymbol{c} \\
0 & \text{if } f_Z(\boldsymbol{u}) \neq z_0 \text{ or } f_{\boldsymbol{C}}(\boldsymbol{u}) \neq \boldsymbol{c}, \text{ and } Z = z_0
\end{cases} \\
P(\boldsymbol{U})
\end{cases}
\tag{237}
$$

Here $\{f_Z, f_{\boldsymbol{C}}, f_Y, \mathcal{U}, P(\boldsymbol{U})\}$ are chosen to match the observed trajectory of agent interactions, i.e., such that $P^{\mathcal{M}_d^2}(\boldsymbol{v}) = P^{\widehat{\mathcal{M}}_d}(\boldsymbol{v})$ for all $\boldsymbol{v} \in \mathrm{supp}_{\boldsymbol{V}}$.

Then, under $\mathcal{M}_d^2$,

$$
\Psi(d, \boldsymbol{c}) = \mathbb{E}_{P^{\mathcal{M}^2}}[\, Y_{d,z_1,\boldsymbol{c}} \,] - \mathbb{E}_{P^{\mathcal{M}^2}}[\, Y_{d,z_0,\boldsymbol{c}} \,]
\tag{238}
$$

$$
= \sum_{\boldsymbol{u}} \mathbb{E}_{P^{\mathcal{M}^2}}[\, Y_{d,z_1,\boldsymbol{c}} \mid \boldsymbol{u} \,] P^{\mathcal{M}^2}(\boldsymbol{u})
\tag{239}
$$

$$
- \sum_{\boldsymbol{u}} \mathbb{E}_{P^{\mathcal{M}^2}}[\, Y_{d,z_0,\boldsymbol{c}} \mid \boldsymbol{u} \,] P^{\mathcal{M}^2}(\boldsymbol{u})
\tag{240}
$$

$$
= \sum_{\boldsymbol{u}} \mathbb{E}_{P^{\mathcal{M}^2}}[\, Y_d \mid z, \boldsymbol{u}, \boldsymbol{c} \,] P^{\mathcal{M}^2}(\boldsymbol{u})
\tag{241}
$$

$$
- \sum_{\boldsymbol{u}} \mathbb{E}_{P^{\mathcal{M}^2}}[\, Y_d \mid z, \boldsymbol{u}, \boldsymbol{c} \,] P^{\mathcal{M}^2}(\boldsymbol{u})
\tag{242}
$$

$$
= \mathbb{E}_{P^{\mathcal{M}_d^2}}[\, Y \mid z_1, \boldsymbol{c}, \{\boldsymbol{u} : f_Z(\boldsymbol{u}) = z_1, f_{\boldsymbol{C}}(\boldsymbol{u}) = \boldsymbol{c}\} \,] P^{\mathcal{M}^2}(\{\boldsymbol{u} : f_Z(\boldsymbol{u}) = z_1, f_{\boldsymbol{C}}(\boldsymbol{u}) = \boldsymbol{c}\})
\tag{243}
$$

$$
+ \mathbb{E}_{P^{\mathcal{M}_d^2}}[\, Y \mid z_1, \boldsymbol{c}, \{\boldsymbol{u} : f_Z(\boldsymbol{u}) \neq z_1 \text{ or } f_{\boldsymbol{C}}(\boldsymbol{u}) \neq \boldsymbol{c}\} \,] P^{\mathcal{M}^2}(\{\boldsymbol{u} : f_Z(\boldsymbol{u}) \neq z_1 \text{ or } f_{\boldsymbol{C}}(\boldsymbol{u}) \neq \boldsymbol{c}\})
\tag{244}
$$

$$
- \mathbb{E}_{P^{\mathcal{M}_d^2}}[\, Y \mid z_0, \boldsymbol{c}, \{\boldsymbol{u} : f_Z(\boldsymbol{u}) = z_0, f_{\boldsymbol{C}}(\boldsymbol{u}) = \boldsymbol{c}\} \,] P^{\mathcal{M}^2}(\{\boldsymbol{u} : f_Z(\boldsymbol{u}) = z_0, f_{\boldsymbol{C}}(\boldsymbol{u}) = \boldsymbol{c}\})
\tag{245}
$$

$$
- \mathbb{E}_{P^{\mathcal{M}_d^2}}[\, Y \mid z_0, \boldsymbol{c}, \{\boldsymbol{u} : f_Z(\boldsymbol{u}) \neq z_0 \text{ or } f_{\boldsymbol{C}}(\boldsymbol{u}) \neq \boldsymbol{c}\} \,] P^{\mathcal{M}^2}(\{\boldsymbol{u} : f_Z(\boldsymbol{u}) \neq z_0 \text{ or } f_{\boldsymbol{C}}(\boldsymbol{u}) \neq \boldsymbol{c}\})
\tag{246}
$$

$$
= \mathbb{E}_{P^{\mathcal{M}_d^2}}[\, Y \mid z_1, \boldsymbol{c} \,] P^{\mathcal{M}_d^2}(z_1, \boldsymbol{c}) + 1 - P^{\mathcal{M}_d^2}(z_1, \boldsymbol{c}) - \mathbb{E}_{P^{\mathcal{M}_d^2}}[\, Y \mid z_0, \boldsymbol{c} \,] P^{\mathcal{M}_d^2}(z_0, \boldsymbol{c}).
\tag{247}
$$

This expression is the same one as the analytical bound showing that it is tight. $\qquad\square$

Definitions of harm (defined with respect to a causal model) can also be split in two groups: causal and counterfactual accounts. (Beckers et al., 2022) exemplify the causal account as defining a *decision $d$ to harm a person if and only $d$ is a cause of harm*. Recall that the counterfactual account has the same structure but differs in the second clause, instead defining a *decision $d$ to harm a person if and only if she would have been better off if $d$ had not been taken*. Here, we quantify how "good" or "beneficial" a particular situation $\boldsymbol{V} = \boldsymbol{v}$ is with a binary utility $Y \in \{y_0, y_1\}$ that we assume is tracked in experiments (it might capture, for example, the value of sensitive environmental variables). A formalisation of this causal account of harm, with respect to an AI's internal model, is given in the following definition.

**Definition 11** (Causal Harm Gap). *Consider an agent with internal model $\widehat{\mathcal{M}}$ and utility $Y \in \{y_0, y_1\}$. The agent's expected causal harm of a decision $d$ with respect to a baseline $d_0$ that obtained the non-harmful outcome $y_0$ in context $\boldsymbol{c}$, is*

$$
\Omega(d_1, d_0, \boldsymbol{c}) := \mathbb{E}_{\widehat{P}}[\, Y_{d_1} \mid y_0, d_0, \boldsymbol{c} \,].
\tag{248}
$$

This probability expresses the capacity of $d_1$ to produce a harmful event $Y = y_1$ that implies a transition from the absence to the presence of $d_1$ and $y_1$, we condition the probability on situations where $d_1$ and $y_1$ are absent, i.e. $D = d_0, Y = y_0$.

**Theorem 10.** *Consider an agent with utility $Y$ grounded in a domain $\mathcal{M}$. Then,*

$$
\frac{P_{d_1}(y_1 \mid \boldsymbol{c}) - P(y_{1,d_1} \mid \boldsymbol{c})}{P_{d_0}(y_0 \mid \boldsymbol{c}) P(d_0 \mid \boldsymbol{c})} \leqslant \Omega(d_1, d_0, \boldsymbol{c}) \leqslant \frac{P_{d_1}(y_1 \mid \boldsymbol{c}) - P_{d_1}(y_1 \mid \boldsymbol{c}) P(d_1 \mid \boldsymbol{c})}{P_{d_0}(y_0 \mid \boldsymbol{c}) P(d_0 \mid \boldsymbol{c})}.
\tag{249}
$$

*Proof.* Note that the causal harm gap may be equivalently written,

$$\Omega(d_1, d_0, \boldsymbol{c}) := \widehat{P}(y_{1,d_1} \mid y_0, d_0, \boldsymbol{c}). \tag{250}$$

The lower and upper bounds may be derived considering the following,

$$\widehat{P}(y_{1,d_1} \mid \boldsymbol{c}) = \widehat{P}(y_{1,d_1}, y_0, d_0 \mid \boldsymbol{c}) + \widehat{P}(y_{1,d_1}, y_1, d_0 \mid \boldsymbol{c}) + \widehat{P}(y_{1,d_1}, y_0, d_1 \mid \boldsymbol{c}) + \widehat{P}(y_{1,d_1}, y_1, d_1 \mid \boldsymbol{c}) \tag{251}$$

$$= \widehat{P}(y_{1,d_1}, y_0, d_0 \mid \boldsymbol{c}) + \widehat{P}(y_{1,d_1}, y_1, d_0 \mid \boldsymbol{c}) + \widehat{P}(y_{1,d_1}, y_1, d_1 \mid \boldsymbol{c}) \tag{252}$$

$$= \widehat{P}(y_{1,d_1}, y_0, d_0 \mid \boldsymbol{c}) + \widehat{P}(y_{1,d_1}, y_1 \mid \boldsymbol{c}) \tag{253}$$

$$\leqslant \widehat{P}(y_{1,d_1}, y_0, d_0 \mid \boldsymbol{c}) + \widehat{P}(y_{1,d_1} \mid \boldsymbol{c}) \tag{254}$$

$$\widehat{P}(y_{1,d_1} \mid \boldsymbol{c}) = \widehat{P}(y_{1,d_1}, y_0, d_0 \mid \boldsymbol{c}) + \widehat{P}(y_{1,d_1}, y_1 \mid \boldsymbol{c}) \tag{255}$$

$$\geqslant \widehat{P}(y_{1,d_1}, y_0, d_0 \mid \boldsymbol{c}) + \widehat{P}(y_{1,d_1}, y_1, d_1 \mid \boldsymbol{c}) \tag{256}$$

$$= \widehat{P}(y_{1,d_1}, y_0, d_0 \mid \boldsymbol{c}) + \widehat{P}(y_{1,d_1}, d_1 \mid \boldsymbol{c}) \qquad \text{by consistency} \tag{257}$$

$$= \widehat{P}(y_{1,d_1}, y_0, d_0 \mid \boldsymbol{c}) + \widehat{P}(y_{1,d_1}, d_1 \mid \boldsymbol{c})\widehat{P}(d_1 \mid \boldsymbol{c}). \tag{258}$$

$\widehat{P}(d_1 \mid \boldsymbol{c})$ stands for the AI's policy in the source environment, i.e., the probability it uses for choosing decision $d_1$ in situation $\boldsymbol{c}$. Re-arranging these equations this implies,

$$\frac{\widehat{P}(y_{1,d_1} \mid \boldsymbol{c}) - \widehat{P}(y_{1,d_1} \mid \boldsymbol{c})}{\widehat{P}(y_{0,d_0} \mid \boldsymbol{c})\widehat{P}(d_0 \mid \boldsymbol{c})} \leqslant \Omega(d_1, d_0, \boldsymbol{c}) \leqslant \frac{\widehat{P}(y_{1,d_1} \mid \boldsymbol{c}) - \widehat{P}(y_{1,d_1} \mid \boldsymbol{c})\widehat{P}(d_1 \mid \boldsymbol{c})}{\widehat{P}(y_{0,d_0} \mid \boldsymbol{c})\widehat{P}(d_0 \mid \boldsymbol{c})}. \tag{259}$$

And by grounding,

$$\frac{P_{d_1}(y_1 \mid \boldsymbol{c}) - P(y_{1,d_1} \mid \boldsymbol{c})}{P_{d_0}(y_0 \mid \boldsymbol{c})P(d_0 \mid \boldsymbol{c})} \leqslant \Omega(d_1, d_0, \boldsymbol{c}) \leqslant \frac{P_{d_1}(y_1 \mid \boldsymbol{c}) - P_{d_1}(y_1 \mid \boldsymbol{c})P(d_1 \mid \boldsymbol{c})}{P_{d_0}(y_0 \mid \boldsymbol{c})P(d_0 \mid \boldsymbol{c})}. \tag{260}$$

$\square$

