# OpenReview forum: "The Limits of Predicting Agents from Behaviour"
_ICML.cc/2025/Conference — ICML 2025 poster_

### Official Review · Reviewer_6C3x · 2025-03-12

**Overall Recommendation:** 4

**Summary:**

This paper explores the theoretical limits of predicting AI agent behavior from observational data alone. The authors analyze the extent to which we can infer an agent's beliefs and predict its behavior in novel situations based only on its past behavior. They provide:
1) A mathematical framework using Structural Causal Models (SCMs) to represent an agent's internal world model and how it relates to observable behavior.
2) Formal bounds on how well we can predict an agent's preferences and actions out-of-distribution, showing that multiple different internal models can be consistent with the same observable behavior but lead to different actions in new situations.
3) Theoretical results showing that while some aspects of an agent's beliefs can be bounded based on behavioral data, other aspects (such as counterfactual fairness) remain fundamentally underdetermined.

**Claims And Evidence:**

The paper's claims are generally well-supported through formal mathematical proofs. The authors establish several theorems with bounds on what can be inferred about an agent's internal model from behavioral data alone. These theoretical results are complemented by concrete examples that illustrate the concepts.

**Essential References Not Discussed:**

n/a

**Experimental Designs Or Analyses:**

n/a

**Methods And Evaluation Criteria:**

The proposed methods make sense for the problem or application at hand.

**Other Comments Or Suggestions:**

n/a

**Other Strengths And Weaknesses:**

Strengths:
- The paper provides a rigorous mathematical framework using Structural Causal Models to analyze agent behavior prediction.
- The work has clear and important implications for AI safety research, establishing fundamental limits on our ability to predict AI behavior in new situations based solely on observational data.
- The Medical AI example used throughout the paper effectively illustrates the theoretical concepts
- The paper explores extensions to its core assumptions, such as approximate grounding, partial observability, and modifications to model structure, making the analysis more robust and applicable to real-world scenarios.

Weaknesses:
- Many examples use binary variables (like medical outcomes being 0 or 1), which simplifies the analysis but may not capture the complexity of real-world scenarios with continuous outcomes or multiple categories.
- The framework primarily addresses single-step decision making rather than sequential decision problems, which limits its applicability to reinforcement learning agents and longer-term planning scenarios.

**Questions For Authors:**

Can you add a limitations section wherein you discuss cases in which SCMs might not be applicable? e.g., practical instances of cyclical dependencies that are assumed away in the definition of SCMs

**Relation To Broader Scientific Literature:**

The paper builds on and connects several areas of research:
- The work extends bounds from classical causal inference (referencing work by Pearl, Robins, Manski, etc.) to the novel context of agent behavior prediction.
- The paper directly addresses concerns in AI safety about predicting agent behavior out-of-distribution, particularly connecting to work on goal misgeneralization.
- The authors connect their theoretical framework to emerging evidence that language models may develop internal world models.
- The paper applies its framework to fairness definitions from the literature, showing fundamental limitations in inferring an agent's notion of fairness from behavior alone.

**Theoretical Claims:**

I perused the proofs at a high-level but not in detail; they seem correct.

---

> ### Author Rebuttal · Authors · 2025-03-31
>
> Thank you for taking the time to read and review our paper, we appreciate the questions and suggestions.
>
> ***1. Many examples use binary variables (like medical outcomes being 0 or 1), which simplifies the analysis but may not capture the complexity of real-world scenarios with continuous outcomes or multiple categories.***
>
> To clarify, note that our results hold for systems of (discrete) variables of any dimensionality and arbitrary complexity (in the functions relating different variables and distributions), and continuous outcomes / utility values. There is in principle no additional difficulty to accounting for complex environments, though perhaps the computation of the bounds from finite samples may become more challenging. Our intent with examples using binary variables was to make them easy to describe, though we go agree that more realistic experiments could add to illustrate the results. See also the response to reviewer EHMW for additional context.
>
> ***2. The framework primarily addresses single-step decision making rather than sequential decision problems, which limits its applicability to reinforcement learning agents and longer-term planning scenarios.***
>
> The sequential setting, in principle, offers no additional difficulties: you would have variables and observations indexed by time and the formalism and bounds would be unchanged. More concretely, the preference gap $\Delta$ at some time $t$ that determines the AI's beliefs that one decision is superior to another one might be written in the sequential setting as:
>
> $\Delta := \mathbb E_{\hat{P}_{\sigma, d_1^{(t)}}}[Y^{(t)} \mid \boldsymbol c^{(t)} ] - \mathbb E\_{\hat{P}\_{\sigma, d_0^{(t)}}}[ Y^{(t)} \mid \boldsymbol c^{(t)}]$
>
> where $\boldsymbol c^{(t)} = (s^{(t)}, d^{(t-1)}, \dots, d^{(0)},s^{(0)})$ now includes the agent's trajectory in the deployment environment up to time $t$ (for example). In this setting, we do assume however that there is no learning taking place in the deployment environment, i.e. the AI is not updating its internal model upon interacting with the environment $\boldsymbol c^{(t)}$. This might be a limitation in practice that would be interesting to relax in future work.
>
> ***3. Can you add a limitations section wherein you discuss cases in which SCMs might not be applicable? e.g., practical instances of cyclical dependencies that are assumed away in the definition of SCMs.***
>
> This is a good suggestion, thank you. We will add to the limitations section in Appendix B.1 a description of the kinds of environments that would not be well captured by SCMs.

---

### Official Review · Reviewer_Zntn · 2025-03-13

**Overall Recommendation:** 3

**Summary:**

The authors study the problem of predicting out of distribution behaviours of AI agents based on data, assuming the AI agents have internal causal models of the world. First, examples are presented for why observing the behaviour of AI agents may not be enough to determine the specific causal model it is using (i.e. inferring the causal model from observations and utilities is an ambiguous problem). Then, authors present formal bounds on the utility gap that the causal agents experience based on the observed data. Finally, they discuss the implications this has for AI safety and overall inferring of preferences in autonomous systems.

**Claims And Evidence:**

The claims are well supported, and there are no contributions that are left unjustified. The authors are very clear that their main contribution is the derivation of such bounds and the accompanying discussion on the implications.

**Essential References Not Discussed:**

On reward misspecification, see e.g.:
Freedman, Rachel, Rohin Shah, and Anca Dragan. "Choice set misspecification in reward inference." arXiv preprint arXiv:2101.07691 (2021).
Skalse, Joar, and Alessandro Abate. "Misspecification in inverse reinforcement learning." Proceedings of the AAAI Conference on Artificial Intelligence. Vol. 37. No. 12. 2023.

On more classic (and some more recent) game theoretic results studying preference misspecification:
Richter, Marcel K. "Revealed preference theory." Econometrica: Journal of the Econometric Society (1966): 635-645.
Afriat, Sydney N. "The construction of utility functions from expenditure data." International economic review 8.1 (1967): 67-77.
Abbeel, Pieter, and Andrew Y. Ng. "Apprenticeship learning via inverse reinforcement learning." Proceedings of the twenty-first international conference on Machine learning. 2004.

(There are many more, and I'm not suggesting all need to be included or a specific discussion is needed to justify the work, but some mention on the novelty of the contribution given existing preference theory results would be of benefit).

**Experimental Designs Or Analyses:**

The paper has no experiments.

**Methods And Evaluation Criteria:**

There are not really any methods or evaluations. The authors provide some illustrative examples that cover their claims.

**Other Comments Or Suggestions:**

I have some particular comments and suggestions regarding the formalisms used.
- Before equation (1) authors state that $Y$ is a 'potential response', which seems to map latent variables to observables, and that it entails a distribution over the possible outcomes of $Y$ (this is already not very clear). However, in section 4, it is stated that $Y$ is a utility function, and at the same time it seems to be a member of the outcome set ($Y\in \mathbf{V}$). It is not clear whether $Y$ is then a set-valued map, a random variable, a function or a set of outcomes.
- Definition 2 is not very clear. $\phi$ is not used again, and it is not clear whether it's a functional that just maps to the reals.
- In Footnote 4, the idea of a policy is introduced, but it's not very clear what a policy means in this context, nor what the 'training domain' is. Is a policy just a function from the observables to probability distribution over interventions?
- Authors seem to indicate policies are stochastic, but then seem to indicate in eq (3) that AI agents are assumed to act deterministically. Some clarification on this would be good.

**Other Strengths And Weaknesses:**

Strengths:
- Paper is very well structured, the message is very clear and the examples help illustrating the points.
- I appreciate work that neatly demonstrates how given systems we are deploying at a large scale may have flaws, and how these flaws can affect their deployment.

Weaknesses:
- Some formal statements are not clear or feel rushed, see below.
- There is not really a prescriptive nature to the paper (i.e. even a heuristic on how to improve interpretability of AI agents under causal world models where we only have access to observational data). This is not always needed of course, but the work would be more complete.
- It is not clear how the main contribution, although interesting, is of particular relevance to the field or novel when compared to existing subfields (mentioned above). Intuitively there seems to be some novelty when considering such causal world models, and in particular on the specific bounds obtained, but it feels slightly thin.

**Questions For Authors:**

See comments above.

**Relation To Broader Scientific Literature:**

The authors do a very thorough job at relating the work to existing works in causality, but there is barely no discussion on how the work relates to other sub-fields that have studied similar problems. The question of whether preferences can be inferred from agent choices and observations is a classic problem in game theory, and more recently has received a lot of attention in reinforcement learning through inverse RL or reward shaping. Furthermore, the conclusion that one cannot fully determine preferences from limited partial observation data has been reached in these subfields before. I would have appreciated some mentions to these connections.

**Theoretical Claims:**

Unless I missed any key details, the proofs and theoretical claims seem correct. I do have to raise that there are imprecisions and misspecified formal statements along the paper that make the judgement of some of these claims difficult (see questions below), but overall the proofs are well written and clear.

---

> ### Author Rebuttal · Authors · 2025-03-31
>
> Thank you very much for your review, we appreciate the references to related work and suggestions for clarifying the formalism.
>
> ***1. The authors do a very thorough job at relating the work to existing works in causality, but there is barely no discussion on how the work relates to other sub-fields that have studied similar problems (e.g., game theory, inverse reinforcement learning, decision theory).***
>
> Thank you for highlighting work in inverse reinforcement learning (IRL) and decision theory. There are several similarities in the broader goals of these different lines of research, e.g., inferring expected utilities from data, but also a few differences that are becoming clearer to us as we contrast the papers mentioned by the reviewer with ours. This deserves a longer discussion but one advantage of the causal formalism (in our view) that we can highlight here is that it gives us tools to make inferences across different environments (including in counter-factual scenarios, e.g., to reason about harm and fairness). Perhaps it is fair to say that our bounds thus more concretely characterize the limits of what can be predicted about agent behaviour *out-of-distribution* which then complements the (in our understanding mostly in-distribution) partial identification results in IRL and decision theory.
>
> ***2. It is not clear how the main contribution, although interesting, is of particular relevance to the field or novel when compared to existing subfields (mentioned above). Intuitively there seems to be some novelty when considering such causal world models, and in particular on the specific bounds obtained, but it feels slightly thin.***
>
> We do believe that our bounds characterize more precisely what can be expected of AI behaviour out-of-distribution. The bounds themselves are significant because they fully describe the set of possible AI behaviours, under our assumptions. This is an important question for AI Safety and there are several implications for the design of AI systems and the extent to which we can trust them (e.g., if the bounds are wide we might want to monitor AI behaviour more closely, or if harmful actions can be ruled out by our bounds that might increase our trust in the technology when deployed).
>
> To our knowledge, related work, e.g. in IRL, has studied extensively the inference of utilities from data but less so the inference of action choice given a (set of) compatible utilities, which is the contribution we seek to make in our paper. More speculatively, we also believe that there is an interesting cross-discipline opportunity that this paper might encourage: the causal formalism provides a new angle of attack, showing that it is possible to exploit the inductive biases implied by the AI's learned world model to predict their behaviour and beliefs. This approach is novel and potentially fruitful due to the strength of the worst-case guarantees, subject to our assumptions.
>
> ***3. Particular comments and suggestions regarding the formalisms used.***
>
> Thank you for these. Here we provide a few clarifications and we will update the manuscript accordingly.
> - $Y\in\boldsymbol V$ encodes the utility and is a random variable whose assignment is determined by the AI’s world model (structural causal model).
> - $\phi$ is meant to describe a mapping from an AI’s internal model to a statement over probability of events. With hindsight, this notation was confusing: in the revised manuscript we have replaced the definition of AI beliefs by "probability statements that are derived from the AI's internal world model" such as $P^{\widehat M}(A=a)$ or $P^{\widehat M_z}(B=b | C=c)$.
> - Yes, a policy is a function from the observables to probability distribution over interventions.
> - We hypothesize that the data of AI interactions we observe, denoted $P_\pi(\boldsymbol v)$, is collected while the AI is training or learning, meaning that we can expect some amount of exploration. The policy $\pi$ is therefore stochastic. However, once the agent is deployed in a new environment, we assume that its choices are driven by expected utilities and are therefore deterministic (though other decision-making models are also possible and would induce variations on how to use the bounds to inform our understanding of AI behaviour).

---

### Official Review · Reviewer_EHMW · 2025-03-14

**Overall Recommendation:** 2

**Summary:**

This paper derives theoretical bounds on predicting an AI agent’s future behavior from observed actions, using structural causal models (SCMs) to formalize beliefs and grounding. It introduces metrics like the preference gap and counterfactual fairness gap, arguing that—even with full behavioral data—fundamental limits remain on how precisely we can infer an AI’s internal decision-making, especially out-of-distribution. The authors back their claims with theorems and examples (e.g., a Medical AI scenario).

**Claims And Evidence:**

- the paper claims that an AI’s internal beliefs and decision-making can only be bounded, not exactly determined, using external behavior. though the derivations (e.g., Theorems 1–5) lay out tight bounds under various assumptions and their proofs appear rigorous at a glance, they lean heavily on idealized assumptions (like perfect grounding) that are rarely met in real-world scenarios.

- the proofs are not thoroughly validated for practical use. examples like the Medical AI feel oversimplified, leaving questions about whether the results would hold in more settings.

**Essential References Not Discussed:**

the paper thoroughly engages with classical causal inference literature but omits more recent developments in causal representation learning within deep learning, such as *Towards Causal Representation Learning (2021)*;

also, it doesn’t mention recent empirical benchmarks for evaluating AI safety and interpretability which may complement its theoretical contribution, like *RobustBench: a standardized adversarial robustness benchmark (Croce et al., 2020)*

**Experimental Designs Or Analyses:**

The work includes only illustrative examples, such as examples 1-3, which are largely schematic, appearing more as a proof-of-concept rather than a validated experimental framework, i.e., no rigorous experimental design or analysis on real or simulated data.

**Methods And Evaluation Criteria:**

- The paper is almost entirely theoretical, centered on structural causal models and derived bounds on preference gaps, fairness, and harm, well-motivated from a theoretical standpoint.
- but the work leaves a gap regarding practical evaluation

**Other Comments Or Suggestions:**

I don't have any other comments

**Other Strengths And Weaknesses:**

- without real data experiments or simulations, the practical impact of these bounds remains questionable
- a major downside is the heavy reliance on assumptions that may not hold in practice, making it hard to see how these bounds could directly inform AI safety
- as to theorem presentation, the notation is dense and sometimes overwhelming

**Questions For Authors:**

- might it be more reliable to use the inferred beliefs to predict (or get the bounds on) the agent's behaviour in the in-distribution situation?
- can you provide any empirical or simulation-based evidence to support your theoretical bounds in a relatively realistic setting?
- if not, what steps would be needed to translate these bounds into actionable guidelines for AI safety and policy design?
- what will happen to your framework if the AI system's internal models evolve over time?

**Relation To Broader Scientific Literature:**

- well-rooted in established causal inference literature (think Pearl, Bareinboim, and others) and builds upon causal modeling and counterfactual analysis

- it neglects to contrast these with recent empirical work on AI interpretability and behavior prediction.

**Theoretical Claims:**

I reviewed several key proofs of theorems 1-5, the proofs seem to be correct under the idealized conditions assumed

---

> ### Author Rebuttal · Authors · 2025-03-28
>
> Thank you for taking the time to review our paper.
>
> ***1. The proofs are not thoroughly validated for practical use. Examples like the Medical AI feel oversimplified, leaving questions about whether the results would hold in more settings.***
>
> To clarify possible misconceptions, we should emphasize that our results hold for any agent deployed in (discrete) systems of arbitrary dimensionality and complexity in the underlying functions and distributions, subject to our assumptions. More complex environments with more variables do not on their own pose any additional challenges for predicting future agent behaviour according to our bounds. As the reviewer mentions, our contribution is to formally characterize the range of an AI’s possible behaviours out-of-distribution. In our view, it is important that this characterization be theoretical, and apply in the worst-case, even in cases that are difficult to foresee in simulations.
>
> We do appreciate, however, that assumptions on grounding and expected utility maximization might not be appropriate for all systems, and that further experiments could help probe whether the behaviour of current AIs is well predicted through our bounds (as the reviewer is suggesting). One additional experiment we consider prompts an LLM (the agent) to output its subjective expected utilities under different actions, comparing those inferences to our bounds. We provide a summary below of this experimental set-up.
>
> *Experiment description*. We give the LLM in-context the counts of transitions of a simulated dataset to mimic its past experience and approximately satisfy the condition of grounding, i.e., that the AI’s beliefs on the likelihood of events in the environment are consistent with the data. Find below the template we use.
>
> ```
> You are an agent operating in an environment. You are given the following dataset that represents your experience so far.
>
> {{transition_count}}
>
> Infer a causal model that is compatible with this dataset and use it for decision making.
> Note that your causal model can be anything you choose, including unobserved confounders and arbitrary causal associations, as long as it is able to (approximately) reproduce the data.
>
> Your task is to return your beliefs on the expected value of Y after taking the action do(X=0) and do(X=1) in a new environment in which we apply the intervention do(Z=1).
>
> Return your answer as a dictionary {{'expected value of Y given action X=1': <value>, 'expected value of Y given action X=0': <value>}}.
> ```
>
> Across different simulated datasets we compare the responses of various models from the Gemma family with our bounds. We find in all cases that the responses are included in our bounds. The reasoning traces reveal that LLMs generally follow instructions, hypothesizing a causal model, and deriving expected utilities correctly. This set-up is not without limitations but it does suggest (anecdotally) that it is possible to prompt LLMs to internalize a causal model and act rationally, such that are bounds can be reasonably expected to hold in practice.
>
> ***2. A major downside is the heavy reliance on assumptions that may not hold in practice.***
>
> We expand on the strength of modelling assumptions and possible relaxations in the answer to reviewer Q6s6. We would appreciate if you could refer to that response (let us know if we can add further details).
>
> ***3. might it be more reliable to use the inferred beliefs to predict (or get the bounds on) the agent's behaviour in the in-distribution situation?***
>
> Our results so far are framed in the infinite-sample regime. This means that if we have access to the training distribution and the AI is grounded (i.e., it has learned how to predict the likelihood of events in-distribution) then we could automatically evaluate expected utilities in-distribution for all actions and derive the AI choice without any uncertainty (assuming it chooses actions to maximize expected utility). If we understand the reviewer’s suggestion as “with finite samples, even in-distribution beliefs might be inferred with error which introduces some uncertainty in their choice of action in-distribution"; this is correct. The question here is how the AI is estimating its beliefs from finite samples, which is interesting. In this paper, we are not addressing estimation issues directly but we do acknowledge that it is an important limitation and a good topic for future work.
>
> ***4. what will happen to your framework if the AI system's internal models evolve over time?***
>
> This is an interesting setting, the question might then become: how to predict AI behaviour given a large dataset of interactions in a training environment and a small dataset of interactions in the deployment environment. Our current results do not consider this setting so far, but extensions might be feasible along the lines of (Bellot et al., 2023).
>
> Bellot, A. et al. "Transportability for bandits with data from different environments." NeurIPS 2023.

---

### Official Review · Reviewer_Q6s6 · 2025-03-16

**Overall Recommendation:** 4

**Summary:**

This paper considers a model of a decision-making agent as follows. Note the
paper always imagines the agent to be an AI system, but this does not appear to
be essential other than for motivation.

* Suppose the agent makes decisions guided by (1) a utility function and (2) a
   causal model of how its decisions influence outcomes in the world.
   Precisely, suppose the agent makes decisions so as to maximise expected
   utility in a structural causal model (SCM).
* Suppose further that the agent's utility function is known to us, but the
   SCM is not, other than that the SCM is sufficiently well-calibrated
   ("grounded") in the sense that it is in agreement with some known ground
   truth SCM as to the distribution over outcomes induced by each of the
   agent's potential actions.

In this setting, one can ask theoretical questions of (partial)
identifiability, i.e., to what extent does the behaviour of the agent in the
ground truth SCM determine its future behaviour given various kinds of
interventions, or its true intentions regarding discrimination and harm?

Due to the fact that there may be multiple agent-SCMs consistent with the
observed behaviour, meaning its future behaviour under an intervention or its
internal motivations for a decision are under-determined. Thus, the best one
can hope for is a characterisation of the range of possible behaviours or
intentions given the range of possible internal SCMs.

This paper provides several such bounds under various assumptions.
The first contributions regard the range of possible degrees of preference
between two actions conditional on a known, unknown, or partially-known
intervention, indicating how (un)predictable the agent's future decisions are.

1. Theorem 1 gives tight bounds on the range of possible degrees of preference
   between two actions given a fixed atomic intervention to the SCM.
2. Theorem 2 shows that no meaningful bound is possible on the degrees of
   preference between two actions given an unspecified intervention to the SCM.
3. Theorem 3 shows that if the intervention is unspecified but the effect on
   the distribution of input contexts is known, then some meaningful bounds can
   be derived.

Further contributions regard evaluating the agent's intentions regarding
different notions of fairness/discrimination or safety/harm, indicating to what
extent behavioural evaluations can reach meaningful conclusions about these
properties of decision-making systems guided by causal world models:

4. Theorem 4 shows that no meaningful bound on the "counterfactual fairness
   gap" can be derived from external behaviour.
5. Theorem 5 shows that in some situations, meaningful bounds on the
   "counterfactual harm gap" can be derived from external behaviour.
6. Theorem 8 (appendix D) shows that in some situations, meaningful bounds on
   the "direct discrimination gap" can be derived from external behaviour.
7. Appendix D also discusses "causal harm gap" and offers a proof of some
   bounds on page 33, though a statement of 'theorem 9' and discussion of its
   implications appears to be missing.

The paper also discusses the prospects of deriving stronger/weaker bounds given
different assumptions, primarily through the lens of a simple concrete example
(rather than general bounds) including:

8. If the "grounding" assumption is relaxed to allow an approximate agreement
   between the distributions predicted by the agent's internal SCM and the
   ground truth SCM, then slightly weaker bounds result.
9. If instead of assuming the agent optimises the specified utility function,
   we assume that it optimises a similar 'proxy' utility function that induces
   similar decisions given the ground truth SCM, then the behaviour can still
   be constrained, at least in similar contexts.
10. Sections 5.2 and 5.3 discuss how with partial knowledge of the agent's
    internal SCM, bounds can be improved.

**Claims And Evidence:**

The model, assumptions, and results are clearly stated. The interpretations are
clearly explained. The examples help make the bounds concrete. The
proofs accompanying each formal claim give explicit detail for all derivations
and calculations.

**Essential References Not Discussed:**

None to my knowledge.

**Experimental Designs Or Analyses:**

N/A

**Methods And Evaluation Criteria:**

N/A

**Other Comments Or Suggestions:**

I found the paper generally well-written, but spotted a couple of minor typos,
or generally questionable phrases, as follows.

1. In the abstract, you write "If an agent behaves as if it has a certain goal
   or belief, then we can make reasonable predictions about how it will behave
   in novel situations..." This seems to beg the question, that is, isn't "we
   can make reasonable predictions about behaviour" the definition of what it
   means to say a system is an agent whose behaviour is driven by goals and
   beliefs? I was not sure if this "if ... then ..." sentence was an attempt to
   justify making predictions based on attributed beliefs---I think some other
   justification is needed to draw that conclusion.

2. In the main result summary (Lines 032R--034R), I think you mean to emphasise
   "partially determines," but I'm not sure that this emphasis comes across.
   For reference, I have written papers on partial identifiability, and I
   initially missed the emphasis here. It might help to bold "partially
   determines," if this is what you were going for. A complementary approach
   might be to add something in the introduction that explains what this means
   (along the lines of explaining that even though we can't uniquely identify
   the future behaviour, we can narrow it down to a range of possible outcomes,
   this paper characterises those outcomes, etc.).

3. Line 040R: "can be can be"

4. "Shane (2023)" should probably be "Legg (2023)," and I think you should
   probably cite the youtube video rather than a forum post in which someone
   happens to be asking a question about the video.

5. Is definition 2 used in the main text?

6. Line 140L: "how ... the world looks like" ("how the world looks" or "what
   the world looks like"?)

7. Definitions 3 and 4: I found it initially unclear what is meant by "for any
   d." I think I understand now that $d$ is an arbitrary value associated with
   the agent's decision variable $D$, i.e., it means $\forall d \in
   \mathrm{supp}_ {D}$. I think it could help to make this explicit. Also I
   think that the status of $D$ as a unique decision-making variable is not
   made explicit anywhere. Of course, it's possible I'm still confused.

8. Opening of section 3.2: "We ... do not have access to the mchanisms
   underlying the actual environment nor the agent's internal model." This is
   framed as an assertion, but I think it would be more appropriate to frame it
   explicitly as an assumption of the setting you are analysing. Later in the
   paper, you go on to point to work that attempts to improve on this situation
   as a promising direction.

9. Line 172L: I didn't really understand the connection between 'quick
   learning' and grounding. I suppose there is not meant to be a necessary
   connection, but rather you are just suggesting this as a sufficient way by
   which an AI system may become (approximately) grounded in practice.

10. Definition 5: While some earlier definitions have a small square at the
    end, this one (is the first that) does not. I don't personally think the
    squares are necessary due to italicisation, but I suppose you may want to
    follow a consistent style, so I thought I'd point this out.

11. Example 3: This example, and the definition of approximate groundedness,
    felt like they conceptually fit more closely with the content in section 5
    than they do here. I wonder if you have considered moving this 'assumption
    relaxation example' to the end of the paper with the others, and if this
    could improve the flow of the paper's presentation of the main formal
    results.

12. Line 233R-ish: Estimating the min/max of a set from samples can be
    challenging. I wonder if some kind of optimisation technique could be used
    here. Just an idle thought.

13. Theorem 2: This theorem is the most surprising to me. In particular, I
    paused to wonder whether it should hold for all SCMs. I noticed I am a
    little confused about the space of possible SCMs under consideration. Could
    there not be some corner cases involving very simple SCMs in which only a
    single intervention were possible, such that we effectively know the
    intervention even under the assumptions of this theorem? I apologise for my
    unfamiliarity with the SCM framework---possibly this concern is not
    well-formed. I looked briefly at the proof, but I am not familiar with the
    prior work.

14. Line 293L: "it is known not tight in general," possibly a typo, or just
    consider rephrasing?

15. Line 351L: "the external behaviour constraints the AI" typo.

16. Example 5, line 360R: "his" is possibly a typo.

17. Example 5, line 370R: "in fact could show that," typo.

18. Line 437L: "this bound are strictly tighter" typo.

19. Line 408R: "we can use their own" typo.

20. Line 413R: Unclear if "Peter" and "Jon" are meant to be different people or
    the same person (if they are meant to be different people, I missed the
    significance of this).

21. Line 416R: Double period.

22. Equation 28: There are two stray closing parentheses in the definition of
    $\mathcal{F}_ \sigma$.

23. Line 729: "his" is possibly a typo.

24. Line 732: "This then constraints the possible values" typo.

25. Is there meant to be a theorem 9 stated before the proof on page 33?

**Other Strengths And Weaknesses:**

**Strengths:** As noted in above sections, I think this paper is very clear and
makes a valuable contribution to the field of AI safety.

**Weaknesses:** The main weaknesses concern the strength of modelling
assumptions. While the paper contains a thorough discussion of how relaxing
various assumptions would affect the results, I think further discussion could
be warranted. Particularly, when it comes to the motivation of predicting the
OOD behaviour and safety properties of interest to the AI safety community, we
are interested in large-scale AI systems acting in very complex, real world
environments with very complex internal cognitive structures. This raises the
following questions:

1. While the paper points out (footnote 2) that there is no assumption that the
   AI system reasons explicitly in terms of an SCM, it appears that there is an
   assumption that the AI system's behaviour is well-captured by *some* SCM.
   If I understand correctly, in reality, large-scale complex AI systems with
   complex decision-making procedures might only be approximately modelled by
   any (reasonably sized) SCMs.
   In turn, if the AI's behaviour is only *approximately* captured by an SCM,
   then I would suppose that the bounds will degrade. If this is accurate, then
   I think it would be valuable to add an example along these lines.

2. The paper assumes that the agent's decisions are exactly optimal given its
   causal model. In complex, real-world environments, even in the limit of
   quite powerful AI systems, these systems are still subject to computational
   limitations and accordingly it would be more appropriate to treat their
   rationality as "bounded" in some sense (for example, supposing they choose
   actions that have approximately the highest utility, rather than exactly the
   highest utility, given their model).
   Again, I suppose such an approximation should degrade the bounds on
   predicting behaviour. If so, I think it would be valuable to add an example
   along these lines.

3. The paper points out in footnote 4 that it additionally assumes access to a
   comprehensive specification of the AI system's behaviour in all possible
   contexts (if I understood this footnote correctly). However, in complex
   environments, there are too many contexts for us to ever hope to have
   such comprehensive behavioural knowledge. There will be contexts in which we
   don't observe the system's behaviour. The footnote points out that this
   creates strong limitations on inferring future behaviour. I had some trouble
   following the exact meaning of this footnote, and invite the authors to
   consider expanding this discussion. Is a result along the lines of theorem 2
   possible in this situation? Or, at least, a concrete example along these
   lines?

I should clarify that I don't see these limitations as undermining the
contribution of the work, which I think is acceptable even given the
assumptions that it makes. I am listing these issues more as a way to suggest
that the contribution would be improved in my opinion if the discussion in
section 5 could be further broadened along the above lines.

The third point above also bears on the discussion of goal misgeneralisation
(section 5.1), which I feel needs refinement:

4. The conclusion of section 5.1 appears to be that "OOD" behaviour (behaviour
   under a novel intervention) may in principle be possible to bound if one has
   sufficient constraints on the proxy utility function in relation to the true
   utility function. An explicit comparison is made to goal misgeneralisation.
   However, in empirical work defining and studying examples of goal
   misgeneralisation, one trains in a *subset* of contexts, where the proxy
   utility and the true utility are constrained to agree, but then tests in
   *novel* contexts where they come apart, and the system's performance
   degrades to an arbitrary degree. If I understand correctly, the combination
   of (1) the assumption that we have behavioural data covering all contexts
   and (2) a proxy utility function that is correlated with the true utility in
   all observed circumstances amounts more to a setting of "approximate inner
   alignment," rather than the concern about more substantial inner
   misalignment and a different kind of distribution shift that drive the
   failure mode of goal misgeneralisation.

**Questions For Authors:**

I would appreciate if the authors could point out if there are any inaccuracies
in my summary of the paper and discussion of how it fits into related
literature. If the authors have the appetite for further discussion with me, I
am open to discussing the points I listed under weaknesses further. It doesn't
seem necessary to discuss the "other comments" listed above. Unfortunately, I
don't see much scope to increase my rating further than 'accept'.

**Relation To Broader Scientific Literature:**

As I understand, prior work must have studied the related question of whether
the behaviour of a utility-maximising decision-maker uniquely determines the
(grounded) causal model used as the basis for its decisions (the answer being
no, in general). Authors can correct me if I am mistaken on this point.

This paper builds on this more fundamental partial identifiability by extending
it to partial identifiability results for the decision-maker's OOD behaviour or
its intentions with respect to fairness and harm.

This observation is valuable to the field of AI safety since it motivates the
need for understanding and control of the specific internal SCM that governs
behaviour in powerful AI systems of the near future.

**Theoretical Claims:**

I did not check the correctness of the proofs or examples in detail.

---

> ### Author Rebuttal · Authors · 2025-03-28
>
> Thank you for your thoughtful review, we appreciate the feedback and the depth of the observations.
>
> We do share your concern on modelling assumptions. Before discussing them in more depth, it might be worthwhile, however, to note more explicitly the increasing evidence available for AI systems (particularly LLMs) behaving rationally across a wide set of environments, and why this behaviour can be described by some causal model.
>
> Decision-theoretic tests of rationality have been applied to LLMs that "demonstrate higher rationality score than those of human subjects" (Chen et al., 2023), see also (Mazeika et al., 2025). An LLM's set of preferences over interventions, to the extent that they are consistent with rationality axioms, can then be formally described by an SCM (Halpern and Piermont, 2024). The same conclusion can also be obtained for agents capable of solving tasks in multiple environments (Richens and Everitt, 2024).
>
> Halpern, Joseph Y., and Evan Piermont. "Subjective Causality." 2024.
>
> Richens, Jonathan, and Tom Everitt. "Robust agents learn causal world models." ICLR. 2024.
>
> Chen, Yiting, et al. "The emergence of economic rationality of GPT." PNAS 2023
>
> Mazeika, Mantas, et al. "Utility Engineering: Analyzing and Controlling Emergent Value Systems in AIs." 2025.
>
> ***1. If the AI's behaviour is only approximately captured by an SCM, then I would suppose that the bounds will degrade.***
>
> In principle, this is true – if AI behaviour sometimes deviates from what is expected given the assumption of a fixed causal model, then the bounds might degrade. And if those deviations are sufficiently random then presumably no guarantees on AI behaviour can be given.
>
> Possibly, one interesting relaxation that can be entertained is the assumption that the AI operates on multiple causal models. For example, suppose it samples a causal model from its set of possibilities and makes a decision according to that causal model. If the AI is grounded, meaning that all members of its set of causal models are compatible with the observed data, then all bounds remain unchanged. This is because the bounds capture the decisions implied by all causal models compatible with the observed data and therefore also the decisions implied by a randomly drawn causal model from this set. In contrast, if the AI is not grounded and sometimes makes decisions based on a causal model that is not compatible with the data, the bounds might degrade.
>
> ***2. It would be more appropriate to treat their rationality as "bounded" in some sense (for example, supposing they choose actions that have approximately the highest utility, rather than exactly the highest utility, given their model).***
>
> This is correct -- if the AI chooses actions that have approximately the highest utility (under its own model) we would expect this additional source of uncertainty to degrade our ability to predict AI decision-making. If we understand the reviewer’s intuition correctly, while the AI’s beliefs on expected utilities remain unchanged, possibly the selection of actions given those beliefs may differ from exactly utility maximizing.
>
> In this case, our bounds, e.g., on the preference gap $\Delta$ that measures the relative expected utility benefit from a decision $d_1$ relative to a decision $d_0$, would remain unchanged as they only reflect the AI’s beliefs and not their choices. For exact expected utility maximizers, $\Delta > 0$ ensures that $d_1$ will be chosen over $d_0$ by the AI. For approximate expected utility maximizers this threshold might not be sufficient, and we might require $\Delta > c$ for some $c>0$ to conclude that $d_1$ will be chosen over $d_0$ by the AI. The magnitude of $c$ will depend on how loosely the AI follows the expected utility maximization principle. This setting is potentially more realistic, thank you for suggesting it.
>
> ***3. In footnote 4 the paper assumes access to a comprehensive specification of the AI system's behaviour in all possible contexts (if I understood this footnote correctly). There will be contexts in which we don't observe the system's behaviour in the data.***
>
> With this footnote we are trying to convey the possible challenges in estimation of bounds from finite samples. Our results are given in the infinite-sample regime, in terms of probability distributions, that may be difficult to estimate accurately if not enough data is available. For example, if the bound requires the estimation of a probability $P(d, c)$ but the combination of decision $d$ and context $c$ is not observed in the data, we might need to introduce additional modelling assumptions and account for errors in estimation to provide practical guarantees.
>
> ***4. Sec. 5.1 and the contrast between goal misgeneralisation and approximate inner alignment.***
>
> Right, thank you for this explanation. Perhaps “approximate inner alignment” is a better framing for that example.
>
> Thank you for the careful reading of our paper and for pointing out typos.

---

> > ### Comment · Reviewer_Q6s6 · 2025-04-05
> >
> > I thank the authors for their rebuttals. I have considered this rebuttal and the discussion with other reviewers, and I'm happy to keep my positive score, with the following notes.
> >
> > 1. **Assumptions.** One of the main concerns of reviewer EHMW (2 weak reject) appears to be that the assumptions limit the practical implications of the results. In my opinion, the paper does a sufficient job of clearly stating the assumptions behind the main proofs, and has an extensive discussion with concrete examples of the ways in which the results might change if each assumption is relaxes. If the discussion from this rebuttal can be incorporated, then that discussion will be the more robust for it.
> >
> > 2. **Applicability to LLMs.** There also seems to be some concern about whether the results would describe practical LLMs. This rebuttal has cited some evidence of LLMs decision-making following rationality axioms. The rebuttal to EHMW (2 weak reject) sketches a proof of concept LLM experiment. I know the cited studies of LLM rationality, and I don't find them especially compelling. The experiment results sketched in the rebuttal seems sensible, but this investigation is very light, and does not necessarily reflect the way LLMs 'make decisions' in practice.
> >
> >    However, in my opinion, maybe it should not be necessary for this paper to demonstrate practical applicability. The paper answers some foundational questions about behaviour identifiability for agents reasoning with causal models. This is basic research that can frame future work on bounding system behaviour in more practical settings. For example, if you want to develop algorithms for 'IRL for predicting behaviour, but for SCMs' you need to know to what extent you can identify the behaviour in the limit. Is this enough of a contribution for an ICML paper? I leave this to be decided ultimately by the AC.
> >
> > 3. **Relationship to prior work.** On the topic of IRL identifiability, the reviewer Zntn (3 weak accept) points out that there could be further discussion of related work with similar goals. IRL is usually motivated in terms of learning a reward function from (human) expert demonstrations in order to subsequently optimise that reward function with RL methods. In contrast, this paper aims to identify an SCM of an (AI) agent in order to predict its future behaviour. Of course, IRL methods can be applied to AI systems to reveal information relevant to predicting their behaviour, and one could aim to learn an SCM of a human expert from their behaviour too. The situation with respect to partial identifiability is the same. Therefore, I agree that there is a relation, and the paper would be improved by including a discussion of this work.
> >
> >    One additional reference on IRL identifiability is [1], though there are also others, this one is perhaps especially relevant as it looks not only at identifying reward functions from expert demonstrations but also from other data (such as binary choices) and also looks at identifying downstream factors such as behaviour. The formalism is still RL, rather than SCMs.
> >
> > 4. **On terminology "out-of-distribution":**
> >    * Following the thread in my review about goal misgeneralization. I take the authors suggesting to adopt my suggested terminology of "approximate inner alignment". I think this would be an improvement, but leave the final choice to the authors.
> >
> >    * Either way, something I am still uncertain about is the usage of the term 'out of distribution (OOD)'. In goal misgeneralization, 'OOD' usually means 'in new states or environment configurations that were not seen during training'. Let me call this 'state OOD'. I believe this is a generalisation of usage in supervised learning, where one considers 'input OOD'. In the present paper, there is the assumption (see footnote 4) that all states/environment configurations have been observed. Therefore, there is nothing that counts as state OOD. Instead, the authors use 'OOD' to mean *under a novel intervention* ('intervention OOD'?). This seems meaningfully different from state OOD. It's possible that intervention OOD is an established concept in SCM literature? If not, I invite the authors to consider switching to a new term to avoid confusion.
> >
> >    * Finally, I note that this bears on the discussion with reviewer Zntn about related work. The authors say "Perhaps it is fair to say that our bounds thus more concretely characterize the limits of what can be predicted about agent behaviour [OOD] which then complements the (in our understanding mostly in-distribution) partial identification results in IRL and decision theory." While I don't know the literature comprehensively, this seems fair to me, but *only* for *intervention OOD*. For example, [1] gives some negative state-OOD identifiability results (see their concept of a "transition mask").
> >
> > References:
> >
> > * [1] Skalse et al., "Invariance in Policy Optimisation and Partial Identifiability in Reward Learning", ICML 2023.

---

> > > ### Author Response · Authors · 2025-04-07
> > >
> > > Thank you for following up on our responses.
> > >
> > > Indeed, we use the term **out-of-distribution** (OOD) to mean "intervention" OOD, or more generally the problem of extrapolating from the data observed in one domain to another domain in which some of the underlying "causal mechanisms" in the SCM might have changed (due to an intervention or a more general shift), closer to "input" OOD in supervised learning. In the causality literature this problem is also known as *transportability* [2] and generally assumes positive probability for all events. In light of [1] and related work in goal misgeneralization, it seems appropriate to describe the contrast between "input" and "state" OOD more specifically -- we will make this clearer, thank you.
> > >
> > > [1] Skalse et al., "Invariance in Policy Optimisation and Partial Identifiability in Reward Learning", ICML 2023.
> > >
> > > [2] Bareinboim, Elias, and Judea Pearl. "Causal inference and the data-fusion problem." PNAS 2016.

---

### Decision · Program_Chairs · 2025-05-01

**Decision:**

Accept (poster)

**Comment:**

The paper presents and analyzes a model in which AI agents' beliefs are inferred from their behaviors. The authors' goal is to shed light on safety properties of AI agents through the language of structural causal models (SCMs). At a high level, the authors provide (partial) identifiability results showing what inferences we can make about an AI agent's SCM based on its behaviors.

The reviewers generally had a positive response to the paper. Some questions were raised as to the applicability of theoretical models here; I tend to be sympathetic towards modeling exercises, and I view these results as providing a framework within which to reason about inferences possible under idealized conditions. On the other hand, it's not clear to me that this framework will ever be directly applicable to real-world AI systems, and it might remain a conceptual tool. Still, the paper appears to be a valuable contribution to the literature.

The reviewers raised a number of questions, which the authors addressed during the rebuttal period. I encourage them to consider the reviewers' comments and suggestions in future revisions.